# Bures-Wasserstein Flow Matching for Graph Generation

**Keyue Jiang**[1,2]    **Jiahao Cui**[1,3]    **Xiaowen Dong**[4]    **Laura Toni**[1,2]

[1] AI Centre, University College London
[2] Department of Electronic and Electrical Engineering, University College London
[3] Department of Computer Science, University College London
[4] Department of Engineering Science, University of Oxford
`keyue.jiang.18@ucl.ac.uk`

## Abstract

Graph generation has emerged as a critical task in fields ranging from drug discovery to circuit design. Contemporary approaches, notably diffusion and flow-based models, have achieved solid graph generative performance through constructing a probability path that interpolates between reference and data distributions. However, these methods typically model the evolution of individual nodes and edges independently and use linear interpolations in the disjoint space of nodes/edges to build the path. This disentangled interpolation breaks the interconnected patterns of graphs, making the constructed probability path irregular and non-smooth, which causes poor training dynamics and faulty sampling convergence. To address the limitation, this paper first presents a theoretically grounded framework for probability path construction in graph generative models. Specifically, we model the joint evolution of the nodes and edges by representing graphs as connected systems parameterized by Markov random fields (MRF). We then leverage the optimal transport displacement between MRF objects to design a smooth probability path that ensures the co-evolution of graph components. Based on this, we introduce BWFlow, a flow-matching framework for graph generation that utilizes the derived optimal probability path to benefit the training and sampling algorithm design. Experimental evaluations in plain graph generation and molecule generation validate the effectiveness of BWFlow with competitive performance, better training convergence, and efficient sampling.

## 1 Introduction

Thanks to the capability of graphs in representing complex relationships, graph generation (Zhu et al., 2022; Liu et al., 2023a) has become an essential task in various fields such as protein design (Ingraham et al., 2019), drug discovery (Bilodeau et al., 2022), and social network analysis (Li et al., 2023). Among contemporary generative models, diffusion and flow models have emerged as two compelling approaches for their ability to achieve state-of-the-art performance in graph generation (Niu et al., 2020; Vignac et al., 2023a; Eijkelboom et al., 2024; Qin et al., 2024; Hou et al., 2024). In particular, these generative models can be unified under the framework of stochastic interpolation (Albergo & Vanden-Eijnden, 2023), which consists of four procedures (Lipman et al., 2024): 1) Drawing samples from the reference (source) distribution $p_0(\cdot)$ and/or the data (target) distribution $p_1(\cdot)$ for training set assembly; 2) Constructing a time-continuous probability path $p_t(\cdot), 0 \le t \le 1$ interpolating between $p_0$ and $p_1$; 3) Training a model to reconstruct the probability path by either approximating the score function or velocity fields (ratio matrix in the discrete case); and 4) sampling from $p_0$ and transforming it through the learned probability path to get samples that approximately follow $p_1$.

A core challenge in this framework is constructing the probability path $p_t$. Existing text and image generative models, operating either in the continuous (Ho et al., 2020; Song et al., 2021; Lipman et al., 2023; Liu et al., 2023b) or discrete (Campbell et al., 2022; Sun et al., 2023; Campbell et al., 2024; Gat et al., 2024; Minello et al., 2025) space, typically rely on linear interpolation between source and target distributions to construct the path. Graph generation models, including diffusion (Niu et al.,

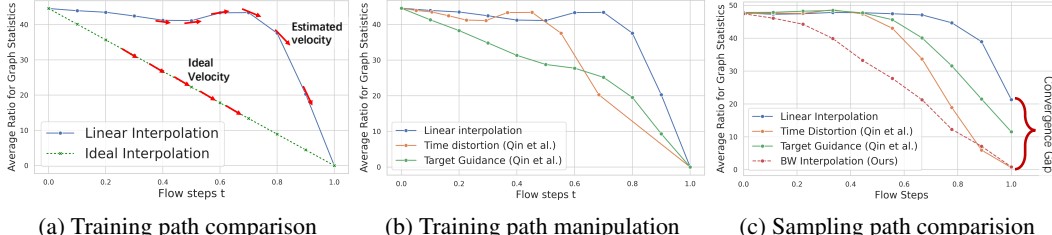

(a) Training path comparison  (b) Training path manipulation  (c) Sampling path comparision

Figure 1: Probability path visualization. Since the probability is intractable, the average maximum mean discrepancy ratio (y-axis) of graph statistics between interpolants and the data points is used as a proxy for the probability. Lower means closer to the data distribution (details in Section I.6).

2020; Vignac et al., 2023a; Haefeli et al., 2022; Xu et al., 2024; Siraudin et al., 2024) and flow-based models (Eijkelboom et al., 2024; Qin et al., 2024; Hou et al., 2024), inherit this design by modeling every single node and edge independently and linearly build paths in the disjoint space. However, we argue this approach to be inefficient because it neglects the strong interactions and relational structure inherent in graphs, i.e., the significance of a node heavily depends on the configuration of its neighbors. We first show the negative impact of linear path through a motivating example.

**The limitation of linear interpolation.** In flow models[1], intuitively, the velocity is trained by approximating the tangent of the probability path in the training set, while generation follows the learned velocity to reconstruct the path toward the data distribution. The blue line in Fig. 1a illustrates the training path obtained by linear interpolation: it remains flat until a transition point at $t \approx 0.8$[2], after which it drops sharply. This non-smooth path leads to poor velocity estimation (red arrow) as: 1) the critical transition region $0.8 < t < 1$ is less explored, causing potential underfitting, and 2) the velocities trained in the flat region $t < 0.8$ fail to guide the model toward the target distribution. Consequently, sampling struggles to converge to the data distribution as shown in Fig. 1c. Ideally, we need a smooth probability path like the green line in Fig. 1a to ensure that at every time $t$, the model takes stable, meaningful steps toward the target distribution. More evidence in App. I.5.

We attribute this issue to the linear path construction that fails to capture global co-evolution (Haasler & Frossard, 2024) of the graph components, and cannot guarantee an optimal transport displacement between non-Euclidean graph distributions (Formal explanations in Section 2.2). Thus, the constructed path is suboptimal with a sharp transition from reference to data distribution or even deviates from the valid graph domain (Kapusniak et al., 2024). Though not explicitly mentioned, Qin et al. (2024) mitigates this issue through heuristic strategies to smooth the path, which we conceptually visualize in Fig. 1b and discussed in App. F.1. This shows a potential benefit in manipulating the path and we aim at building a theoretically grounded framework for probability path construction.

**Proposed solution.** To this end, we draw on statistical relational learning and model graphs using Markov Random Fields (MRFs) (Taskar et al., 2007; Qu et al., 2019). MRFs organize the nodes/edges as an interconnected system and interpolating between two MRFs captures the joint evolution of the graph system. Extending Haasler & Frossard (2024), we derive a closed-form Wasserstein distance between graph distributions and leverage it to construct Bures-Wasserstein (BW) interpolation that ensures the OT displacement between graph objects. We then integrate these insights into a flow-matching framework called BWFlow. Specifically, BWFlow operates on smooth, globally coherent velocity fields, exclusively constructed by BW interpolation, to generate graphs (see Fig. 1c). Crucially, BWFlow admits simulation-free computation of densities and velocities along the entire path, which translates into efficient, stable training and sampling.

**Contributions. First**, observing that the linear interpolation used in existing models is suboptimal, we propose a theoretically grounded framework for probability path construction and velocity estimation in graph generation. **Second**, through parameterizing graphs as MRFs, we introduce BWFlow, a flow-matching model for graph generation that constructs probability paths respecting the graph geometry and develops smooth velocities without heuristic path manipulations. **Third**, BWFlow was

---

[1]This work focus on flow models and left the generalization to diffusions in App. F.2.

[2]The specific value is an empirical observation and does not have a theoretical significance. Fig. 6 illustrates that it differs across datasets.

tested on plain graph and molecule generation, exhibiting better performance and dynamics, such as fast and stable training convergence and efficient sampling.

## 2 PRELIMINARIES

### 2.1 FLOW MATCHING FOR GRAPH GENERATION

**Flow matching (FM).** Generative modeling considers fitting a mapping from state space $\mathcal{S} \to \mathcal{S}$ that transforms the samples from source distribution, $X_0 \sim p_0$, to samples from target data distribution, $X_1 \sim p_1$[3]. Continuous normalizing flow (Chen et al., 2018) parameterizes the transformation through a push-forward equation that interpolates between $p_0$ and $p_1$ and constructs a probability path $p_t(\mathcal{X}) = [\psi_t p_0](\mathcal{X})$ through a time-dependent function $\psi_t$ (a.k.a flow). A vector field $u_t$, defined as $\frac{d}{dt}\psi_t(\mathcal{X}) = u_t(\psi_t(\mathcal{X}))$ with $\psi_0(\mathcal{X}) = \mathcal{X}$, is said to generate $p_t$ if $\psi_t$ satisfies $X_t := \psi_t(X_0) \sim p_t$ for $X_0 \sim p_0$. The FM (Lipman et al., 2023) is designed to match the real velocity field through:

$$\mathcal{L}_{\text{FM}}(\theta) = \mathbb{E}_{t, X_t \sim p_t(\cdot)} \|v_\theta(X_t) - u_t(X_t)\|^2. \tag{1}$$

where $v_\theta(\cdot) : \mathcal{S} \to \mathcal{S}$ is the parameterized velocity field and $t \sim \mathcal{U}[0,1]$.

**Conditional flow matching (CFM).** Given that the actual velocity field and the path are not tractable (Tong et al., 2024), one can construct the per-sample conditional flow. We condition the probability paths on variable $Z \sim \pi(\cdot)$ (for instance, a pair of source and target points $Z = (X_0, X_1)$) and re-write $p_t(\mathcal{X}) = \mathbb{E}_{\pi(\cdot)} p_t(\mathcal{X} \mid Z)$ and $u_t(\mathcal{X}) = \mathbb{E}_{\pi(\cdot)} u_t(\mathcal{X} \mid Z)$ where the conditional path and the velocity field are tractable. The CFM aims at regressing a velocity $v_\theta(\cdot)$ to $u_t(\mathcal{X} \mid Z)$ by the loss,

$$\mathcal{L}_{\text{CFM}}(\theta) := \mathbb{E}_{t, Z \sim \pi(\cdot), p_t(\cdot|Z)} \|v_\theta(X_t) - u_t(X_t \mid Z)\|^2, \tag{2}$$

where it is shown that the CFM optimization has the same optimum as the FM (Tong et al., 2024).

**Graphs as statistical objects.** When considering graph generation with CFM, the very first step is to model graphs as statistical objects. For notation, we let $\mathcal{G} = \{\mathcal{V}, \mathcal{E}, \mathcal{X}\}$ denote an undirected graph random variable with edges $\mathcal{E} = \{e_{uv}\}$, nodes $\mathcal{V} = \{v\}$, and node features $\mathcal{X} = \{x_v\}$. A graph realization is denoted as $G = \{V, E, X\} \sim p(\mathcal{G})$. We consider a group of latent variables that controls the graph distribution, specifically the node feature mean $\boldsymbol{X} = [\boldsymbol{x}_1, \boldsymbol{x}_2, \ldots, \boldsymbol{x}_{|\mathcal{V}|}]^\top \in \mathbb{R}^{|\mathcal{V}| \times K}$, the weighted adjacency matrix $\boldsymbol{W} \in \mathbb{R}^{|\mathcal{V}| \times |\mathcal{V}|}$, and the Laplacian matrix $\boldsymbol{L} = \boldsymbol{D} - \boldsymbol{W} \in \mathbb{R}^{|\mathcal{V}| \times |\mathcal{V}|}$, with $\boldsymbol{D} = \text{diag}(\boldsymbol{W}\boldsymbol{1})$ being the degree matrix (and $\boldsymbol{1}$ the all-one vector). In a nutshell, graphs are sampled from $G \sim p(\mathcal{G}; \boldsymbol{G}) = p(\mathcal{X}, \mathcal{E}; \boldsymbol{X}, \boldsymbol{W})$.

**Generation with CFM.** The new graphs are sampled through iteratively building $G_{t+dt} = G_t + v_t^\theta(G_t) \cdot dt$ with initial $G_0 \sim p_0$ and a trained velocity field $v_t^\theta(G_t)$, so that the medium points follows $G_t \sim p_t(\mathcal{G})$ and terminates at $p_1$. The velocity can be trained either via numerical approximation (i.e., $v_t^\theta(G_t) \approx (G_{t+dt} - G_t)/dt$) or through $x$-prediction (Gat et al., 2024) which parameterize $v_t^\theta(G_t)$ as,

$$v_t^\theta(G_t) = \mathbb{E}_{G_0 \sim p_0(\mathcal{G}), G_1 \sim p_{1|t}^\theta(\cdot|G_t)} [v_t(G_t \mid G_0, G_1)] \tag{3}$$

As such, training the velocity fields is replaced by a denoiser $p_{1|t}^\theta(\cdot \mid G_t)$ to predict the clean datapoint, which is equivalent to maximizing the log-likelihood (Qin et al., 2024; Campbell et al., 2024),

$$\mathcal{L}_{\text{CFM}} = \mathbb{E}_{G_1 \sim p_1(\cdot), G_0 \sim p_0(\cdot), t \sim \mathcal{U}_{[0,1]}(\cdot), G_t \sim p_{t|0,1}(\cdot|G_1, G_0)} [\log p_{1|t}^\theta(G_1 \mid G_t)] \tag{4}$$

where $t$ is sampled from a uniform distribution $\mathcal{U}_{[0,1]}$ and $G_t \sim p_{t|0,1}$ can be obtained in a simulation-free manner. This framework avoids the evaluation of the conditional vector field at training time, which both increases the model robustness and training efficiency.

To proceed, a closed form of $p_t(\cdot \mid G_0, G_1)$ is required to construct both the probability path and the velocity field $v_t(G_t \mid G_0, G_1)$. A common selection to decompose the probability density assumes independency for each node and edge (Hou et al., 2024; Qin et al., 2024; Eijkelboom et al.,

---

[3]For clarity, we denote the calligraphic style $\mathcal{X}$ being the random variable, the plain $X$ the relevant realizations and the bold symbol $\boldsymbol{X}$ the latent variables (parameters) that controls the distributons, i.e. $X \sim p(\mathcal{X}; \boldsymbol{X})$.

2024) giving $p(\mathcal{G}) = p(\mathcal{X})p(\mathcal{E}) = \prod_{v \in \mathcal{V}} p(x_v) \prod_{e_{uv} \in \mathcal{E}} p(e_{uv})$. Choosing $\pi(\cdot) = p_0(\mathcal{G}) p_1(\mathcal{G})$, the boundary conditions follow $p_i(\mathcal{G}) = \delta(\mathcal{X}_i = \boldsymbol{X}_i) \cdot \delta(\mathcal{E}_i = \boldsymbol{W}_i), \forall i = \{0, 1\}$ with $\delta$ the dirac function. This decomposition is further combined with linear interpolation to build the path, as introduced in (Tong et al., 2024), where,

$$
\begin{aligned}
p_t(x_v \mid G_0, G_1) &= \alpha_t[X_1]_v + \sigma_t[X_0]_v, \text{ and } u_t(x_v \mid X_0, X_1) = [X_1]_v - [X_0]_v, \\
p_t(e_{uv} \mid G_0, G_1) &= \alpha_t[E_1]_{uv} + \sigma_t[E_0]_{uv} \text{ and } u_t(e_{uv} \mid G_0, G_1) = [E_1]_{uv} - [E_0]_{uv}.
\end{aligned}
\tag{5}
$$

where $\alpha_t$ and $\sigma_t$ are two parameters to make the boundary condition satisfied, the selection is discussed in App. F.2. Similarly, discrete flow matching frameworks for graph generation (Qin et al., 2024; Siraudin et al., 2024; Xu et al., 2024) is also based on linear interpolation, where the interpolant is sampled from a categorical distribution whose probabilities are simply linear interpolation between the boundary conditions.

## 2.2 WHY WE NEED MORE THAN LINEAR INTERPOLATIONS FOR GRAPH GENERATION?

**When we can use linear interpolation?** Existing literature (Liu et al., 2023b; Albergo & Vanden-Eijnden, 2023) argues that the probability path $p_t(\mathcal{X} \mid Z)$ should be chosen to recover the optimal transport (OT) displacement interpolant (McCann, 1997). The (Kantorovich) optimal transport problem is to find the transport plan between two probability measures, $\eta_0$ and $\eta_1$, with the smallest associated transportation cost defined as follows.

> **Definition 1** (Wasserstein Distance). Denote the possible coupling as $\pi \in \Pi(\eta_0, \eta_1)$, which is a joint measure on $\mathcal{S} \times \mathcal{S}$ whose marginals are $\eta_0$ and $\eta_1$ respectively. With $c(X, Y)$ being the cost of transporting the mass between $X$ and $Y$, the Wasserstein distance is defined as,
> $$
> \mathcal{W}_c(\eta_0, \eta_1) = \inf_{\pi \in \Pi(\eta_0, \eta_1)} \int_{\mathcal{S} \times \mathcal{S}} c(X, Y) \, d\pi(X, Y). \tag{6}
> $$

*When the data follow Euclidean geometry and both boundary distributions $p_0, p_1$ follow isotropic Gaussians*, the path shown in Eq. (5) with $\sigma_t \to 0$ becomes a solution to Eq. (6) (Tong et al., 2024).

However, given that graphs are non-Euclidean and interconnected objects violating the aforementioned conditions, linearly interpolating nodes/edges with Eq. (5) cannot guarantee the OT displacement in graph generation. Blindly using the approach will result in suboptimal probability path and lead to a problematic velocity estimation (Chen & Lipman, 2024; Kapusniak et al., 2024). To illustrate, recall that the velocity is trained via either approximating a) $(G_{t+dt} - G_t)/dt$ or b) $(G_1 - G_t)/(1 - t)$. For both strategies, approximating the path similar to Fig. 1a exposes two issues: 1) Most of the training points ($G_t$) center around areas with high ratio, while the critical part for sampling is the region with intermediate ratio, i.e. the points corresponding to $0.8 < t < 1$. The velocity model has the risk of underfitting in those regions, posing a risk when deployed for sampling. 2) Especially for strategy a, when $t$ is small, the velocity through numerical approximation $(G_{t+dt} - G_t)/dt$ may not even point correctly to the data distribution. Thus, when sampling, the model is difficult to determine the correct direction in the early stage, which will lead to convergence failure.

To establish a good velocity estimation that yields better training and generation dynamics, an ideal training probability path should: 1) adequately explore the landscape of $G_t$ so that the velocity at intermediate points is well-trained. 2) correctly estimate the velocity pointing to the data distribution. It is worth noting that the superior performance achieved by previous work (Siraudin et al., 2024; Qin et al., 2024) is partially attributed to their implicit manipulation of the path to satisfy these conditions. The techniques used, including target guidance, time distortion, and stochasticity injection, are conceptually visualized in Fig. 1b with discussions in App. F.1.

## 3 METHODOLOGY

In this paper, we aim to build a theoretically grounded framework for probability path construction in graph generative models without the reliance on heuristic path manipulations. To this end, we introduce Bures–Wasserstein Flow Matching (BWFlow), a novel graph generation framework that is built upon the OT displacement when modeling graphs with Markov Random Fields (MRFs). We begin by casting graphs in an MRF formulation in Section 3.1. We then derive the BWFlow

framework in Section 3.2 by formulating and solving the OT displacement problem on the MRF, thereby yielding the fundamental components, interpolations and velocity fields, for FM-based graph generation. Finally, in Section 3.3, we extend BWFlow to discrete FM regimes, enabling its application across a broad spectrum of graph-generation tasks. A schematic overview of the entire BWFlow is illustrated in Fig. 2.

## 3.1 GRAPH MARKOV RANDOM FIELDS

We borrow the idea from MRF as a remedy to modeling the complex system organized by graphs, which intrinsically captures the underlying mechanism that jointly generates the nodes and edges. Mathematically, we assume the joint probability density distribution (PDF) of node features and graph structure as $p(\mathcal{G}; \boldsymbol{G}) = p(\mathcal{X}, \mathcal{E}; \boldsymbol{X}, \boldsymbol{W}) = p(\mathcal{X}; \boldsymbol{X}, \boldsymbol{W})p(\mathcal{E}; \boldsymbol{W})$ where the node features and graph structure are interconnected through latent variables $\boldsymbol{X}$ and $\boldsymbol{W}$. For node features $\mathcal{X}$, we follow the MRF assumption in Zhu et al. (2003) and decompose the density into the node-wise potential $\varphi_1(v), \forall v \in \mathcal{V}$ and pair-wise potential $\varphi_2(u, v), \forall e_{uv} \in \mathcal{E}$:

$$p(\mathcal{X}; \boldsymbol{X}, \boldsymbol{W}) \propto \prod_v \underbrace{\exp\left\{-(\nu + d_v)\|\boldsymbol{V}x_v - \boldsymbol{\mu}_v\|^2\right\}}_{\varphi_1(v)} \prod_{u,v} \underbrace{\exp\left\{w_{uv}\left[(\boldsymbol{V}x_u - \boldsymbol{\mu}_u)^\top(\boldsymbol{V}x_v - \boldsymbol{\mu}_v)\right]\right\}}_{\varphi_2(u,v)},$$

(7)

with $\|\cdot\|$ the $L_2$ norm, $(\cdot)^\dagger$ the pseudo-inverse, $\boldsymbol{V}$ the transformation matrix modulating the graph feature emission, and $\boldsymbol{\mu}_v$ the node-specific latent variable mean. Eq. (7) can be expressed as a colored Gaussian distribution in Eq. (8) given that $\boldsymbol{V}x_v \sim \mathcal{N}(\boldsymbol{\mu}_v, (\nu\boldsymbol{I} + \boldsymbol{L})^{-1})$. We further assume that edges are emitted via a Dirac delta, $\mathcal{E} \sim \delta(\boldsymbol{W})$, yielding our definition of Graph Markov Random Fields (GraphMRF). The derivation can be found in App. A.2.

---

**Definition 2** (Graph Markov Random Fields). GraphMRF statistically describes graphs as,

$$p(\mathcal{G}; \boldsymbol{G}) = p(\mathcal{X}, \mathcal{E}; \boldsymbol{X}, \boldsymbol{W}) = p(\mathcal{X}; \boldsymbol{X}, \boldsymbol{W}) \cdot p(\mathcal{E}; \boldsymbol{W}) \text{ where } \mathcal{E} \sim \delta(\boldsymbol{W}) \text{ and}$$
$$\text{vec}(\mathcal{X}) \sim \mathcal{N}\left(\boldsymbol{X}, \boldsymbol{\Lambda}^\dagger\right), \text{ with } \boldsymbol{X} = \text{vec}(\boldsymbol{V}^\dagger\boldsymbol{\mu}), \boldsymbol{\Lambda} = (\nu\boldsymbol{I} + \boldsymbol{L}) \otimes \boldsymbol{V}^\top\boldsymbol{V}.$$

(8)

The $\otimes$ is the Kronecker product, $\text{vec}(\cdot)$ is the vectorization operator and $\boldsymbol{I}$ is the identity matrix.

---

**Remark 1.** GraphMRF explicitly captures node–edge dependencies and preserves the advantages of colored Gaussian distributions. Section 3.2 will soon show that this yields closed-form interpolation and velocity, and the probability path constructed from GraphMRFs remains on the graph manifold that respects the underlying non-Euclidean geometry.

**Remark 2.** We emphasize that transforming a graph into the MRF domain actually enhances the modeling ability of global information encoded in the low-frequency part of graph spectra. This parallels the behavior observed in diffusion models with latent space, where latent representations retain a larger proportion of low-frequency information, which is proven helpful in generative models. We refer to App. A.3 for a discussion.

---

## 3.2 BURES-WASSERSTEIN FLOW MATCHING FOR GRAPH GENERATION

**The optimal transport displacement between graph distributions.** Given that the joint probability of graphs decomposed as $p(\mathcal{G}) = p(\mathcal{X}; \boldsymbol{X}, \boldsymbol{W})p(\mathcal{E}; \boldsymbol{W})$ and the measure factorized to $\eta_{\mathcal{G}_j} = \eta_{\mathcal{X}_j} \cdot \eta_{\mathcal{E}_j}$ with $j \in \{0, 1\}$, the graph Wasserstein distance between $\eta_{\mathcal{G}_0}$ and $\eta_{\mathcal{G}_1}$ is written as,

(Graph Wasserstein Distance) $\quad d_{\text{BW}}(\mathcal{G}_0, \mathcal{G}_1) \coloneqq \mathcal{W}_c(\eta_{\mathcal{G}_0}, \eta_{\mathcal{G}_1}) = \mathcal{W}_c(\eta_{\mathcal{X}_0}, \eta_{\mathcal{X}_1}) + \mathcal{W}_c(\eta_{\mathcal{E}_0}, \eta_{\mathcal{E}_1}).$

We extend Haasler & Frossard (2024) and analytically derive the graph Wasserstein distance using the OT formula between Gaussians Dowson & Landau (1982); Olkin & Pukelsheim (1982); Takatsu (2010) (see Lemma 2 proved in App. B.1) as follows.

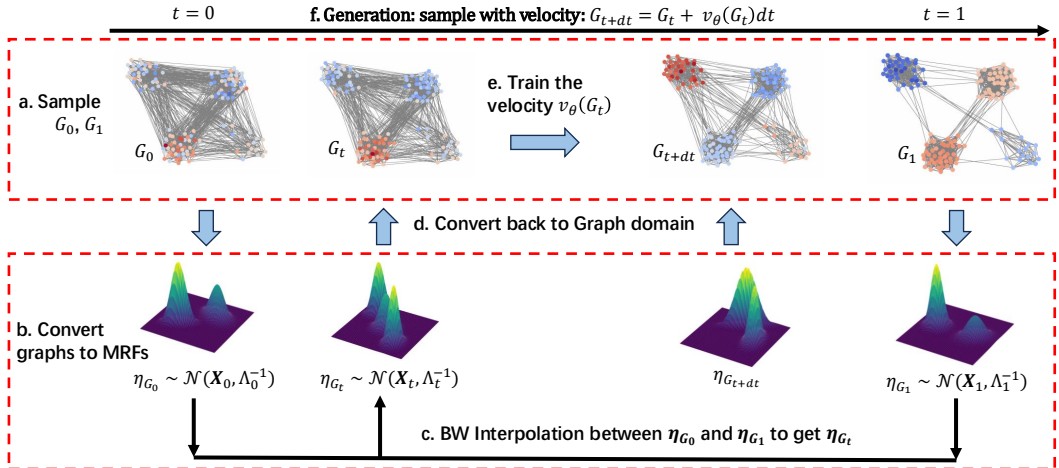

Figure 2: Schematic overview of BWFlow, which consists of: a) Sample the marginal graph condition $G_0$ and $G_1$; b) Convert graphs to MRFs; c) Interpolate to get intermediate points; d) Convert back to get $G_t$; e) Train velocity based on $G_t$; and f) Generate new points with the trained velocity.

**Proposition 1** (Bures-Wasserstein Distance). *Consider two same-sized graphs $\mathcal{G}_0 \sim p(\mathcal{X}_0, \mathcal{E}_0)$ and $\mathcal{G}_1 \sim p(\mathcal{X}_1, \mathcal{E}_1)$ with $\mathbf{V}$ shared for two graphs, described by the distribution in Definition 2. When the graphs are equipped with graph Laplacian matrices $\mathbf{L}_0$ and $\mathbf{L}_1$ satisfying 1) is Positive Semi-Definite (PSD) and 2) has only one zero eigenvalue. The Bures-Wasserstein distance between these two random graph distributions is given by*

$$d_{BW}(\mathcal{G}_0, \mathcal{G}_1) = \|\mathbf{X}_0 - \mathbf{X}_1\|_F^2 + \beta \operatorname{trace}\left(\mathbf{L}_0^\dagger + \mathbf{L}_1^\dagger - 2\left(\mathbf{L}_0^{\dagger/2}\mathbf{L}_1^\dagger\mathbf{L}_0^{\dagger/2}\right)^{1/2}\right), \quad (9)$$

*as $\nu \to 0$ and $\beta$ is a constant related to the norm of $\mathbf{V}^\dagger$. The proof can be found in Section B.2.*

Based on the Bures-Wasserstein (BW) distance, we then derive the OT interpolant for two graphs, which is the solution of the displacement minimization problem described as,

$$\mathcal{G}_t = \arg\min_{\tilde{\mathcal{G}}} (1-t)d_{\mathrm{BW}}(\mathcal{G}_0, \tilde{\mathcal{G}}) + t d_{\mathrm{BW}}(\tilde{\mathcal{G}}, \mathcal{G}_1). \quad (10)$$

**The probability path.** The interpolation is obtained through solving Eq. (10) with the BW distance defined in Proposition 1, we prove the minimizer of the above problem has the form in Proposition 2. The proof can be found in App. C.1.

**Proposition 2** (Bures-Wasserstein interpolation). *The graph minimizer of Eq. (10), $\mathcal{G}_t = \{\mathcal{V}, \mathcal{E}_t, \mathcal{X}_t\}$, have its node features following a colored Gaussian distribution, $\mathcal{X}_t \sim \mathcal{N}(\mathbf{X}_t, \mathbf{\Lambda}_t^\dagger)$ with $\mathbf{\Lambda}_t = (\nu \mathbf{I} + \mathbf{L}_t) \otimes \mathbf{V}^\top \mathbf{V}$ and edges following $\mathcal{E}_t \sim \delta(\mathbf{W}_t)$, specifically,*

$$\mathbf{L}_t^\dagger = \mathbf{L}_0^{1/2}\left((1-t)\mathbf{L}_0^\dagger + t\left(\mathbf{L}_0^{\dagger/2}\mathbf{L}_1^\dagger\mathbf{L}_0^{\dagger/2}\right)^{1/2}\right)^2 \mathbf{L}_0^{1/2}, \quad \mathbf{X}_t = (1-t)\mathbf{X}_0 + t\mathbf{X}_1 \quad (11)$$

The interpolant provides a closed form for the induced probability path $p(\mathcal{G}_t \mid G_0, G_1)$ and the velocity $v(\mathcal{G}_t \mid G_0, G_1)$ that is easy to access without any simulation.

**The velocity.** We consider the reparameterization as in Eq. (3) and derive the conditional velocity $v_t(G_t \mid G_1, G_0)$ as in Proposition 3.

> **Proposition 3** (Bures-Wasserstein velocity). *For the graph $\mathcal{G}_t$ following BW interpolation in Proposition 2, the conditional velocity at time $t$ with observation $G_t$ is given as,*
>
> $$v_t(E_t \mid G_0, G_1) = \dot{W}_t = diag(\dot{L}_t) - \dot{L}_t, \quad v_t(X_t \mid G_0, G_1) = \frac{1}{1-t}(X_1 - X_t) \tag{12}$$
>
> $$with \; \dot{L}_t = 2L_t - TL_t - L_tT \; and \; T = L_0^{1/2}(L_0^{\dagger/2} L_1^{\dagger} L_0^{\dagger/2})^{1/2} L_0^{1/2}$$
>
> *where $W_t = D_t - L_t$ and $L_t$ defined in Eq. (11). Derivation can be found in Section C.2.*

With Proposition 2 and Proposition 3, we are now able to formally construct the algorithms for Bures-Wasserstein flow matching. Taking continuous flow matching as an example, Algorithm 1 and 2 respectively introduce the training and sampling pipelines for our BWFlow.

> **Remark**: Similar to denoiser/noise-prediction parameterization, there exist multiple ways to establish or numerically approximate the BW interpolation and velocity for training and inference. The choice will have an impact on training and sampling dynamics, such as stability and efficiency. We provide a discussion of the design space and the trade-offs in App. E.

### 3.3 DISCRETE BURES-WASSERSTEIN FLOW MATCHING FOR GRAPH GENERATION

Up to now we are working on the scenario when $p(\mathcal{X} \mid X, W)$ is a Gaussian and $p(\mathcal{E} \mid W)$ is a Dirac distribution. However, previous studies have observed a significant improvement of the discrete counterpart of the continuous graph generation models Vignac et al. (2023a); Xu et al. (2024); Qin et al. (2024). To benefit our model from such a nature, we derive the discrete Bures-Wasserstein flow matching for graph generation.

**The discrete probability path.** We design the probability path as discrete distributions,

$$p_t(x_v \mid G_0, G_1) = \text{Categorical}([X_t]_v), \quad p_t(e_{uv} \mid G_0, G_1) = \text{Bernoulli}([W_t]_{uv})$$
$$\text{s.t. } p_0(\mathcal{G}) = \delta(G_0, \cdot), p_1(\mathcal{G}) = \delta(G_1, \cdot) \tag{13}$$

where $W_t = D_t - L_t$ with $X_t$ and $L_t$ defined the same in Eq. (11). We consider the fact that the Dirac distribution is a special case when the Categorical/Bernoulli distribution has probability 1 or 0, so the boundary condition $p_0(\mathcal{G}) = \delta(G_0, \cdot), p_1(\mathcal{G}) = \delta(G_1, \cdot)$ holds. Even though we are not sampling from Gaussian distributions anymore, it is possible to approximate the Wasserstein distance between two multivariate discrete distributions with the Gaussian counterpart so the conclusions, such as optimal transport displacements, still hold. We left the discussion in App. D.2.

**The discrete velocity fields.** The path of node features $\mathcal{X}_t$ can be re-written as $p_t(\mathcal{X}) = (1 - t)\delta(\cdot, X_0) + t\delta(\cdot, X_1)$ so the conditional velocity can be accessed through $v_t(X_t \mid G_0, G_1) = [\delta(\cdot, X_1) - \delta(\cdot, X_t)]/(1-t)$. However, the probability path of edges $\mathcal{E}_t$, shown in Eqs. (11) and (13), cannot be written as a mixture of two boundary conditions given the non-linear interpolation. To this end, we derive in App. D.3 that the discrete velocity follows,

$$v_t(E_t \mid G_1, G_0) = (1 - 2E_t) \frac{\dot{W}_t}{W_t \circ (1 - W_t)}, \tag{14}$$

where $W_t = D_t - L_t$, $\dot{W}_t = diag(\dot{L}_t) - \dot{L}_t$ with $L_t, \dot{L}_t$ defined in Eqs. (11) and (12) respectively. With the interpolation and velocity defined, the discrete flow matching is built in Algorithms 3 and 4.

## 4 EXPERIMENTS

We evaluate BWFlow through both the plain graph generation and real-world molecule generation tasks. We first outline the experimental setup in Section 4.1, followed by a general model comparison in Section 4.2. Next, we conduct behavior analysis in Section 4.3 to understand the superior training/sampling dynamics BWFlow can bring.

---

**Algorithm 1:** BWFlow Training

**Input:** Ref. dist $p_0$ and dataset $\mathcal{D} \sim p_1$.
**Output:** Trained model $f_\theta(G_t, t)$.

1  Initialize model $f_\theta(G_t, t)$;
2  **while** $f_\theta$ *not converged* **do**
3      Sample batched $\{G_0\} \sim p_0, \{G_1\} \sim \mathcal{D}$;
    `/* Construct Prob.path    */`
4      Sample $t \sim \mathcal{U}(0, 1)$;
5      Calculate the BW interpolation
    $p(G_t \mid G_0, G_1)$ via Eq. (11);
    `/* x-prediction           */`
6      $p_{1|t}^\theta(\cdot \mid G_t) \leftarrow f_\theta(G_t, t)$;
7      Loss calculation via Eq. (4);
8      optimizer.step();

---

**Algorithm 2:** BWFlow Sampling

**Input:** Reference distribution $p_0$, Trained Model
    $f_\theta(G_t, t)$, Small time step dt,
**Output:** Generated Graphs $\{\hat{G}_1\}$.

1  Initialize samples $\{\hat{G}_0\} \sim p_0$;
2  Initialize the model $p_{1|t}^\theta(\cdot \mid G_t) \leftarrow f_\theta(G_t, t)$ **for**
    $t \leftarrow 0$ **to** $1 - dt$ ***by*** $dt$ **do**
    `/* x-prediction           */`
3      Predict $\tilde{G}_1 \leftarrow p_{1|t}^\theta(\cdot \mid \hat{G}_t)$;
    `/* Velocity calculation    */`
4      Calculate $v_\theta(\hat{G}_t \mid \hat{G}_0, \tilde{G}_1)$ via Eq. (12);
    `/* Numerical Sampling      */`
5      Sample $\hat{G}_{t+dt} \sim \hat{G}_t + v_\theta(\hat{G}_t)dt$

---

## 4.1 EXPERIMENT SETTINGS

**Dataset.** We evaluate the models' ability of plain graph generation on three benchmark datasets following Martinkus et al. (2022); Vignac et al. (2023a); Bergmeister et al. (2024), specifically, **planar**, **tree**, and **SBM** (stochastic blocking models) graphs. Two datasets, namely MOSES (Polykovskiy et al., 2018) and GUACAMOL (Brown et al., 2019), are benchmarked to test the model performance on 2D molecule generation. For 3D molecule generation with coordinate data, we test the model on QM9 (Ramakrishnan et al., 2014) and GEOM-DRUGS (Axelrod & Gómez-Bombarelli, 2020).

**Metrics.** In plain graph generation, the evaluation metrics include the percentage of Valid, Unique, and Novel (**V.U.N.**) graphs, and the average maximum mean discrepancy ratio (**A.Ratio**) of graph statistics between the set of generated graphs and the test set are reported (details in Section I.6). For molecule generation, we test two scenarios with and without bond type information, where the latter validates the capacity of our methods in generating the graph structures. To this end, we develop a new relaxed metric to measure the stability and validity of atoms and molecules when bond types are not available. Specifically, the atom-wise stability is relaxed as:

$$\text{Stability of Atom } i\text{: } s_i = \mathbb{I}[\exists \{b_{ij}\}_{j \in \mathcal{N}_i} \in \prod_{j \in \mathcal{N}_i} B_{ij} : \sum_{j \in \mathcal{N}_i} b_{ij} = \text{EV}_i], \text{ with the identity function } \mathbb{I}.$$

This means atom $i$ is "relaxed-stable" if there is at least one way to pick allowed bond types ($B_{ij}$) to its neighbors $\mathcal{N}_i$ so that their total exactly matches the expected valences $\text{EV}_i$. Such a relaxed stability of atoms (**Atom.Stab.**) inherently defines molecule stability (**Mol.Stab.**) and the **validity** of a molecule. In addition to these metrics, distribution metrics including charge distributions and total variation for atom and angles are also used. Details in App. I.6.

**Setup.** To isolate the impact from model architecture, we follow Qin et al. (2024) to fix the backbone model as the same graph transformers. To fairly compare the impact of probability path construction, we disabled the path manipulation strategies such as time distortion and target guidance from Qin et al. (2024) and predictor-corrector in Siraudin et al. (2024) (the general comparison with all strategies enabled is left in App. I.2). More experimental details can be found in App. I.1.

## 4.2 MAIN RESULTS FOR GRAPH GENERATION

**Plain graph generation.** Given that the training dynamics on these benchmarks are extremely unstable and performance continues to fluctuate significantly even after convergence (the transparent jagged curve in Fig. 3c gives a visualization), we calculate the average results over the last 5 checkpoints (CAVG) to reflect the model behaviour after convergence. The exponentially moving average (EMA) with decay 0.999 is applied to the model. As illustrated in Table 1, BWFlow consistently outperforms other benchmarks in terms of A.Ratio and exceeds most competitors on Planar and SBM in V.U.N. The lone exception is the tree graphs, where our model falls short. We attribute this gap to the fundamentally different spectral pattern for tree graphs, which we provide a discussion in App. A.3.

Table 1: Plain graph generation performance. The path manipulation methods, e.g. target guidance in Qin et al. (2024) and predictor-corrector in Siraudin et al. (2024), are disabled to purely evaluate the impact of path construction. This table unifies the path distortion designs as in Table 10 and presents the CAVG results. We reproduce the state-of-the-art diffusion/flow model for comparison, while other models evaluated on best-checkpoint results are in the Table 11. The full statistics in Table 13.

| Model | Class | Planar | | Tree | | SBM | |
|---|---|---|---|---|---|---|---|
| | | V.U.N. ↑ | A.Ratio ↓ | V.U.N. ↑ | A.Ratio ↓ | V.U.N. ↑ | A.Ratio ↓ |
| Train set | — | 100 | 1.0 | 100 | 1.0 | 85.9 | 1.0 |
| DiGress (CAVG) (Vignac et al., 2023a) | Diffusion | 61.5±10.1 | 9.9 ±3.3 | 56.0 ±11.0 | 8.9±3.2 | 56.0±8.5 | 3.5±0.5 |
| DisCo (CAVG) (Xu et al., 2024) | Diffusion | 57.5± 2.5 | 9.0± 1.4 | / | / | 55.0± 5.9 | 11.6±2.9 |
| HSpectre (Bergmeister et al., 2024) | Diffusion | 67.5 | 3.0 | 82.5 | 2.1 | 75.0 | 10.5 |
| GruM (CAVG) (Jo et al., 2024) | Diffusion | 74.4±5.15 | 3.2±0.4 | 52.5±3.2 | 2.4±0.7 | 73.5±6.7 | 2.6±0.6 |
| Cometh (CAVG) (Siraudin et al., 2024) | Diffusion | 80.5± 5.79 | 3.0± 0.6 | **84.5**± 7.8 | 2.0± 0.4 | 77.5± 5.7 | 4.7± 0.6 |
| DeFoG (CAVG) (Qin et al., 2024) | Flow | 77.5±8.37 | 3.5±1.7 | 83.5±10.8 | 1.9±0.4 | **85.0**±7.1 | 3.4±0.4 |
| BWFlow (CAVG) | Flow | **84.8**±6.44 | **2.4**±0.9 | 81.5±4.9 | **1.3**±0.2 | 84.5±8.0 | **2.3**±0.5 |

| Dataset | Interpolation | Metrics | | | | | | | |
|---|---|---|---|---|---|---|---|---|---|
| | | $\mu$ | V.U.N(%) | Mol.Stab. | Atom.Stab. | Connected(%) | Charge($10^{-2}$) | Atom($10^{-2}$) | Angles(°) |
| QM9 (with h) | MiDi | 1.01 | 93.13 | 93.98 | 99.60 | 99.21 | 0.2 | 3.7 | 2.21 |
| | FlowMol | 1.01 | 87.53 | 88.45 | 99.13 | 99.09 | 0.4 | 4.2 | 2.72 |
| | BWFlow | 1.01 | **96.45** | **97.84** | **99.84** | **99.24** | **0.1** | **2.3** | **1.96** |
| GEOM (with h) | Midi | 1.34 | 78.23 | 32.42 | 89.61 | **79.15** | 0.6 | 11.2 | 9.6 |
| | FlowMol | 1.34 | 82.20 | 36.90 | 94.60 | 59.98 | 0.4 | 8.8 | 6.5 |
| | BWFlow | **1.20** | **87.75** | **46.80** | **95.08** | 73.53 | **0.1** | **6.5** | **3.96** |

Table 2: Quantitative experimental results on 3D Molecule Generation with explicit hydrogen.

**Molecule generation.** Table 2 gives the results on the 3D molecule generation task with explicit hydrogen, where we ignore the bond type but just view the adjacency matrix as a binary one for validating the power of generating graph structures. Interestingly, the empirical results show that even without edge type, the models already can capture the molecule data distribution. And our BWFlow significantly outperforms the SOTA models, including MiDi Vignac et al. (2023b) and FlowMol Dunn & Koes (2024). We believe a promising future direction is to incorporate the processing of multiple bond types into our framework, which would potentially raise the performance by a margin.

## 4.3 BEHAVIOR ANALYSIS

**BWFlow provides smooth velocity in probability paths**. To illustrate how BWFlow models the smooth evolution of graphs, we compute the A.Ratio on SBM datasets (the figures for the others are in Fig. 6) between generated graph interpolants and test data for $t \in [0, 1]$, as shown in Fig. 3a. In contrast to the linear (arithmetic) interpolation, BW interpolation initially exposes the model to more out-of-distribution samples with increased A.Ratio. After this early exploration, the A.Ratio monotonously converges, yielding a smooth interpolation between the reference graphs and the data points. This behavior enhances both

Table 3: Model performance in small sampling steps. DeFoG-1 and DeFoG 2 are without and with path manipulation respectively.

| Model | Planar | | SBM | |
|---|---|---|---|---|
| | V.U.N. ↑ | A.Ratio ↓ | V.U.N. ↑ | A.Ratio ↓ |
| Cometh | 17.0± 4.5 | 7.5± 2.7 | 43.0± 7.5 | 3.3± 0.9 |
| DeFoG-1 | 24.5±6.5 | 6.6±0.9 | 32.5±8.8 | 7.9±0.7 |
| DeFoG-2 | 72.0±7.4 | 6.3±1.9 | 47.5±2.0 | 3.1±0.9 |
| BWFlow | **77.0**±3.7 | **4.1**±1.0 | **52.0**±5.1 | **2.6**±0.9 |

the model robustness and velocity estimation, which helps in covering the convergence gap in the generation stage as in Fig. 1c. In comparison, harmonic and geometric interpolations step outside the valid graph domain, making the learning ill-posed.

**The impact of interpolation methods on the model performance.** Fig. 3b illustrated a bar plot that compares interpolation methods on the ability of generating valid plain graphs measured by V.U.N., which shows the superiority of BW interpolation in capturing graph distributions (full results in Table 7). Fig. 3c illustrated an example (in planar graph generation) of the convergence curve at the training stage, which suggests that BWFlow can bring a faster convergence speed compared to FM methods constructed with linear (arithmetic) interpolations. Additionally, we test when the sampling step size is only 3% of the original one (30 vs 1k), and report the results in Table 3. The results show that BWFlow significantly succeeds in generating high-quality graphs when sampling steps are small.

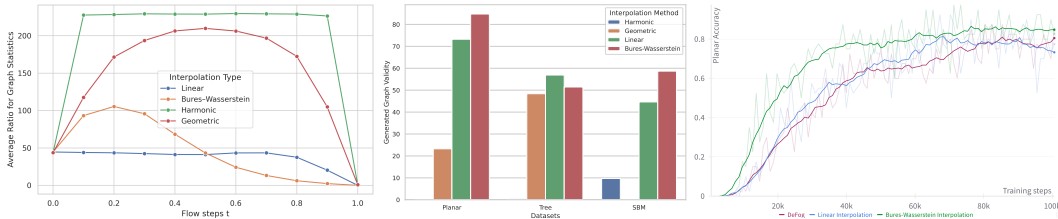

(a) The evolution of graph statistics ratio along the probability path.

(b) The impact of interpolation methods on the performance.

(c) Convergence analysis of BW-Flow and flows with linear interpolations.

Figure 3: Ablation studies for Bures-Wasserstein Flow Matching.

## 5 DISCUSSION AND FUTURE WORK

In this paper, we introduce BWFlow, a flow matching model that intergrates the non-Euclidean and interconnected properties of graphs for graph generation. While we show BWFlow exhibits outstanding performance in various graph generation tasks, there still remains a few solid future work.

*Extension to multiple relation types.* As our framework is built upon the interpolation parameterized by the Graph Laplacian, it is not easily generalizable to the graph generation with multiple edge types. We made preliminary attempts in App. F.5 but a comprehensive design is still required.

*Efficient Probability Path Construction.* Our BW interpolation induces an extra $O(N^3)$ linear algebra operations (noted not reflecting the model complexity) in path construction. When scaled up to large but sparse graphs, the complexity can be reduced to $O(TN^2)$ (with $T$ the iteration steps) through iterative solving such as least-squares with QR factorization. We conduct a preliminary experiment for this in App. F.4 and leave further development as future work.

*The generality of GMRF.* As discussed in App. A.3, GMRF enhances algorithm's ability in modelling graphs with narrow spectral spread while does not improve on graphs with wide spectral spread. Thus, adapting GMRF to graphs with more complex spectral patterns remains a promising direction.

## ACKNOWLEDGEMENTS

K.J. was supported by the UKRI Engineering and Physical Sciences Research Council (EPSRC) [grant number EP/R513143/1]. The authors thank Jinmei Zhang, Yiming Qin, Isabel Haasler, and Pascal Frossard for the valuable inspirations and discussions at the early stage of this project, as well as the anonymous reviewers for their helpful comments and insights.

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

## A GRAPH MARKOV RANDOM FIELDS: BACKGROUND AND THEORY

### A.1 BACKGROUND OF MARKOV RANDOM FIELDS

Markov random fields (MRFs) were originally developed to describe the dynamics of interconnected physical systems such as molecules and proteins (Weigt et al., 2009; Bach et al., 2020). MRFs are energy-based models that have the following probability density:

$$p(\mathcal{X}) = \frac{1}{Z} \prod_c \phi_c(x_c) = \frac{1}{Z} e^{-U(\mathcal{X})/kT}, \tag{15}$$

where the energy $U(\mathcal{X})$ is used to describe the whole connected system. For instance, in the molecule system that consists of atoms and bonds, the overall energy is decomposed into the atom-wise potential $\varphi_1(v), \forall v \in \mathcal{V}$ and bond-wise potential $\varphi_2(u, v), \forall e_{uv} \in \mathcal{E}$. MRFs serve as a natural and elegant way to describe general graph systems.

The energy-based models have an intrinsic relationship with generative models. As an example, Song et al. (2021) derived the relationship between diffusion models and the Langevin dynamics, which is used to describe the evolution of an energy-based model. It is shown that the diffusion models are trying to approximate the score function $\nabla_{\mathcal{X}} \log p(\mathcal{X})$. In the energy-based models, the score function is just the gradient of energy, $\nabla_{\mathcal{X}} \log p(\mathcal{X}) = -\nabla_{\mathcal{X}} U(\mathcal{X})$, and the Langevin dynamics becomes transiting between states with different energies.

The idea of our paper originated from the two facts: MRFs are energy-based model describing connected systems, and the energy-based models have their intrinsic relationship with the diffusion/flow models. Thus, if a model is required to describe the evolution of the whole graph system, we believe it is natural to consider constructing a probability path for two graph distributions with MRFs as the backbone.

### A.2 DERIVATION OF GRAPH MARKOV RANDOM FIELDS

The hierarchical graphical model for GraphMRF is visualized in Fig. 4. With such a modelling, the following decomposition holds:

$$\begin{aligned} p(\mathcal{G} \mid \boldsymbol{G}) &= p(\mathcal{X}, \mathcal{E} \mid \boldsymbol{X}, \boldsymbol{W}) \\ &= p(\mathcal{X} \mid \mathcal{E}, \boldsymbol{X}, \boldsymbol{W}) p(\mathcal{E} \mid \boldsymbol{X}, \boldsymbol{W}) \\ &= p(\mathcal{X} \mid \boldsymbol{X}, \boldsymbol{W}) p(\mathcal{E} \mid \boldsymbol{W}) \end{aligned}$$

where the node features and graph structure are interconnected through latent variables $\boldsymbol{X}$ and $\boldsymbol{W}$. The first step follows the chain rule, and the second steps utilize the properties of the Markov Random Fields, i.e 1) the graph structure serves as a prior and is generated first, and 2) the node features are emitted based on that structure. This makes $\boldsymbol{W}$ alone governs the structural prior, i.e. $p(\mathcal{E} \mid \boldsymbol{W}, \boldsymbol{X}) = p(\mathcal{E} \mid \boldsymbol{W})$. For notation clarity, we distinguish the dependency along same hierarchy in the graphical model by using $p(\cdot \mid \cdot)$ with the difference hierarchies as $p(\cdot; \cdot)$ ('|' v.s. ';').

We then show the derivation of Definition 2, which is restated here:

> **Definition 3** (Graph Markov Random Fields). GraphMRF statistically describes graphs as,
>
> $$p(\mathcal{G}; \boldsymbol{G}) = p(\mathcal{X}, \mathcal{E}; \boldsymbol{X}, \boldsymbol{W}) = p(\mathcal{X}; \boldsymbol{X}, \boldsymbol{W}) \cdot p(\mathcal{E}; \boldsymbol{W}) \text{ where } \mathcal{E} \sim \delta(\boldsymbol{W}) \text{ and}$$
> $$\mathrm{vec}(\mathcal{X}) \sim \mathcal{N}\left(\boldsymbol{X}, \boldsymbol{\Lambda}^{\dagger}\right), \text{ with } \boldsymbol{X} = \mathrm{vec}(\boldsymbol{V}^{\dagger}\boldsymbol{\mu}), \boldsymbol{\Lambda} = (\nu \boldsymbol{I} + \boldsymbol{L}) \otimes \boldsymbol{V}^{\top}\boldsymbol{V}. \tag{16}$$
>
> The $\otimes$ is the Kronecker product, $\mathrm{vec}(\cdot)$ is the vectorization operator and $\boldsymbol{I}$ is the identity matrix.

*Derivation:*

We start from

$$p(\mathcal{X}; \boldsymbol{X}, \boldsymbol{W}) \propto \prod_v \exp\left\{-(\nu + d_v)\|\boldsymbol{V}x_v - \boldsymbol{\mu}_v\|^2\right\} \prod_{u,v} \exp\left\{w_{uv}\left[(\boldsymbol{V}x_u - \boldsymbol{\mu}_u)^{\top}(\boldsymbol{V}x_v - \boldsymbol{\mu}_v)\right]\right\}. \tag{17}$$

We assume that the linear transformation matrix has dimension $\boldsymbol{V} \in \mathbb{R}^{K' \times K}$ given that $x_v \in \mathbb{R}^K$ and define a transformed variable

$$h_v \equiv \boldsymbol{V}x_v - \boldsymbol{\mu}_v \in \mathbb{R}^{K'}, \text{ stacking as } \mathcal{H} \in \mathbb{R}^{|\mathcal{V}| \times K'}. \tag{18}$$

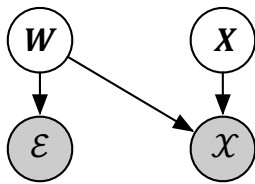

Figure 4: The graphical model for GraphMRF.

The probability becomes

$$P(\mathcal{H}; \boldsymbol{X}, \boldsymbol{W}) \propto \prod_v \exp\left\{-(\nu + d_v)\|h_v\|^2\right\} \prod_{u,v} \exp\left\{w_{uv} h_u^\top h_v\right\}. \tag{19}$$

Then, the terms inside the exponent in Eq. (19) become

$$-\sum_v (\nu + d_v)\|h_v\|^2 + \sum_{u,v} w_{uv} h_u^\top h_v = -\sum_v (\nu + d_v) h_v^\top h_v + \sum_{u,v} w_{uv} h_u^\top h_v$$

$$= -\sum_{u,v} h_u^\top \Big[(\nu + d_u)\delta_{uv} - w_{uv}\Big]h_v,$$

where the Kronecker delta $\delta_{uv} = 1$ if $u = v$ and 0 else. We define a squared matrix $\Lambda'$ to arrange the inner term, which can be written as,

$$\boldsymbol{\Lambda}' = \nu \boldsymbol{I} + \boldsymbol{L} \quad \text{with} \quad \boldsymbol{\Lambda}'_{uv} = (\nu + d_u)\delta_{uv} - w_{uv}. \tag{20}$$

$\boldsymbol{I}$ is the identity matrix. Thus, the exponent in compact matrix form gives

$$-\frac{1}{2}\operatorname{Tr}(\mathcal{H}^\top \boldsymbol{\Lambda}' \mathcal{H}), \text{ where } \mathcal{H} = \begin{pmatrix} h_1 \\ h_2 \\ \vdots \\ h_{|\mathcal{V}|} \end{pmatrix}. \tag{21}$$

It is possible to rearrange the exponent as

$$\operatorname{Tr}(\mathcal{H}^\top \boldsymbol{\Lambda}' \mathcal{H}) = \operatorname{vec}(\mathcal{H})^\top (\boldsymbol{\Lambda} \otimes \boldsymbol{I}) \operatorname{vec}(\mathcal{H}), \tag{22}$$

where $\otimes$ denotes the Kronecker product. This is exactly in the form of a multivariate colored Gaussian. Thus, the joint distribution of $\operatorname{vec}(\mathcal{H})$ (of dimension $|\mathcal{V}|K'$) is given by

$$\operatorname{vec}(\mathcal{H}) \sim \mathcal{N}\Big(0, \ (\nu \boldsymbol{I} + \boldsymbol{L}))^{-1} \otimes \boldsymbol{I}_{K'}\Big), \tag{23}$$

Recall that $h_v = \boldsymbol{V} x_v - \boldsymbol{\mu}_v$, we obtain

$$\operatorname{vec}(\mathcal{H}) = (\boldsymbol{I} \otimes \boldsymbol{V}) \operatorname{vec}(\mathcal{X}) - \operatorname{vec}(\boldsymbol{\mu}). \tag{24}$$

Since the transformation is linear, the distribution over $\mathcal{X}$ remains Gaussian. By the properties of linear transformations of Gaussians, if

$$\operatorname{vec}(\mathcal{H}) \sim \mathcal{N}(\operatorname{vec}(\boldsymbol{\mu}), \boldsymbol{\Sigma}_h), \operatorname{vec}(\mathcal{X}) = (\boldsymbol{I} \otimes \boldsymbol{V}^\dagger) \operatorname{vec}(\mathcal{H}), \tag{25}$$

then

$$\operatorname{vec}(\mathcal{X}) \sim \mathcal{N}\Big((\boldsymbol{I} \otimes \boldsymbol{V}^\dagger) \operatorname{vec}(\boldsymbol{\mu}), \ (\boldsymbol{I}_n \otimes \boldsymbol{V}^\dagger) \boldsymbol{\Sigma}_\mathcal{H} (\boldsymbol{I}_n \otimes \boldsymbol{V}^\dagger)^\top\Big). \tag{26}$$

Thus, using the mixed-product property of the Kronecker product,

$$(\boldsymbol{I} \otimes \boldsymbol{V}^\dagger)((\nu \boldsymbol{I} + \boldsymbol{L})^{-1} \otimes \boldsymbol{I})(\boldsymbol{I}_n \otimes \boldsymbol{V}^\dagger)^\top = (\boldsymbol{L} + \nu \boldsymbol{I})^{-1} \otimes (\boldsymbol{V}^\dagger \boldsymbol{V}^{\dagger\top}) \tag{27}$$

Finally, the joint distribution over $\mathcal{X}$ is

$$\operatorname{vec}(\mathcal{X}) \sim \mathcal{N}(\boldsymbol{X}, \boldsymbol{\Sigma}),$$
$$\text{with } \boldsymbol{X} = (\boldsymbol{I} \otimes \boldsymbol{V}^\dagger) \operatorname{vec}(\boldsymbol{\mu}) = \operatorname{vec}(\boldsymbol{V}^\dagger \boldsymbol{\mu}) \tag{28}$$
$$\text{and } \boldsymbol{\Sigma} = (\nu \boldsymbol{I} + \boldsymbol{L})^{-1} \otimes (\boldsymbol{V}^\dagger \boldsymbol{V}^{\dagger\top}),$$

We use the following lemma:

**Lemma 1.** *Given two invertible matrices $\boldsymbol{A}$ and $\boldsymbol{B}$, their Kronecker product satisfies $(\boldsymbol{A} \otimes \boldsymbol{B})^{-1} = \boldsymbol{A}^{-1} \otimes \boldsymbol{B}^{-1}$.*

So that we get

$$\operatorname{vec}(\mathcal{X}) \sim \mathcal{N}\big(\boldsymbol{X}, \Lambda^\dagger\big), \text{ with } \boldsymbol{X} = \operatorname{vec}(\boldsymbol{V}^\dagger \boldsymbol{\mu}), \Lambda = (\nu \boldsymbol{I} + \boldsymbol{L}) \otimes \boldsymbol{V}^\top \boldsymbol{V}. \tag{29}$$

which ends the derivation.

### A.3 THE APPLICABILITY OF GRAPH MARKOV RANDOM FIELDS

In the experiments, we realize that our BWFlow does not achieve satisfactory results in tree dataset. This evokes us to investigate why this happened, and what properties tree graphs preserve that Graph Markov random fields fail to capture. In what follows, we will provide a comprehensive analysis on the GMRF and show why GMRF achieves a slightly worse performance in trees. We summarize the finding as,

> **Take-home Message:** Modelling Graphs with GMRF enhance algorithm's ability in capturing the global properties encoded in the low-frequency components of the spectra, while retaining a similar capacity in capturing the high frequency components that encodes the fine-grained structures. With such a behavior, graphs with narrow spectral spread are more suitable to be modelled by GraphMRF. In datasets requiring less global modeling ability, BWFlow still preserves comparable capacity to the SOTA models.

We now formally analyze GMRF. Given that our GMRF have an explicit form to constrain the graph distribution, it inherits certain inductive biases and we have to properly understand their generality, i.e., when we could use GraphMRF to model the distribution.

**Plain graph Markov random fields.** To understand the scenarios which we can utilize MRF to model graphs, we first consider the simplest case when $V$ is rectangular orthogonal (semi-orthogonal) matrix such that $V^\top V = I$ and the mean $\mu = 0$, the probability density becomes,

$$P(X, L) \propto \exp(-X^\top(L + \nu I)X) = \exp(-\sum_{\{u,v\}\in\mathcal{E}} W_{uv}(x_u - x_v)^2 - \nu \sum_u x_u^2) \tag{30}$$

As $\nu \to 0$, the exponent term inside becomes

$$\mathcal{S}(X, L) = -\sum_{\{u,v\}\in\mathcal{E}} W_{uv}(x_u - x_v)^2 = \text{trace}(X^\top L X), \tag{31}$$

where we name $\mathcal{S}(X, L)$ as the smoothness of the graph features. The smoothness measures how similar the neighbors connected are. For instance, if there exists an edge between node $u$ and $v$ weighted as $W_{uv}$, the likelihood will be higher if $x_u$ and $x_v$ be similar, so that $\|x_u - x_v\|^2$ are small. This suggests that the probability will be higher if the $\mathcal{S}(X, L)$ is small.

With this pattern, it can be already shown that GMRF can capture the graphs with high smoothness. In the literature of spectral graph theory, the smooth graphs are commonly preserve highly dense low-frequencey components, which are mainly responsible for representing the global properties. Such a pattern can model the homophily, smoothness, planarity, and clustering of graphs, which suggest why BWFlow excels in modeling distributions such as SBM and planar graphs.

**Graph Markov random fields with embeddings.** Now we move one step further to consider the graphs with linear transformer matrix $V$. we can consider alinear transformer from $x_v$ to $h_v$, i.e., giving $h_v = V x_v - \mu$ which gives the probability density as,

$$P(\mathcal{H}, L) \propto \exp(-\text{trace}(\mathcal{H}^\top(L + \nu I)\mathcal{H})) = \exp(-w_{uv}\|h_u - h_v\|^2 - \nu \sum_u \|h_u\|_F^2) \tag{32}$$

where, $\mathcal{H} = [h_1, \ldots h_v, \ldots h_N]^\top$.

Linear transformation provides a map from the feature space to the latent space, which can be considered as an embedding method to empower the models with better expressiveness. As a simple example, when the $V$ provides a negative projection, the mapping can capture the heterophily relationships, which means the nodes connected are dissimilar.

Coincidently, this aligns well with the famous embedding method Node2Vec as in Grover & Leskovec (2016), where the edge weights are proportional to the negative distance, or the inner product of the embeddings. i.e.,

$$W_{uv} \propto \exp(-\|V x_u - V x_v\|_F^2) \tag{33}$$

In Jiang et al. (2025) it is derived that learning the parameters of MRFs is intrinsically equivalent to learning embeddings similar to Node2Vec. As such, the expressiveness of MRFs are as good as Node2Vec, which grants its usage to molecule graphs, protein interaction networks, social networks,

and knowledge graphs. In our paper we make the assumption is that the linear mapping from $\boldsymbol{X}$ the observation is shared. This requirement translates to that the two graphs should have the same embedding space and feature space, which is practical if the reference distribution and data distributions share the same space.

**Graphs without features.** We wish to emphasize that even though the GraphMRF is constructed under the assumption that graph features exist, it is capable of modeling the non-attributed graphs, such as planar and SBM graphs. To do so, we consider the optimization over the Rayleigh function: It is shown that, if $v_1, \ldots, v_{k-1}$ are orthonormal eigenvectors for $\lambda_1, \ldots, \lambda_{k-1}$, then the eigenvalues satisfy,

$$\lambda_k = \min_{\substack{\boldsymbol{x} \neq 0 \\ \boldsymbol{x} \perp \boldsymbol{v}_1, \ldots, \boldsymbol{v}_{k-1}}} R(\boldsymbol{x}), \text{ with } R(\boldsymbol{x}) = \frac{\boldsymbol{x}^T \boldsymbol{L} \boldsymbol{x}}{\boldsymbol{x}^T \boldsymbol{x}} \tag{34}$$

In such a scenario, the graphs are no longer related to the actual node features, but instead, the eigenvectors $\boldsymbol{v}_k$ serve as an intrinsic graph feature. It is noted that $R(\boldsymbol{x})$ is the normalized form of the smoothness as in Eq. (30). This means that if the graphs are smooth (the spectrum of the graph Laplacian focuses on the low-frequency components), the MRF model would give a higher probability compared to the graphs with wider spectral spread. This pattern emphasizes the low-frequency components when building the graph interpolations, which makes the whole algorithm succeed in modelling the graph distributions of most plain graph datasets, such as Planar, SBM, TLS, COMM20 datasets. There is no surprise that our algorithm gives better training and sampling dynamics in those datasets.

It is also worth pointing out that there exists exceptions that GraphMRF is less capable of modelling. Tree graphs are an example of graphs with wider spectral spread. They are acyclic and minimally connected, lacking local clustering and cycles that would result in a highly concentrated spectrum. For this type of graphs, BWFlow will not have a significant benefit in modelling capacity. However, graphs with wider spectral concentration are relatively rare and most are artificial, such as scale-free and expander graphs. In contrast, real-world graphs, such as social networks, citation graphs, traffic networks, and molecular graphs, often contain cycles and densely connected subgraphs. They exhibit strong community structure, local regularity and high redundancy. These characteristics contribute to a highly concentrated Laplacian spectrum, with most eigenvalues clustered together, which aligns well with the GraphMRF prior. Thus, we believe using GMRF and BWFlow to solve graph generation is in general a valuable method.

**Connection to Latent Generative Models.** Although BWFlow is not directly formulated as a latent diffusion/flow model, it shares a strong conceptual connection with this family of generative approaches, especially for graphs (Siraudin & Morris, 2026; Osman et al., 2025). Latent diffusion typically aims to 1) improve computational efficiency and 2) map data into a smoother space where modeling becomes easier. BWFlow does not perform dimensionality reduction, but it is motivated by a similar principle: transforming graphs into a smoother domain can stabilize training and improve sampling efficiency. In BWFlow, this domain is the MRF space, which is manually constructed rather than learned, but exhibits properties desirable in latent diffusion. As discussed previously, transforming a graph into the MRF representation amplifies low-frequency (global) components, paralleling observations in latent diffusion where early latent representations retain a larger proportion of low-frequency information—an effect known to benefit generative modeling Park et al. (2023). This perspective provides an alternative interpretation of why BWFlow yields robust and efficient generation dynamics.

Furthermore, this viewpoint suggests a promising direction for simplifying our current design. In the present implementation, the MRF representation is mapped back to the graph domain before learning the velocity field via a graph transformer. From a latent-diffusion standpoint, an alternative would be to directly parameterize the velocity in the MRF space and train via KL divergence between colored Gaussian distributions, potentially improving efficiency. We leave this direction for future exploration.

## B PROOFS

### B.1 WASSERSTEIN DISTANCE BETWEEN TWO COLORED GAUSSIAN DISTRIBUTIONS

We first prove the lemma that captures the Wasserstein distance between two colored Gaussians, which will be used in deriving our Bures-Wasserstein distances in graph generations.

> **Lemma 2.** *Consider two measures $\eta_0 \sim \mathcal{N}(\boldsymbol{\mu}_0, \Sigma_0)$ and $\eta_1 \sim \mathcal{N}(\boldsymbol{\mu}_1, \Sigma_1)$, describing two colored Gaussian distributions with mean $\boldsymbol{\mu}_0, \boldsymbol{\mu}_1$ and covariance matrices $\Sigma_0, \Sigma_1$. Then the Wasserstein distance between these probability distributions is given by*
>
> $$\left(\mathcal{W}_2(\eta_0, \eta_1)\right)^2 = \|\boldsymbol{\mu}_0 - \boldsymbol{\mu}_1\|^2 + \operatorname{Tr}\left(\Sigma_0 + \Sigma_1 - 2\left(\Sigma_0^{1/2}\Sigma_1\Sigma_0^{1/2}\right)^{1/2}\right).$$

*Proof.* We first state the following proposition.

**Proposition 4.** *(Translation Invariance of the 2-Wasserstein Distance for Gaussian Measures) Consider two measures $\eta_0 \sim \mathcal{N}(\boldsymbol{\mu}_0, \Sigma_0)$ and $\eta_1 \sim \mathcal{N}(\boldsymbol{\mu}_1, \Sigma_1)$ and their centered measure as $\tilde{\eta}_0 = \mathcal{N}(0, \Sigma_0)$ and $\tilde{\eta}_1 = \mathcal{N}(0, \Sigma_1)$, the squared Wasserstein distance decomposes as*

$$\mathcal{W}_2^2(\eta_0, \eta_1) = \|\boldsymbol{\mu}_0 - \boldsymbol{\mu}_1\|_2^2 + \mathcal{W}_2^2(\tilde{\eta}_0, \tilde{\eta}_1)$$

*Proof*:

Consider two random vectors $\mathcal{X}, \mathcal{Y}$ distributed as $\eta_0, \eta_1$,

$$\mathcal{X} = \boldsymbol{\mu}_0 + \tilde{\mathcal{X}}, \mathcal{Y} = \mu_1 + \tilde{\mathcal{Y}}, \text{ with } \tilde{\mathcal{X}} \sim \tilde{\eta}_0, \tilde{\mathcal{Y}} \sim \tilde{\eta}_1.$$

For any coupling $(\mathcal{X}, \mathcal{Y})$, we consider the expected squared Euclidean distance,

$$\begin{aligned}
\mathbb{E}_{\mathcal{X},\mathcal{Y}}\|\mathcal{X} - \mathcal{Y}\|^2 &= \mathbb{E}_{\mathcal{X},\mathcal{Y}}\left\|\boldsymbol{\mu}_0 - \boldsymbol{\mu}_1 + (\tilde{\mathcal{X}} - \tilde{\mathcal{Y}})\right\|^2. \\
&= \|\boldsymbol{\mu}_0 - \boldsymbol{\mu}_1\|^2 + 2\left\langle \boldsymbol{\mu}_0 - \mu_1, \tilde{\mathcal{X}} - \tilde{\mathcal{Y}}\right\rangle + \mathbb{E}_{\tilde{\mathcal{X}},\tilde{\mathcal{Y}}}\|\tilde{\mathcal{X}} - \tilde{\mathcal{Y}}\|^2
\end{aligned} \tag{35}$$

Since $\tilde{\mathcal{X}}$ and $\tilde{\mathcal{Y}}$ both have zero mean, we have $\mathbb{E}[\tilde{\mathcal{X}} - \tilde{\mathcal{Y}}] = 0$ so the cross-term vanishes. Thus,

$$\mathbb{E}\|\mathcal{X} - \mathcal{Y}\|^2 = \|\boldsymbol{\mu}_0 - \boldsymbol{\mu}_1\|^2 + \mathbb{E}\|\tilde{\mathcal{X}} - \tilde{\mathcal{Y}}\|^2 \tag{36}$$

Take the definition of 2-Wasserstein distance, the infimum over all couplings directly yields

$$\begin{aligned}
\left(\mathcal{W}_2(\eta_0, \eta_1)\right)^2 &= \inf_{\pi \in \Pi(\eta_0, \eta_1)} \int \|\mathcal{X} - \mathcal{Y}\|^2 \, d\pi(\mathcal{X}, \mathcal{Y}). \\
&= \|\boldsymbol{\mu}_0 - \boldsymbol{\mu}_1\|^2 + \mathcal{W}_2^2(\tilde{\eta}_0, \tilde{\eta}_1)
\end{aligned} \tag{37}$$

This completes the proof of Proposition 4.

Now we prove the flowing proposition, which will give us our lemma.

**Proposition 5.** *Given two centered measures as $\tilde{\eta}_0 = \mathcal{N}(0, \Sigma_0)$ and $\tilde{\eta}_1 = \mathcal{N}(0, \Sigma_1)$*

$$\mathcal{W}_2^2(\tilde{\eta}_0, \tilde{\eta}_1) = \operatorname{Tr}\left(\Sigma_0 + \Sigma_1 - 2\left(\Sigma_1^{1/2}\Sigma_0\Sigma_1^{1/2}\right)^{1/2}\right). \tag{38}$$

*proof.* The coupling $\pi$ of $\tilde{\eta}_0$ and $\tilde{\eta}_1$ is a joint Gaussian measure with zero mean and covariance matrix

$$\Sigma_c = \begin{pmatrix} \Sigma_0 & C \\ C^T & \Sigma_1 \end{pmatrix} \geq 0, \tag{39}$$

where $C$ is the cross-covariance and $\geq$ means the matrix is positive semi-definitive (PSD). The expected squared distance between the two random vectors $(\mathcal{X}, \mathcal{Y})$ drawn from $\pi$ is then described as,

$$\begin{aligned}
\mathbb{E}\|\mathcal{X} - \mathcal{Y}\|^2 &= \operatorname{Tr}(\mathbb{E}[(\mathcal{X} - \mathcal{Y})(\mathcal{X} - \mathcal{Y})^\top]) \\
&= \operatorname{Tr}(\Sigma_0) + \operatorname{Tr}(\Sigma_1) - 2\operatorname{Tr}(C).
\end{aligned} \tag{40}$$

The definition of Wasserstein distance gives,

$$\mathcal{W}_c(\eta_0, \eta_1) = \inf_{\pi \in \Pi(\eta_0, \eta_1)} \mathbb{E}\|\mathcal{X} - \mathcal{Y}\|^2 \tag{41}$$

Thus, minimizing the Wasserstein distance is equivalent to maximizing $\mathrm{Tr}(\boldsymbol{C})$ over all $\boldsymbol{C}$ subject to the joint covariance is positive semi-definite (PSD). It turns out (see Dowson & Landau (1982); Olkin & Pukelsheim (1982); Takatsu (2010) ) that the condition in Eq. (39) is equivalent to,

$$\Sigma_1 - \boldsymbol{C}^\top \Sigma_0^{-1} \boldsymbol{C} \geq 0 \leftrightarrow \Sigma_0^{-1/2} \boldsymbol{C} \Sigma_1^{-1/2} \text{ has operator norm } \leq 1 \tag{42}$$

So we denote $\boldsymbol{K} \coloneqq \Sigma_0^{-1/2} \boldsymbol{C} \Sigma_1^{-1/2}$ with $\|\boldsymbol{K}\|_{\mathrm{op}} \leq 1$. Then

$$\mathrm{Tr}(\boldsymbol{C}) = \mathrm{Tr}\left(\Sigma_0^{1/2} \boldsymbol{K} \Sigma_1^{1/2}\right) = \mathrm{Tr}\left(\boldsymbol{K} \Sigma_1^{1/2} \Sigma_0^{1/2}\right).$$

Using von Neumann trace inequality, its trace inner-product with $\boldsymbol{K}$ is maximized by choosing $\boldsymbol{K} = \boldsymbol{I}$ on the support.

$$\max_{\|\boldsymbol{K}\|_{op} \leq 1} \mathrm{Tr}(\boldsymbol{K}\boldsymbol{A}) = \mathrm{Tr}\left(\boldsymbol{M}^{1/2}\right), \quad \boldsymbol{M} = \sqrt{\boldsymbol{A}\boldsymbol{A}^\top} = \Sigma_1^{1/2} \Sigma_0 \Sigma_1^{1/2}$$

Hence the optimal value of $\mathrm{Tr}(\boldsymbol{C})$ is

$$\mathrm{Tr}(\boldsymbol{C}^*) = \mathrm{Tr}\left[\left(\Sigma_1^{1/2} \Sigma_0 \Sigma_1^{1/2}\right)^{1/2}\right]$$

Substituting this optimal value into the expression of Wasserstein distance, we obtain

$$\mathcal{W}_2^2\left(\tilde{\eta}_0, \tilde{\eta}_1\right) = \mathrm{Tr}(\Sigma_0) + \mathrm{Tr}(\Sigma_1) - 2 \, \mathrm{Tr}\left[\left(\Sigma_1^{1/2} \Sigma_0 \Sigma_1^{1/2}\right)^{1/2}\right]. \tag{43}$$

This completes the proof of proposition 5. Taking Proposition 4 and Proposition 5 together, we proved Lemma 2.

### B.2 DERIVATION OF THE GRAPH WASSERSTEIN DISTANCE UNDER MRF

We then prove the Bures-Wasserstein distance for two graph distributions. We restate Proposition 1,

---

**Proposition 6** (Bures-Wasserstein Distance). *Consider two same-sized graphs $\mathcal{G}_0 \sim p\left(\mathcal{X}_0, \mathcal{E}_0\right)$ and $\mathcal{G}_1 \sim p\left(\mathcal{X}_1, \mathcal{E}_1\right)$ with $\boldsymbol{V}$ shared for two graphs, described by the distribution in Definition 2. When the graphs are equipped with graph Laplacian matrices $\boldsymbol{L}_0$ and $\boldsymbol{L}_1$ satisfying 1) is Positive Semi-Definite (PSD) and 2) has only one zero eigenvalue. The Bures-Wasserstein distance between these two random graph distributions is given by*

$$d_{BW}(\mathcal{G}_0, \mathcal{G}_1) = \|\boldsymbol{X}_0 - \boldsymbol{X}_1\|_F^2 + \beta \mathrm{Tr}\left(\boldsymbol{L}_0^\dagger + \boldsymbol{L}_1^\dagger - 2\left(\boldsymbol{L}_0^{\dagger/2} \boldsymbol{L}_1^\dagger \boldsymbol{L}_0^{\dagger/2}\right)^{1/2}\right), \tag{44}$$

*as $\nu \to 0$ and $\beta$ is a constant related to the norm of $\boldsymbol{V}$.*

---

Specifically, Definition 2 uses graph Markov random fields to describe a graph as

$$p(\mathcal{G}; \boldsymbol{G}) = p(\mathcal{X}, \mathcal{E}; \boldsymbol{X}, \boldsymbol{W}) = p(\mathcal{X}; \boldsymbol{X}, \boldsymbol{W}) \cdot p(\mathcal{E}; \boldsymbol{W}) \text{ where } \mathcal{E} \sim \delta(\boldsymbol{W}) \text{ and}$$
$$\mathrm{vec}(\mathcal{X}) \sim \mathcal{N}\left(\boldsymbol{X}, \boldsymbol{\Lambda}^\dagger\right), \text{ with } \boldsymbol{X} = \mathrm{vec}(\boldsymbol{V}^\dagger \boldsymbol{\mu}), \boldsymbol{\Lambda} = (\nu \boldsymbol{I} + \boldsymbol{L}) \otimes \boldsymbol{V}^\top \boldsymbol{V}. \tag{45}$$

With the graph Wasserstein distance defined as,

$$(\text{Graph Wasserstein Distance}) \quad d_{\mathrm{BW}}(\mathcal{G}_0, \mathcal{G}_1) \coloneqq \mathcal{W}_c\left(\eta_{\mathcal{G}_0}, \eta_{\mathcal{G}_1}\right) = \mathcal{W}_c(\eta_{\mathcal{X}_0}, \eta_{\mathcal{X}_1}) + \mathcal{W}_c(\eta_{\mathcal{E}_0}, \eta_{\mathcal{E}_1}).$$

We first consider calculating $\mathcal{W}_c(\eta_{\mathcal{X}_0}, \eta_{\mathcal{X}_1})$. Specifically, this is the distance between two colored Gaussian measures where

$$\eta_i \sim \mathcal{N}\left(\boldsymbol{\mu}_i', \Sigma_i\right), \quad i = 0, 1,$$
$$\text{where } \boldsymbol{\mu}_i' = \boldsymbol{V}_i \otimes \boldsymbol{\mu}_i \text{ and } \Sigma_i^{-1} = \Lambda_i = (\nu \boldsymbol{I} + \boldsymbol{L}_i) \otimes (\boldsymbol{V}_i^\top \boldsymbol{V}_i). \tag{46}$$

where we first assume that these two Gaussians are emitted from different linear transformation matrices $\boldsymbol{V}_0$ and $\boldsymbol{V}_1$. This will bring us the most general and flexible form that could be universally applicable, and potentially can bring more insights to future work. Next, we will inject a few assumptions to arrive at a more practical form for building the flow matching models.

An important property of Kronecker product: Given two invertible matrices $\boldsymbol{A}$ and $\boldsymbol{B}$, their Kronecker product satisfies $(\boldsymbol{A} \otimes \boldsymbol{B})^{-1} = \boldsymbol{A}^{-1} \otimes \boldsymbol{B}^{-1}$. Using such a property, in the limit as $\nu \to 0$, we have

$$\Lambda_i \to \boldsymbol{L}_i \otimes (\boldsymbol{V}_i^\top \boldsymbol{V}_i) \quad \implies \quad \Sigma_i = \boldsymbol{L}_i^{-1} \otimes (\boldsymbol{V}_i^\top \boldsymbol{V}_i)^{-1}. \tag{47}$$

According to Lemma 2, the squared 2-Wasserstein distance between two Gaussian measures is given by

$$\mathcal{W}_2^2(\eta_0, \eta_1) = \underbrace{\|\boldsymbol{\mu}_0' - \boldsymbol{\mu}_1'\|^2}_{\text{Mean term}} + \underbrace{\mathrm{Tr}\left(\Sigma_0 + \Sigma_1 - 2\left(\Sigma_0^{1/2}\Sigma_1\,\Sigma_0^{1/2}\right)^{1/2}\right)}_{\text{Covariance Term}}. \tag{48}$$

**Mean Term.** Since $\boldsymbol{\mu}_i' = \boldsymbol{V} \otimes \boldsymbol{\mu}_i$, the mean difference becomes

$$\|\boldsymbol{\mu}_0' - \boldsymbol{\mu}_1'\|^2 = \|\boldsymbol{V}_0\boldsymbol{\mu}_0 - \boldsymbol{V}_1\boldsymbol{\mu}_1\|_F^2 = \|\boldsymbol{X}_0 - \boldsymbol{X}_1\|_F^2 \tag{49}$$

**Covariance term.** Using the property of the Kronecker product, the square root of Eq. (47) factors in as

$$\Sigma_i^{1/2} = \boldsymbol{L}_i^{-1/2} \otimes (\boldsymbol{V}_i^\top \boldsymbol{V}_i)^{-1/2}. \tag{50}$$

and

$$\Sigma_0^{1/2}\Sigma_1\,\Sigma_0^{1/2} = \left(\boldsymbol{L}_0^{-1/2}\boldsymbol{L}_1^{-1}\boldsymbol{L}_0^{-1/2}\right) \otimes \left((\boldsymbol{V}_0^\top \boldsymbol{V}_0)^{-1/2}(\boldsymbol{V}_1^\top \boldsymbol{V}_1)^{-1}(\boldsymbol{V}_0^\top \boldsymbol{V}_0)^{-1/2}\right) \tag{51}$$

We first look into the term related to $\boldsymbol{V}_0$ and $\boldsymbol{V}_1$, which is,

$$\mathrm{Tr}\left((\boldsymbol{V}_0^\top \boldsymbol{V}_0)^{-1/2}(\boldsymbol{V}_1^\top \boldsymbol{V}_1)^{-1}(\boldsymbol{V}_0^\top \boldsymbol{V}_0)^{-1/2}\right) = \mathrm{Tr}\left((\boldsymbol{V}_1^\top \boldsymbol{V}_1)^{-1}(\boldsymbol{V}_0^\top \boldsymbol{V}_0)^{-1/2}(\boldsymbol{V}_0^\top \boldsymbol{V}_0)^{-1/2}\right)$$
$$= \mathrm{Tr}\left((\boldsymbol{V}_1^\top \boldsymbol{V}_1)^{-1}(\boldsymbol{V}_0^\top \boldsymbol{V}_0)^{-1}\right) \tag{52}$$

As $\mathrm{Tr}(\boldsymbol{A} + \boldsymbol{B}) = \mathrm{Tr}(\boldsymbol{A}) + \mathrm{Tr}(\boldsymbol{B})$ the covariance term becomes

Covariance Term

$$= \mathrm{Tr}\left(\Sigma_0 + \Sigma_1 - 2\left(\Sigma_0^{1/2}\Sigma_1\,\Sigma_0^{1/2}\right)^{1/2}\right)$$

$$= \mathrm{Tr}(\Sigma_0) + \mathrm{Tr}(\Sigma_1) - 2\,\mathrm{Tr}\left((\Sigma_0^{1/2}\Sigma_1\,\Sigma_0^{1/2})^{1/2}\right)$$

$$= \mathrm{Tr}\left(\boldsymbol{L}_0^{-1} \otimes (\boldsymbol{V}_0^\top \boldsymbol{V}_0)^{-1} + \boldsymbol{L}_1^{-1} \otimes (\boldsymbol{V}_1^\top \boldsymbol{V}_1)^{-1} - 2\left(\boldsymbol{L}_0^{-1/2}\boldsymbol{L}_1^{-1}\boldsymbol{L}_0^{-1/2}\right)^{1/2} \otimes (\boldsymbol{V}_1^\top \boldsymbol{V}_1)^{-1/2}(\boldsymbol{V}_0^\top \boldsymbol{V}_0)^{-1/2}\right) \tag{53}$$

Given that $\mathrm{Tr}(\boldsymbol{A} \otimes \boldsymbol{B}) = \mathrm{Tr}(\boldsymbol{A})\,\mathrm{Tr}(\boldsymbol{B})$ and $\mathrm{Tr}(\boldsymbol{V}^\top \boldsymbol{V}) = \|\boldsymbol{V}\|_F^2$ for any real-valued matrix $\boldsymbol{V}$, we can further derive,

$$\text{Covariance Term} = \mathrm{Tr}[(\boldsymbol{V}_0^\top \boldsymbol{V}_0)^{-1}]\,\mathrm{Tr}(\boldsymbol{L}_0^\dagger) + \mathrm{Tr}[(\boldsymbol{V}_1^\top \boldsymbol{V}_1)^{-1}]\,\mathrm{Tr}(\boldsymbol{L}_1^\dagger)$$
$$- 2\,\mathrm{Tr}\left(\boldsymbol{L}_0^{\dagger/2}\boldsymbol{L}_1^\dagger\boldsymbol{L}_0^{\dagger/2}\right)^{1/2} \cdot \mathrm{Tr}[(\boldsymbol{V}_1^\top \boldsymbol{V}_1)^{-1/2}(\boldsymbol{V}_0^\top \boldsymbol{V}_0)^{-1/2}]. \tag{54}$$

Unfortunately, to simplify this equation, we have to make the two gram matrix, $(\boldsymbol{V}_0^\top \boldsymbol{V}_0)^{-1}$ and $(\boldsymbol{V}_1^\top \boldsymbol{V}_1)^{-1}$ agree, i.e., $(\boldsymbol{V}_1^\top \boldsymbol{V}_1)^{-1} = (\boldsymbol{V}_0^\top \boldsymbol{V}_0)^{-1}$. This will be satisfied if and only if there exists an orthogonal matrix $\boldsymbol{Q}$ such that

$$\boldsymbol{V}_1^\dagger = \boldsymbol{V}_0^\dagger \boldsymbol{Q}.$$

Thus, to further process, we simply consider the case when $\boldsymbol{V}_1$ and $\boldsymbol{V}_0$ are exactly the same, i.e., $\boldsymbol{V}_1 = \boldsymbol{V}_0 = \boldsymbol{V}$ (we have already discussed how realistic this assumption is in Section A.3). So that we work under the assumptions that $\|\boldsymbol{V}_0^\dagger\|_F^2 = \|\boldsymbol{V}_1^\dagger\|_F^2 = \beta$, which simplify the trace as

$$\text{Covariance Term} = \beta \cdot \mathrm{Tr}\left(\boldsymbol{L}_0^\dagger + \boldsymbol{L}_1^\dagger - 2\left(\boldsymbol{L}_0^{\dagger/2}\boldsymbol{L}_1^\dagger\boldsymbol{L}_0^{\dagger/2}\right)^{1/2}\right). \tag{55}$$

Combining the mean term and the covariance term, we obtain the Wasserstein distance of $\mathcal{W}_c(\eta_{\mathcal{X}_0}, \eta_{\mathcal{X}_1})$

For calculating $\mathcal{W}_c(\eta_{\mathcal{E}_0}, \eta_{\mathcal{E}_1})$, we have the freedom to choose the cost function when obtaining the Wasserstein distance. Note that $\boldsymbol{W}$ serves as the prior for the Gaussian covariance matrix $\Sigma$, where

the covariance has to be positive-semi definite. Thus, according to Bhatia et al. (2019), a proper distance between two positive semi-definite matrices is measured by

$$\mathcal{W}(\eta_{\mathcal{E}_0}, \eta_{\mathcal{E}_1}) = \left\| \Sigma_0^{1/2} - \Sigma_1^{1/2} \right\|_F^2. \tag{56}$$

Coincidentally, this is another usage case when the Bures-Wasserstein metric is utilized. Putting everything together, the Wasserstein distance in the limit $\nu \to 0$ is

$$d_{\mathrm{BW}}(\mathcal{G}_0, \mathcal{G}_1) = \|\boldsymbol{V}_0 \boldsymbol{\mu}_0 - \boldsymbol{V}_1 \boldsymbol{\mu}_1\|_F^2 + (\beta + 1) \cdot \mathrm{Tr}\left( \boldsymbol{L}_0^\dagger + \boldsymbol{L}_1^\dagger - 2 \left( \boldsymbol{L}_0^{\dagger/2} \boldsymbol{L}_1^\dagger \boldsymbol{L}_0^{\dagger/2} \right)^{1/2} \right).$$

$$= \underbrace{\|\boldsymbol{X}_0 - \boldsymbol{X}_1\|_F^2}_{d_{\boldsymbol{X}}(\boldsymbol{X}_0, \boldsymbol{X}_1)} + (\beta + 1) \cdot \underbrace{\mathrm{Tr}\left( \boldsymbol{L}_0^\dagger + \boldsymbol{L}_1^\dagger - 2 \left( \boldsymbol{L}_0^{\dagger/2} \boldsymbol{L}_1^\dagger \boldsymbol{L}_0^{\dagger/2} \right)^{1/2} \right)}_{d_{\boldsymbol{L}}(\boldsymbol{L}_0, \boldsymbol{L}_1)}. \tag{57}$$

This expression separates the contribution of the mean difference (transformed by $\boldsymbol{V}$) and the discrepancy between the covariance structures (encoded in $\boldsymbol{L}_0$ and $\boldsymbol{L}_1$). This could be further used to derive BW interpolation, which we will show in Section C.1. In the main body, constant $\beta$ actually corresponds to $\beta + 1$ here. This complete our derivation in Proposition 1.

## C  DERIVATION OF BURES-WASSERSTEIN FLOW MATCHING

In order to build the flow matching framework, we need to derive the optimal interpolation and the corresponding velocities for the probability path $p(G_t \mid G_0, G_1)$. This is achieved via the OT displacement between two graph distributions.

### C.1  THE BURES-WASSERSTEIN GRAPH INTERPOLATION

We aim to recover the proposition stated as follows.

> **Proposition 7** (Bures-Wasserstein interpolation). *The graph minimizer of Eq. (10), $\mathcal{G}_t = \{\mathcal{V}, \mathcal{E}_t, \mathcal{X}_t\}$, have its node features following a colored Gaussian distribution, $\mathcal{X}_t \sim \mathcal{N}(\boldsymbol{X}_t, \boldsymbol{\Lambda}_t^\dagger)$ with $\boldsymbol{\Lambda}_t = (\nu \boldsymbol{I} + \boldsymbol{L}_t) \otimes \boldsymbol{V}^\top \boldsymbol{V}$ and edges following $\mathcal{E}_t \sim \delta(\boldsymbol{W}_t)$, specifically,*
>
> $$\boldsymbol{L}_t^\dagger = \boldsymbol{L}_0^{1/2} \left( (1-t) \boldsymbol{L}_0^\dagger + t \left( \boldsymbol{L}_0^{\dagger/2} \boldsymbol{L}_1^\dagger \boldsymbol{L}_0^{\dagger/2} \right)^{1/2} \right)^2 \boldsymbol{L}_0^{1/2}, \quad \boldsymbol{X}_t = (1-t) \boldsymbol{X}_0 + t \boldsymbol{X}_1 \tag{58}$$

The interpolation is an extension of the concept of mean, where in the optimal transport world, the Wasserstein barycenter (mean) of measures $\eta_0, \ldots \eta_{m-1}$ under weights $\lambda_0, \ldots \lambda_{m-1}$ can be derived over the following optimization problem:

$$\bar{\eta} = \arg\min_{\eta} \sum_{j=0}^{m-1} \lambda_j \left( \mathcal{W}_2 \left( \eta, \eta_j \right) \right)^2 \tag{59}$$

When $m = 2$, based on the Bures-Wasserstein (BW) distance, we can define the OT displacement minimization problem on graphs described as,

$$\mathcal{G}_t = \arg\min_{\tilde{\mathcal{G}}} (1-t) d_{\mathrm{BW}}(\mathcal{G}_0, \tilde{\mathcal{G}}) + t d_{\mathrm{BW}}(\tilde{\mathcal{G}}, \mathcal{G}_1). \tag{60}$$

where $d_{\mathrm{BW}}(\mathcal{G}_0, \mathcal{G}_1)$ is described in Proposition 1. The optimal graph interpolation is the solution to the problem.

In the setting of graph, this becomes a two-variable optimization problem, where

$$\mathcal{X}_t, \mathcal{E}_t = \arg\min_{\tilde{\mathcal{X}}, \tilde{\mathcal{E}}} (1-t) d_{\mathrm{BW}}(\mathcal{G}_0, \tilde{\mathcal{G}}) + t d_{\mathrm{BW}}(\tilde{\mathcal{G}}, \mathcal{G}_1). \tag{61}$$

Fortunately, recall in Eq. (57) that our distance measurement $d_{\mathrm{BW}}(\mathcal{G}_0, \mathcal{G}_1)$ is decomposed into $d_{\boldsymbol{X}}(\boldsymbol{X}_0, \boldsymbol{X}_1)$ and $d_{\boldsymbol{L}}(\boldsymbol{L}_0, \boldsymbol{L}_1)$, then the optimization over node and edges are disentangleable into

solving the two sub optimization problem,

$$\text{Sub-question 1:} \quad \bar{\boldsymbol{X}}_t = \arg\min_{\tilde{\boldsymbol{X}}} \quad (1-t)\|\boldsymbol{X}_0 - \tilde{\boldsymbol{X}}\|_F^2 + t\|\tilde{\boldsymbol{X}} - \boldsymbol{X}_1\|_F^2$$

$$\text{Sub-question 2:} \quad \bar{\boldsymbol{L}}_t = \arg\min_{\tilde{\boldsymbol{L}}} \quad (1-t)d_{\boldsymbol{L}}(\boldsymbol{L}_0, \tilde{\boldsymbol{L}}) + td_{\boldsymbol{L}}(\boldsymbol{L}_1, \tilde{\boldsymbol{L}}) \tag{62}$$

This two problems are completely disentangled thus we can solve them separately.

**Sub-question 1**  For the first problem, we simply set the derivate to 0 and get,

$$(1-t)(\tilde{\boldsymbol{X}} - \boldsymbol{X}_0) + t(\tilde{\boldsymbol{X}} - \boldsymbol{X}_1) = 0 \rightarrow \boldsymbol{X}_t = (1-t)\boldsymbol{X}_0 + t(\boldsymbol{X}_1) \tag{63}$$

**Subquestion 2**  The second subproblem is equivalent in deriving the covariance of Bures-Wasserstein interpolation between two Gaussian measures, $\eta_0 \sim \mathcal{N}\left(0, \boldsymbol{L}_0^\dagger\right)$ and $\eta_1 \sim \mathcal{N}\left(0, \boldsymbol{L}_1^\dagger\right)$. This problem has been properly addressed in Haasler & Frossard (2024) and here we just verbose their results. For more details we refer the reader to Haasler & Frossard (2024) for a further discussion.

The optimal transport geodesic between $\eta_0 \sim \mathcal{N}\left(0, \boldsymbol{L}_0^\dagger\right)$ and $\eta_1 \sim \mathcal{N}\left(0, \boldsymbol{L}_1^\dagger\right)$ is defined by $\eta_t = ((1-t)I + t\boldsymbol{T})_\#\eta_0$, where the symbol "#" denotes the push-forward of a measure by a mapping, $\boldsymbol{T}$ is a linear map that satisfies $\boldsymbol{T}\boldsymbol{L}_0^\dagger\boldsymbol{T} = \boldsymbol{L}_1^\dagger$.

We define a new matrix $\boldsymbol{M}$ and do normalization, which leads to,

$$\boldsymbol{T} = \boldsymbol{L}_0^{1/2}\boldsymbol{M}\boldsymbol{L}_0^{1/2} \tag{64}$$

Plug in gives,

$$\boldsymbol{T}\boldsymbol{L}_0^\dagger\boldsymbol{T}^\top = \boldsymbol{L}_0^{1/2}\boldsymbol{M}\boldsymbol{L}_0^{1/2}\boldsymbol{L}_0^\dagger\left(\boldsymbol{L}_0^{1/2}\boldsymbol{M}\boldsymbol{L}_0^{1/2}\right)^\top$$

$$= \boldsymbol{L}_0^{1/2}\boldsymbol{M}\boldsymbol{M}^\top\boldsymbol{L}_0^{1/2}. \tag{65}$$

So that we obtain

$$\boldsymbol{L}_1^\dagger = \boldsymbol{L}_0^{1/2}\boldsymbol{M}\boldsymbol{M}^\top\boldsymbol{L}_0^{1/2} \rightarrow \boldsymbol{M} = (\boldsymbol{L}_0^{\dagger/2}\boldsymbol{L}_1^\dagger\boldsymbol{L}_0^{\dagger/2})^{1/2} \tag{66}$$

Replace $\boldsymbol{T}$ and we get,

$$\boldsymbol{T} = \boldsymbol{L}_0^{1/2}\left(\boldsymbol{L}_0^{\dagger/2}\boldsymbol{L}_1^\dagger\boldsymbol{L}_0^{\dagger/2}\right)^{1/2}\boldsymbol{L}_0^{1/2} \tag{67}$$

Given that the geodesic $\eta_t = ((1-t)\boldsymbol{I} + t\boldsymbol{T})_\#\eta_0$ which also has a Gaussian form $\eta_t \sim \mathcal{N}(0, \Sigma_t)$, We can then write the covariance matrix and obtain

$$\boldsymbol{L}_t^\dagger = \Sigma_t = ((1-t)\boldsymbol{I} + t\boldsymbol{T})\boldsymbol{L}_0^\dagger((1-t)\boldsymbol{I} + t\boldsymbol{T})$$

$$= \boldsymbol{L}_0^{1/2}\left((1-t)\boldsymbol{L}_0^\dagger + t\left(\boldsymbol{L}_0^{\dagger/2}\boldsymbol{L}_1^\dagger\boldsymbol{L}_0^{\dagger/2}\right)^{1/2}\right)\boldsymbol{L}_0^{1/2}\boldsymbol{L}_0^\dagger\boldsymbol{L}_0^{1/2}\left((1-t)\boldsymbol{L}_0^\dagger + t\left(\boldsymbol{L}_0^{\dagger/2}\boldsymbol{L}_1^\dagger\boldsymbol{L}_0^{\dagger/2}\right)^{1/2}\right)\boldsymbol{L}_0^{1/2}$$

$$= \boldsymbol{L}_0^{1/2}\left((1-t)\boldsymbol{L}_0^\dagger + t\left(\boldsymbol{L}_0^{\dagger/2}\boldsymbol{L}_1^\dagger\boldsymbol{L}_0^{\dagger/2}\right)^{1/2}\right)^2\boldsymbol{L}_0^{1/2} \tag{68}$$

Which ends the derivation.

> **Remark 1**: Even though the GraphMRF in Definition 2 does rely on an implicit linear emission matrices $\boldsymbol{V}$, the BW interpolation in Proposition 2 can be obtained without explicitly accessing to the $\boldsymbol{V}$ matrices. The property was attractive as in practice we can construct the probability path without explicitly fitting a $\boldsymbol{V}$ beforehand.

## C.2 DERIVING THE VELOCITY OF BW INTERPOLATION

We first show the general form of the velocity term for the Gaussian and Dirac measures.

**Gaussian measure.**  For a time-parametrized Gaussian density $p_t(x) = \mathcal{N}(x; \boldsymbol{\mu}_t, \Sigma_t)$, the velocity field $v_t(x)$ satisfies the continuity equation

$$\partial_t p_t + \nabla \cdot (p_t v_t) = 0,$$

is an affine function of $x$. And the instantaneous velocity field follows,

$$v_t(\mathcal{X}) = \dot{\boldsymbol{\mu}}_t + \frac{1}{2}\dot{\Sigma}_t\Sigma_t^{-1}(\mathcal{X} - \boldsymbol{\mu}_t).$$

**Dirac measure.** When the measure is a Dirac function,

$$p_t(x) = \delta\left(\cdot, \boldsymbol{\mu}_t\right).$$

We can just consider it as the limited case of the Gaussian measure, when $\Sigma_t \to 0$. So that the velocity at simply takes

$$v_t(x) = \dot{\boldsymbol{\mu}}_t.$$

We then move to prove the following proposition for Bures-wasserstein velocity.

> **Proposition 8** (Bures-Wasserstein velocity). *For the graph $\mathcal{G}_t$ following BW interpolation in Proposition 2, the conditional velocity at time $t$ with observation $G_t$ is given as,*
>
> $$v_t(E_t \mid G_0, G_1) = \dot{\boldsymbol{W}}_t = diag(\dot{\boldsymbol{L}}_t) - \dot{\boldsymbol{L}}_t, \quad v_t(X_t \mid G_0, G_1) = \frac{1}{1-t}(\boldsymbol{X}_1 - \boldsymbol{X}_t) \tag{69}$$
>
> $$\text{with } \dot{\boldsymbol{L}}_t = 2\boldsymbol{L}_t - \boldsymbol{T}\boldsymbol{L}_t - \boldsymbol{L}_t\boldsymbol{T} \text{ and } \boldsymbol{T} = \boldsymbol{L}_0^{1/2}(\boldsymbol{L}_0^{\dagger/2}\boldsymbol{L}_1^{\dagger}\boldsymbol{L}_0^{\dagger/2})^{1/2}\boldsymbol{L}_0^{1/2}$$
>
> *where $\boldsymbol{W}_t = \boldsymbol{D}_t - \boldsymbol{L}_t$ and $\boldsymbol{L}_t$ defined in Eq. (11).*

*Proof:*

**The graph structure velocity.** As we assume the edges, $E_t \sim \delta(\cdot, \boldsymbol{W}_t)$, following a dirac distribution, the velocity is defined as

$$v_t(E_t) = \dot{\boldsymbol{W}}_t.$$

Given that, $\dot{\boldsymbol{W}}_t = diag(\dot{\boldsymbol{L}}_t) - \dot{\boldsymbol{L}}_t$, we transit fo deriving the derivative of the Laplacian matrix, $\dot{\boldsymbol{L}}_t$. Using the fact that,

$$\frac{d}{dt}\left(\boldsymbol{A}^{-1}\right) = -\boldsymbol{A}^{-1}\frac{d\boldsymbol{A}}{dt}\boldsymbol{A}^{-1}$$

we obtain the derivate of Laplacian matrix,

$$\dot{\boldsymbol{L}}_t = \frac{d(\Sigma_t^{\dagger})}{dt} = \Sigma_t^{\dagger}\frac{d\Sigma_t}{dt}\Sigma_t^{\dagger} = \boldsymbol{L}_t\frac{d\Sigma_t}{dt}\boldsymbol{L}_t \tag{70}$$

According to Eq. (68) and Eq. (67), the covariance matrix is defined through the interpolation,

$$\Sigma_t = ((1-t)\boldsymbol{I} + t\boldsymbol{T})\boldsymbol{L}_0^{\dagger}((1-t)\boldsymbol{I} + t\boldsymbol{T}) := \boldsymbol{R}_t\boldsymbol{L}_0^{\dagger}\boldsymbol{R}_t \tag{71}$$

where $\boldsymbol{R}_t = (1-t)\boldsymbol{I} + t\boldsymbol{T}$. Taking the derivative, we get,

$$\dot{\Sigma}_t = \frac{d}{dt}\left(\boldsymbol{R}_t\Sigma_0\boldsymbol{R}_t\right) = \boldsymbol{R}_t'\Sigma_0\boldsymbol{R}_t + \boldsymbol{R}_t\Sigma_0\boldsymbol{R}_t' = (\boldsymbol{T}-\boldsymbol{I})\Sigma_0\boldsymbol{R}_t + \boldsymbol{R}_t\Sigma_0(\boldsymbol{T}-\boldsymbol{I}) \tag{72}$$

Using the fact that $\Sigma_0\boldsymbol{R}_t = \boldsymbol{R}_t\Sigma_0 = \Sigma_t$, we obtain the covariance gradient

$$\dot{\Sigma}_t = (\boldsymbol{T}-\boldsymbol{I})\Sigma_t + \Sigma_t(\boldsymbol{T}-\boldsymbol{I}) \tag{73}$$

So that,

$$\begin{aligned} -\dot{\boldsymbol{L}}_t &= \frac{d(\Sigma_t^{\dagger})}{dt} = \Sigma_t^{\dagger}\frac{d\Sigma_t}{dt}\Sigma_t^{\dagger} = \boldsymbol{L}_t\frac{d\Sigma_t}{dt}\boldsymbol{L}_t \\ &= \boldsymbol{L}_t((\boldsymbol{T}-\boldsymbol{I})\boldsymbol{L}_t^{\dagger} + \boldsymbol{L}_t^{\dagger}(\boldsymbol{T}-\boldsymbol{I}))\boldsymbol{L}_t \\ &= \boldsymbol{L}_t(\boldsymbol{T}-\boldsymbol{I}) + (\boldsymbol{T}-\boldsymbol{I})\boldsymbol{L}_t \\ &= \boldsymbol{L}_t\boldsymbol{T} + \boldsymbol{T}\boldsymbol{L}_t - 2\boldsymbol{L}_t \end{aligned} \tag{74}$$

Thus, $\dot{\boldsymbol{L}}_t = 2\boldsymbol{L}_t - \boldsymbol{L}_t\boldsymbol{T} - \boldsymbol{T}\boldsymbol{L}_t$.

Given that $\boldsymbol{L}_t = \boldsymbol{D}_t - \boldsymbol{W}_t$ so that $\boldsymbol{W}_t = diag(\boldsymbol{L}_t) - \boldsymbol{L}_t$, taking the derivative gives $\dot{\boldsymbol{W}}_t = diag(\dot{\boldsymbol{L}}_t) - \dot{\boldsymbol{L}}_t$. As we assume the edges, $E_t \sim \delta(\cdot, \boldsymbol{W}_t)$, the derivate directly yields the velocity,

$$v_t(E_t \mid G_0, G_1) = \dot{\boldsymbol{W}}_t = diag(\dot{\boldsymbol{L}}_t) - \dot{\boldsymbol{L}}_t.$$

**The node feature velocity.** The instantaneous velocity field follows,

$$v_t(\mathcal{X} \mid G_0, G_1) = \dot{\boldsymbol{\mu}}_t + \frac{1}{2}\dot{\Sigma}_t \Sigma_t^{-1}\left(\mathcal{X} - \boldsymbol{\mu}_t\right).$$

The mean gradient interpolating $\eta_0$ and $\eta_1$ can be written as $\dot{\boldsymbol{\mu}}_t = \boldsymbol{X}_1 - \boldsymbol{X}_0$ and $\boldsymbol{X}_t = (1-t)\boldsymbol{X}_0 + t\boldsymbol{X}_1$. So that the velocity leads to,

$$v_t(\mathcal{X} \mid G_0, G_1) = \boldsymbol{X}_1 - \boldsymbol{X}_0 + \frac{1}{2}\dot{\boldsymbol{L}}_t^\dagger \boldsymbol{L}_t\left(\mathcal{X} - \boldsymbol{X}_t\right).$$

However, in practice, we do not need such a complicated velocity term. We wish to avoid the estimation of complex gradient-inverse term so that we can escape from the complicated computation. Under the assumption that the amplitude of covariance is much smaller than the mean difference, we can omit the second term and just keep the mean difference. Hence the instantaneous velocity is simply described as

$$v_t(X_t \mid G_0, G_1) = \boldsymbol{X}_1 - \boldsymbol{X}_0 = X_1 - X_0 = \frac{1}{1-t}(\boldsymbol{X}_1 - \boldsymbol{X}_t) \tag{75}$$

## D  DISCRETE BURES-WASSERSTEIN FLOW MATCHING FOR GRAPH GENERATION

### D.1  PROBABILITY PATH CONSTRUCTION FOR DISCRETE BURES-WASSERSTEIN FLOW MATCHING

**The discrete probability path.** We design the probability path as discrete distributions,

$$p_t(x_v \mid G_0, G_1) = \text{Categorical}([\boldsymbol{X}_t]_v), \quad p_t(e_{uv} \mid G_0, G_1) = \text{Bernoulli}([\boldsymbol{W}_t]_{uv})$$
$$\text{s.t. } p_0(\mathcal{G}) = \delta(G_0, \cdot), p_1(\mathcal{G}) = \delta(G_1, \cdot) \tag{76}$$

where $\boldsymbol{W}_t = \boldsymbol{D}_t - \boldsymbol{L}_t$ with $\boldsymbol{X}_t$ and $\boldsymbol{L}_t$ defined the same in Eq. (11). We consider the fact that the Dirac distribution is a special case when the Categorical/Bernoulli distribution has probability 1 or 0, so the boundary condition $p_0(\mathcal{G}) = \delta(G_0, \cdot), p_1(\mathcal{G}) = \delta(G_1, \cdot)$ holds. As such, $\boldsymbol{X}_t = (1-t)\boldsymbol{X}_0 + t\boldsymbol{X}_1 \in [0,1]^{|\mathcal{V}|\times K}$. Since the boundary condition for each entry, $[\boldsymbol{X}_0]_v$ and $[\boldsymbol{X}_1]_v$ are two one-hot embeddings, $[\boldsymbol{X}_t]_v = t[\boldsymbol{X}_0]_v + (1-t)[\boldsymbol{X}_1]_v$ would sum to one, which works as a valid probability vector. Thus, $\text{Categorical}([\boldsymbol{X}_t]_v)$ is a K-class categorical distribution.

For the edge distribution, we just consider $e_{uv}$ is conditionally independent of the other given $[\boldsymbol{W}_t]_{uv}$. One thing to emphasize is that, given the nature of Bures-Wasserstein interpolation, the yielded $\boldsymbol{W}_t$ is not always bounded by $[0,1]$ thus we have to hard-clip the boundary.

### D.2  APPROXIMATING WASSERSTEIN DISTANCE IN BERNOULLI DISTRIBUTIONS

To make sure that the individual nodes are structured and developed jointly while doing flow matching, we assume that the $\text{vec}(\mathcal{X})$ still maintains a covariance matrix similar to Eq. (8), which gives $\boldsymbol{\Lambda} = (\nu\boldsymbol{I}+\boldsymbol{L})\otimes\boldsymbol{V}^\top\boldsymbol{V}$ given that $\mathcal{X}$ is emitted from a latent variable $\mathcal{H}$ through an affine transformation and the latent variable has a covariance matrix $(\nu\boldsymbol{I} + \boldsymbol{L})^{-1}$. Different from the Gaussian case, the latent variable would still be a discrete distribution, so that the affine transformation carries the covariance matrix out.

Unfortunately, the Wasserstein distance between two discrete graph distributions that follow Eq. (13) does not have a closed-form solution given the complex interwined nature. However, it is possible to use the central limit theorem applied to $\mathcal{X}$ so that we can approximate the Wasserstein distance of two Bernoulli distributions with the Gaussian counterpart. This approximation works when we are in high-dimensional case (high dimension means $|\mathcal{V}|d$ is moderately large.), and the OT-distance between two such Bernoulli distributions is well-captured by the corresponding Gaussian formula, which we already introduced in Eq. (57).

With such nature, even though we are not sampling from Gaussian distributions anymore, it is possible to approximate the Wasserstein distance between two multivariate discrete distributions with the Gaussian counterpart, so the conclusions, such as optimal transport displacements, still hold. And we can similarly derive the Bures-Wasserstein velocity as in the next section.

### D.3 VELOCITY FOR DISCRETE BURES-WASSERSTEIN FLOW MATCHING

**Node velocity.** For node-wise, the path of node features $\mathcal{X}_t$ can be re-written as $p_t(\mathcal{X}) = (1 - t)\delta(\cdot, \boldsymbol{X}_0) + t\delta(\cdot, \boldsymbol{X}_1)$ so the conditional velocity can be accessed through $v_t(X_t \mid G_0, G_1) = [\delta(\cdot, \boldsymbol{X}_1) - \delta(\cdot, \boldsymbol{X}_t)]/(1 - t)$ similar as the derivation in Gat et al. (2024).

**Edge velocity.** For edge-wise, we look into each entry of the adjacency matrix $\boldsymbol{W}$, and consider a time-dependent Bernoulli distribution, the probability density function is:

$$p_t(e_{uv}) = [\boldsymbol{W}_t]_{uv}^{e_{uv}} (1 - [\boldsymbol{W}_t]_{uv})^{1-e_{uv}}, \qquad e_{uv} \in \{0, 1\}. \tag{77}$$

To properly define a velocity $v(x, t)$, it should follow the continuity equation

$$\frac{\partial}{\partial t} p_t(e_{uv}) + \nabla \cdot (p\, v)_t(e_{uv}) = 0. \tag{78}$$

We use $x$ and $y$ to denote two states of $e_{uv}$ ($p(e_{uv} = x) := p(x), p(e_{uv} = y) := p(y)$), then the divergence term is

$$\nabla \cdot (p\, v)(e_{uv} = x) = \sum_{y \neq x} [p_t(y)\, v_t(y \to x) - p_t(x)\, v_t(x \to y)]. \tag{79}$$

As we are working on a Bernoulli distribution, then the forward equations become

$$\begin{cases} \partial_t p(0) = p(1)\, v_t(1 \to 0) - p(0)\, v_t(0 \to 1), \\ \partial_t p(1) = p(0)\, v_t(0 \to 1) - p(1)\, v_t(1 \to 0). \end{cases} \tag{80}$$

Since $p_t(1) = [\boldsymbol{W}_t]_{uv,}$, we have $\partial_t p(1) = [\dot{\boldsymbol{W}}_t]_{uv}$ and $\partial_t p(0) = -[\dot{\boldsymbol{W}}_t]_{uv}$. Hence

$$p(0)\, v_t(0 \to 1) \; - \; p(1)\, v_t(1 \to 0) = [\dot{\boldsymbol{W}}_t]_{uv}.$$

There are many solutions to the above equation. We chose a symmetric solution so that the transition of $e_{uv} \to 1 - e_{uv}$ with

$$v_t(0 \to 1) = \frac{[\dot{\boldsymbol{W}}_t]_{uv}}{1 - [\boldsymbol{W}_t]_{uv}}, \quad v_t(1 \to 0) = -\frac{[\dot{\boldsymbol{W}}_t]_{uv}}{[\boldsymbol{W}_t]_{uv}}.$$

Finally for concise, we can write write it as a velocity field on states $e_{uv} \in \{0, 1\}$, note $1 - 2e_{uv}$ is $+1$ at $e_{uv} = 0$ and $-1$ at $e_{uv} = 1$. Thus, we have

$$v(e_{uv}, t) = (1 - 2e_{uv}) \frac{[\dot{\boldsymbol{W}}_t]_{uv}}{[\boldsymbol{W}_t]_{uv}\, (1 - \boldsymbol{W}_t]_{uv})}, e_{uv} \in \{0, 1\},$$

which in matrix form gives

$$v_t(E_t \mid G_1, G_0) = (1 - 2E_t) \frac{\dot{\boldsymbol{W}}_t}{\boldsymbol{W}_t \circ (1 - \boldsymbol{W}_t)}. \tag{81}$$

Combine the node velocity and the edge velocity, we can now introduce the Discrete Bures-Wasserstein Flow matching algorithm, with the training and inference part respectively introduced in Algorithm 3 and Algorithm 4.

## E DESIGN SPACE FOR BURES WASSERSTEIN INTERPOLATION AND VELOCITY

In the introduction part, we have already compared different probability paths and how they are impacting the inference time sampling. While the Bures-Wasserstein flow path is shown to produce a better probability path for the model to learn, as we illustrated in Fig. 1a, we have to point out that linear interpolation and the corresponding probability path can still converge to the data distribution with sufficiently large flow steps. As if we conduct sampling with infinite flow steps during the later stage of flow, the samples are still able to arrive at the target distributions. A similar pattern exists in diffusion models when they are considered as a Monte-Carlo Markov Chain, and they need sufficiently large steps to converge. We emphasize that the convergence gap in Fig. 1c would be slowly recovered as the number of flow steps increases.

| **Algorithm 3:** Discrete BWFlow Training |
|---|
| **Input:** Ref. dist $p_0$ and dataset $\mathcal{D} \sim p_1$. |
| **Output:** Trained model $f_\theta(G_t, t)$. |
| 1 Initialize model $f_\theta(G_t, t)$; |
| 2 **while** $f_\theta$ *not converged* **do** |
|     /* Sample Boundary Graphs */ |
| 3     Sample batched $\{G_0\} \sim p_0, \{G_1\} \sim \mathcal{D}$; |
|     /* Prob.path Construction */ |
| 4     Sample $t \sim \mathcal{U}(0,1)$; |
| 5     Calculate the BW interpolation to obtain $\boldsymbol{X}_t, \boldsymbol{W}_t$ via Eq. (11); |
| 6     Sample $G_t \sim p(\mathcal{G}_t \mid G_0, G_1)$ according to Eq. (13); |
|     /* Denoising -- x-prediction */ |
| 7     $p_{1\mid t}^\theta(\cdot \mid G_t) \leftarrow f_\theta(G_t, t)$; |
| 8     Loss calculation via Eq. (4); |
| 9     optimizer.step(); |

| **Algorithm 4:** Discrete BWFlow Sampling |
|---|
| **Input:** Reference distribution $p_0$, Trained Model $f_\theta(G_t, t)$, Small time step dt, |
| **Output:** Generated Graphs $\{\hat{G}_1\}$. |
| 1 Initialize samples $\{\hat{G}_0\} \sim p_0$; |
| 2 Initialize the model $p_{1\mid t}^\theta(\cdot \mid G_t) \leftarrow f_\theta(G_t, t)$ |
|   **for** $t \leftarrow 0$ **to** $1 - dt$ **by** $dt$ **do** |
|     /* Denoising – x-prediction */ |
| 3     Predict $\tilde{G}_1 \leftarrow p_{1\mid t}^\theta(\cdot \mid G_t)$; |
|     /* Velocity calculation */ |
| 4     Calculate $v_\theta(\hat{G}_t \mid \hat{G}_0, \tilde{G}_1)$ via Eq. (14); |
|     /* Numerical Sampling */ |
| 5     Sample $\hat{G}_{t+dt} \sim \hat{G}_t + v_\theta(\hat{G}_t)dt$ |

Given that different sampling algorithms can all bring the samples to the data distributions under certain conditions, we wish to understand the huge design space of Bures Wasserstein interpolation. We list the advantages and disadvantages in different techniques and discuss further when each techniques should be used.

In general, we consider two important steps to construct the flow matching for graph generation, specifically, *training* and *sampling*. In training, the main challenge is to obtain a valid real velocity $u(G_t)$ to be regressed to, so we listed a few strategies that can help us with that. In sampling, the challenge becomes how to reconstruct the probability path through the velocity estimated.

### E.1 THE TRAINING DESIGN

In general, the learning objective in flow matching depends on regressing the velocity term. There are several way to obtain the velocity.

1. **Exact velocity estimation.** Use Eq. (3) as the parameterization and learn $p_\theta(\mathcal{G}_1 \mid G_t)$

2. **Numerical approximation.** In the implementation of Stärk et al. (2024), the derivative is calculated through numerical approximation. To achieve better efficiency in calculating velocity, we simply consider a numerical estimation as in Stärk et al. (2024), where the velocity term is obtained as, $\dot{\boldsymbol{L}}_t = (\boldsymbol{L}_{t+\Delta t} - \boldsymbol{L}_t)/\Delta t$. Regressing on the numerical difference can provide an estimation for the velocity.

3. **AutoDiff.** In Chen & Lipman (2024), the derivative of the probability path is evaluated through Pytorch AutoDiff. However, in practice we find this method unstable.

We summarized the training stage model parameterization in Table 4

| | Continuous Flow Matching | Discrete Flow Matching |
|---|---|---|
| $x$-prediction | $v_t^\theta(G_t) = \frac{1}{1-t}\left[\tilde{G}_t^\theta(G_t) - G_t\right]$ | $v_t^\theta(G_t) = \frac{1}{1-t}\left[p_{1\mid t}^\theta(\mathcal{G}_1 \mid G_t) - \delta(\cdot, G_t)\right]$ |
| Numerical Approximation | $v_t^\theta(G_t) \approx G_{t+dt} - G_t$ | $v_t^\theta(G_t) \approx p(G_{t+dt} \mid G_0, G_1) - p(G_t \mid G_0, G_1)$ |
| AutoDiff | $v_t^\theta(G_t) \approx \dot{G}_t$ | Discrete velocity introduced in Eq. (14) |

Table 4: The model parameterization for flow matching in training stage

### E.2 The sampling design

As we described in the Eq. (3), in our training framework, we actually train a denoised $p_\theta(\mathcal{G}_1 \mid \mathcal{G}_t)$. With such a parameterization and taking discrete flow matching as an example, the sampling can be done through one of the following design choices:

1. **Velocity sampling with x-prediction.** The velocity is designed as,

$$v_\theta(G_t) = \frac{1}{1-t}(p_\theta(\mathcal{G}_1 \mid G_t) - \delta(G_t, \cdot)).$$

   This design directly moves the current point $G_t$ towards the direction pointing to the predicted $G_1$. The target-guided velocity is guaranteed to converge to the data distribution, but the interpolant might lie outside the valid graph domain.

2. **BW velocity sampling.** We use Eq. (14) to directly estimate the velocity and flow the Bures-Wasserstein probability path to generate new data points. This path is smooth in the sense of graph domain. However, this path requires more computational cost.

3. **Probability path reconstruction.** The third option is directly reconstructing the probability path, i.e., we first obtain an estimated point,

$$\tilde{G}_1 \sim p_\theta(\mathcal{G}_1 \mid G_t)$$

   and then construct the data point at $t + dt$, which gives

$$G_{t+dt} \sim p(\mathcal{G}_t \mid \tilde{G}_1, G_0)$$

   through Eq. (12). This is the most computationally costly method, which is obtained through the diffusion models. But this method also provide accurate probability path reconstruction.

In Section 4, we show BW velocity follows a path that minimizes the Wasserstein distance thus provides better performance, but sampling following linear velocity also provides convergence with much lower computational cost. So it is a trade-off to be considered in the real-world application.

| | Continuous Flow Matching | Discrete Flow Matching |
|---|---|---|
| BW Velocity | Eq. (12) | Eq. (14) |
| Target-guided Velocity | $v_\theta(G_t) = \frac{1}{1-t}(\tilde{G}_1 - G_t)$ | $v_\theta(G_t) = \frac{1}{1-t}(p_\theta(\mathcal{G}_1 \mid G_t) - \delta(G_t, \cdot))$ |
| Path Reconstruction | $G_{t+dt} = \delta(G_{t+dt} \mid \tilde{G}_1, G_0)$ | $G_{t+dt} \sim p(G_{t+dt} \mid \tilde{G}_1, G_0)$ |

Table 5: Reconstructing probability path choices in flow matching during inference

## F Discussion and limitations

### F.1 The implicit manipulation of probability path

Though not explicitly mentioned, Qin et al. (2024) makes huge efforts to manipulate the probability path for better velocity estimation by extensively searching the design space, and their finding aligns well with the statement that the velocity should be smooth and consistently directing to the data points: 1) **Time distortion:** (The oragne line in Fig. 5b) the polynomial distortion of training and sampling focus on the later stage of the probability trajectory, providing better velocity estimation in this area. This uneven sampling strategy is equivalent to pushing the probability path left to make it smooth. 2) **Target guidance:** (The green line in Fig. 5b) the target guidance directly estimate the direction from a point along the path towards the termination graph, so that the manipulated probability could smoothly pointing to the data distribution. and 3) **Stochasticity injection:** Stochasticity explores the points aside from the path, which avoid the path to be stuck in the platform area.

### F.2 Potential extension to diffusion models

The probability path construction can be denoted as

$$G_t \sim p(\cdot \mid G_0, G_1) = \alpha_t G_1 + \sigma_t G_0 \tag{82}$$

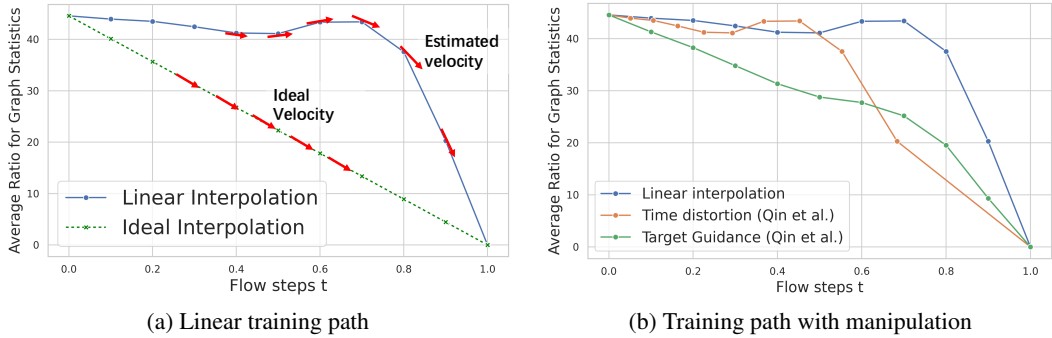

(a) Linear training path  (b) Training path with manipulation

Figure 5: Techniques for manipulating probability path.

At the scenario of flow matching, $\alpha_t = \kappa_t$ and $\sigma_t = 1 - \kappa_t$. The optimal transport flow matching simply use $\alpha_t = t$ and $\sigma_t = 1 - t$

on the other hand, variance-preserving diffusion models constructed the noisy input through a scheduled process, which can be described as,

$$G_t = \sqrt{\bar{\alpha}_t} G_1 + \sqrt{1 - \bar{\alpha}_t} G_0 \tag{83}$$

However, existing diffusion models are still based on the local interpolation, i.e. an element-wise interpolation. This will not compensate the "late convergence" scenario thus we still suffer from the same problem.

Under such an understanding that diffusion is also based on stochastic interpolation, we can easily extend our method to diffusion models by interpolating on the whole graph systems instead of doing so locally. In order to extend the flow matching algorithms with diffusion models, one important thing is to convert the pair-conditioned probability path and velocity to single boundary conditions. For instance, the probability path in flow matching has the form $p(G_t \mid G_1, G_0)$ and the velocity follows $v(G_t \mid G_1, G_0)$. As suggested in Siraudin et al. (2024); Campbell et al. (2022); Xu et al. (2024), the discrete graph diffusion models require a velocity (which is equivalent to a ratio matrix) to perturb the data distribution conditioned on the data points, which we denote as $v(G_t \mid G_1)$. As long as the unilateral conditional velocity has a tractable form, one can first sample a $G_1$ and get $G_t$ through iteratively doing to:

$$G_{t-dt} = G_t - v(G_t \mid G_1)dt$$

starting from $G_1$. So that one can easily construct the probability path $p(G_t \mid G_1)$ to fit into the diffusion model framework. In practice, given that we know the explicit form of $v(G_t \mid G_1, G_{t'})$ (just replace $G_0$ in the expression), the unilateral conditional velocity can be obtained through taking the limitation,

$$v(G_t \mid G_1) = v(G_t \mid G_1, G_t) = \lim_{t' \to t} v(G_t \mid G_1, G_{t'}).$$

Both linear interpolation and our Bures-Wasserstein interpolation can achieve this easily. We just provide a discussion here and will leave this as future work as this paper does not focus on diffusion models but on flow matching models.

## F.3 PERMUTATION INVARIANCE

The Bures-Wasserstein distance between two graph distributions is not permutation invariant, and the minimal value is obtained through the graph alignment. So ideally, to achieve optimal transport, graph alignment and mini-batch matching could provide a better probability path. However, permutation invariance is not always a desired property since we only want to find a path that better transforms from the reference distribution to the data distributions. As an illustration, the widely used linear interpolation to construct graph flow (Qin et al., 2024) does not guarantee permutation invariance as well. And it is proved that, if the measurement is based on Wasserstein distance between two

Gaussian distributions.

$$d_{\mathrm{BW}}(\eta_{\mathcal{G}_0}, \eta_{\mathcal{G}_1}) \le d_{\mathrm{Arithmetic}}(\eta_{\mathcal{G}_0}, \eta_{\mathcal{G}_1})$$

$$\text{with } d_{\mathrm{BW}}(\eta_{\mathcal{G}_0}, \eta_{\mathcal{G}_1}) = \|\boldsymbol{X}_0 - \boldsymbol{X}_1\|_F^2 + \beta \operatorname{trace}\left(\boldsymbol{L}_0^\dagger + \boldsymbol{L}_1^\dagger - 2\left(\boldsymbol{L}_0^{\dagger/2}(\boldsymbol{P}^\top \boldsymbol{L}_1 \boldsymbol{P})^\dagger \boldsymbol{L}_0^{\dagger/2}\right)^{1/2}\right),$$

$$\text{and } d_{\mathrm{Arithmetic}}(\eta_{\mathcal{G}_0}, \eta_{\mathcal{G}_1}) = \|\boldsymbol{X}_0 - \boldsymbol{X}_1\|_F^2 + \beta \|\boldsymbol{L}_0 - \boldsymbol{P}^\top \boldsymbol{L}_1 \boldsymbol{P}\|^2, \forall \boldsymbol{P} \in \text{ potential permutation set}$$

$$(84)$$

### F.4 MITIGATING THE COMPUTATION COMPLEXITY

When constructing the path, our BW interpolation induces an extra $O(N^3)$ linear algebra operations in calculating the matrix inverse of graph Laplacian. We wish to emphasize that this is the basic linear algebra operations in matrix multiplication and does not reflect the model complexity, i.e., the forward/backward propagation and gradient calculation.

Though this computation does not always improve the training clock-time, as we illustrated in Table 9, we feel it valuable to further reduce the computational complexity as the matrix inverse is not properly supported by current GPU design. When the graph size scales up, it also becomes problematic. In order to improve the efficiency, we propose two promising direction:

1. Disentangle the probability path construction from the training and move it to pre-training stage.
2. Approximate the matrix inverse calculation through iterative solving methods.

We provided a preliminary experiment for the second point via solving the inverse by least-squares with QR factorization (LSQR).

**LSQR algorithm,** When given a large, sparse matrix $\boldsymbol{L}$, directly computing $\boldsymbol{L}^{-1}$ yields $O(N^3)$ linear algebra operations. As a remedy, we do not compute the matrix $\boldsymbol{L}^{-1}$ itself, but rather solve it via the linear system:

$$\boldsymbol{L}\boldsymbol{x} = \boldsymbol{b}$$

The solution to this system, $\boldsymbol{x} = \boldsymbol{L}^{-1}\boldsymbol{b}$, provides the desired result without forming the inverse matrix. In order to solve the problem, we formalize the objective as

$$\min_{\boldsymbol{x}} \|\boldsymbol{L}\boldsymbol{x} - \boldsymbol{b}\|_2$$

and conduct least squared minimization for the above problem. The minimization of the above objective can replace the original inverse matrix calculation. THis method is especially useful when $\boldsymbol{L}$ is large, sparse, and potentially ill-conditioned.

LSQR's effectiveness stemas from the factor that the algorithm does not require access to the individual elements of $\boldsymbol{L}$. Instead, it only requires computing the matrix-vector products $\boldsymbol{L}\boldsymbol{x}$. For a sparse matrix with E non-zero elements (exactly the edge number), these products are typically in $O(E)$ time, as opposed to $O(N^2)$ for a dense matrix.

We can use LSQR to find the $j$-th column, $\boldsymbol{c}_j$, of $\boldsymbol{L}^{-1}$, we solve the linear system:

$$\boldsymbol{L}\boldsymbol{c}_j = \boldsymbol{e}_j$$

where $\boldsymbol{e}_j$ is the $j$-th standard basis vector (a vector of all zeros with a 1 in the $j$-th position).

$$\boldsymbol{e}_j = [0, \ldots, 0, 1, 0, \ldots, 0]^T$$

The resulting solution vector will be the $j$-th column of $\boldsymbol{L}^{-1}$. To construct the full inverse, we repeat $N$ times.

When scaled up to large but sparse graphs, the complexity can be reduced to $O(TNE)$ (the previous methods are $O(N^2)$) through iterative solving such as LSQR.

**Newton-Schulz Iteration.** It is also possible to utilize Newton-Schulz Iteration to solve the problem, where the complexity if $O(TN^2)$.

**Preliminary experiment with LSQR** .To show that LSQR does not negatively impact the performance, we conduct preliminary experiments with LSQR instead of the exact pseudo-inverse calculation of the Laplacian matrix. Results clearly show that we could achieve a lower complexity without impacting the quality of the generative model.

Table 6: Model performance using LSQR to approximate the graph Laplacian inverse

| Model | Planar | | Tree | |
|---|---|---|---|---|
| | V.U.N. ↑ | A.Ratio ↓ | V.U.N. ↑ | A.Ratio ↓ |
| BWFlow-LSQR | 85.0± 5.0 | 2.7± 1.4 | 80.1 ± 9.0 | 1.32± 0.3 |
| BWFlow | 84.8± 6.44 | 2.4± 0.9 | 81.5 ± 4.9 | 1.3± 0.2 |

### F.5 EXTENSION TO MULTI-RELATIONAL GRAPHS

A limitation of our method is that it cannot easily capture the generation of graphs with multiple relation types, which we name heterogeneous graphs. Even though we utilize an intuitive solution in the experiment to produce Table 14: we first sample the pure graph structure without edge types to produce the graph backbone, and then sample the edge types via liner interpolated probability on top of the backbone. The solution provides preliminary results for the graph generation in multi-relational graphs, but still requires improvements. Fortunately, there exists a few ways to extend the GraphMRF to heterogeneous graphs (Jiang et al., 2025). An interesting future work can be generalizing our model to heterogeneous graphs by considering GraphMRF variants, such as the H2MN proposed in Jiang et al. (2025).

## G RELATED WORKS

### G.1 DIFFUSION AND FLOW MODELS

Among contemporary generative models, diffusion (Ho et al., 2020) and flow models (Lipman et al., 2023) have emerged as two compelling approaches for their superior performance in generating text and images. In particular, these generative models can be unified under the framework of stochastic interpolation (Albergo & Vanden-Eijnden, 2023), which consists of four procedures (Lipman et al., 2024) as we introduced in Section 1. These contemporary generative models rely on constructing a probability path between data points of an easy-to-sample reference distribution and of the data distribution, and training a machine learning model to simulate the process (Lipman et al., 2024). So that one can sample from the reference (a.k.a source) distribution and iteratively transform it to approximate data samples from the target distribution. Diffusion models construct the probability path with a unilateral path conditioned on the data distribution, where one start sampling from a data point $X_1$ and construct the path $p(\mathcal{X}_t \mid X_1)$. While flow models can condition on either two boundary conditions, $\{X_1, X_0\}$ or just one-side boundary condition $X_1$.

Depending on the space that the algorithm operates on, both models can be categorized into continuous or discrete models. The continuous generative models assume the data distributions are themself lying in continuous space (such as Gaussian) and build models, with examples in diffusion (Ho et al., 2020; Song et al., 2021; Wang et al., 2024) and flow (Lipman et al., 2023; Liu et al., 2023b). The discrete generateive models assume the data follows a discrete distribution, for instance categorical or Bernoulli distributions. Examples include discrete diffusion (Campbell et al., 2022; Sun et al., 2023) and discrete flow models (Campbell et al., 2024; Gat et al., 2024; Minello et al., 2025).

Under the stochastic interpolation framework, the interpolation methods are commonly selected through optimal transport (OT) displacement interpolant (Liu et al., 2023b; Albergo & Vanden-Eijnden, 2023; McCann, 1997). Optimal transport is a classical topic in mathematics that was originally used in economics and operations research (Villani & Society, 2003), and has now become a popular tool in generative models. OT aims for finding the best transport plan between two probability measures with the smallest associated transportation cost. It has been shown that generative models can be combined with technologies such as iterative matching (Tong et al., 2024) and mini batching (Pooladian et al., 2023) to approximate the OT cost, and get a significant boost in their performance in generative modeling.

Another relevant work is Haviv et al. (2024), where the authors explore the flow matching technologies between two Gaussian measures, there they try to interpolate between two Gaussian measures. Our work focuses on a different task of graph generation.

### G.2 GRAPH GENERATION MODELS

Thanks to the capability of graphs in representing complex relationships, graph generation (Zhu et al., 2022; Liu et al., 2023a) has become an essential task in various fields such as protein design (Ingraham et al., 2019), drug discovery (Bilodeau et al., 2022), and social network analysis (Li et al., 2023). The initial attempt at graph generation is formalized through autoregression. For instance, GraphRNN (You et al., 2018) organizes the node interactions into a series of connection events and conducts autoregressive prediction for generation. Later, one shot generation methods such as Variational Graph Auto-Encoder were proposed (Kipf & Welling, 2016; Cao & Kipf, 2018).

Among various generative models, diffusion models and flow-based models have emerged as two compelling approaches for their ability to achieve state-of-the-art performance in graph generation tasks (Niu et al., 2020; Vignac et al., 2023a; Eijkelboom et al., 2024; Qin et al., 2024; Hou et al., 2024). In the early stage, continuous diffusion models were first extended to the task of graph generation (Niu et al., 2020), where they just view the adjacency matrix as a special signal living on the $\mathbb{R}^{|\mathcal{V}| \times |\mathcal{V}|}$ domain. However, these methods fail to capture the natural discreteness of graphs, and Vignac et al. (2023a) first brings discrete diffusion into graph generation. After that, more work (Siraudin et al., 2024; Xu et al., 2024) starts to focus on designing better discrete diffusion models for graph generation.

On the other hands, with the development of flow matching techniques, a few works have been developed to utilize flow models for graph generation and they have achieved huge success. Eijkelboom et al. (2024) utilizes variational flow matching to process categorical data and Qin et al. (2024) developed discrete flow matching for graph generation tasks.

In parallel, there are a number of work that have managed to respect the intrinsic nature of graphs, such as global patterns. For instance, Jo et al. (2024) brings a mixture of graph technique to enhance the performance by explicitly learning final graph structures; Yu & Zhan (2025) mitigates exposure bias and reverse-start bias in graph generation; Hou et al. (2024) improves graph generation through optimal transport flow matching techniques but they still assume the independence between nodes and edges and use hamming distance to measure the transport cost; and Li et al. (2023) gives the large-scale attributed graph generation framework through batching edges.

However, there remain a core challenge: constructing the probability path $p_t$. Existing text and image generative models, operating either in the continuous (Ho et al., 2020; Song et al., 2021; Lipman et al., 2023; Liu et al., 2023b) or discrete (Campbell et al., 2022; Sun et al., 2023; Campbell et al., 2024; Gat et al., 2024; Minello et al., 2025) space, typically rely on linear interpolation between source and target distributions to construct the path. Graph generation models, including diffusion (Niu et al., 2020; Vignac et al., 2023a; Haefeli et al., 2022; Xu et al., 2024; Siraudin et al., 2024) and flow-based models (Eijkelboom et al., 2024; Qin et al., 2024; Hou et al., 2024), inherit this design by modeling every single node and edge independently and linearly build paths in the disjoint space. However, this approach is inefficient because it neglects the strong interactions and relational structure inherent in graphs, i.e., the significance of a node heavily depends on the configuration of its neighbors. While empirical success have been achieved via fine-grained searching on the training and sampling design (Qin et al., 2024) such as target guidance and time distortion, we argue that there remains a fundamental issue of the linear probability path construction, and these strategies only mitigate the problem by manipulating the probability path.

## H COMPARISON WITH OTHER INTERPOLATION METHODS

In the experimental part, we compare our methods with arithmetic (linear) interpolation, geometric interpolation and harmonic interpolation. We state the equation of them respectively as follows.

We consider the boundary graph $G_0$ and $G_1$ with $\boldsymbol{X}_0, \boldsymbol{X}_1 \in \mathbb{R}^{|\mathcal{V}| \times d}$ and $\boldsymbol{W}_0, \boldsymbol{W}_1 \in \mathbb{R}^{|\mathcal{V}| \times |\mathcal{V}|}$. Let $t \in [0, 1]$, we fixed the feature interpolation as,

$$\boldsymbol{X}_t = (1 - t)\boldsymbol{X}_0 + t\boldsymbol{X}_1,$$

the graph structure interpolation can be expressed as,

**Linear interpolation:**
$$\boldsymbol{W}_t = (1 - t)\boldsymbol{W}_0 + t\boldsymbol{W}_1.$$

| Dataset | FM type | Interpolation | V.U.N metrics | | | Spectral Metrics | | | | | |
|---|---|---|---|---|---|---|---|---|---|---|---|
| | | | Novelty | Uniqueness | Validity | Orbit | Spec | Clustering | Degree | Wavelet | Avg. Ratio |
| Planar | Discrete | DeFog | 100 | 100 | 78.25 | 8.98 | 1.45 | 2.09 | 2.65 | 2.38 | 3.51 |
| | | Linear | 100 | 100 | 73.25 | 10.83 | 1.33 | 1.74 | 2.24 | 2.39 | 3.70 |
| | | harmonic | 100 | 100 | 0.00 | 4519.63 | 2.57 | 17.01 | 25.10 | 42.41 | 921.35 |
| | | Geometric | 100 | 100 | 23.25 | 655.66 | 1.61 | 10.17 | 13.25 | 6.68 | 137.47 |
| | | BW | 100 | 100 | 84.75 | 5.14 | 1.27 | 1.69 | 1.78 | 2.02 | 2.38 |
| Tree | Discrete | Defog | 100 | 100 | 61.53 | / | 1.17 | / | 1.27 | 1.51 | 1.32 |
| | | Linear | 100 | 100 | 56.91 | / | 1.16 | / | 1.04 | 1.45 | 1.22 |
| | | harmonic | 100 | 100 | 0.53 | / | 2.32 | / | 1.93 | 3.31 | 2.52 |
| | | Geometric | 100 | 100 | 48.38 | / | 1.62 | / | 2.13 | 2.10 | 1.94 |
| | | BW | 100 | 100 | 51.45 | / | 1.58 | / | 2.56 | 2.13 | 2.09 |
| SBM | Discrete | Linear | 100 | 100 | 44.63 | 2.57 | 1.40 | 1.46 | 15.55 | 7.88 | 5.77 |
| | | harmonic | 100 | 100 | 9.73 | 3.10 | 10.23 | 1.59 | 172.10 | 103.04 | 58.10 |
| | | Geometric | 100 | 100 | 0.0 | 3.11 | 4.45 | 1.80 | 150.41 | 60.60 | 54.65 |
| | | BW | 100 | 100 | 58.70 | 2.03 | 1.50 | 1.50 | 9.04 | 8.41 | 4.51 |

Table 7: Ablation study on interpolation methods when probability path manipulation techniques are all disabled. The clustering and orbit ratios in tree graphs are omitted, given that in the training set, the corresponding statistics are 0. The results go over exponential moving average (decay 0.999) for the last 5 checkpoints. The table is produced with Marginal boundary distributions, without time distortion.

Table 8: Comparison of interpolation methods on 3D Molecule generation with explicit hydrogen in QM9 dataset.

| Flow Type | Interpolation | Metrics | | | | | | | |
|---|---|---|---|---|---|---|---|---|---|
| | | $\mu$ | V.U.N(%) | Mol Stable | Atom Stable | Connected(%) | Charge($10^{-2}$) | Atom($10^{-2}$) | Angles(°) |
| Discrete | MiDi | 1.01 | 93.13 | 93.98 | 99.60 | 99.21 | 0.2 | 3.7 | 2.21 |
| | Linear | 1.01 | 87.53 | 88.45 | 99.13 | 99.09 | 0.4 | 4.2 | 2.72 |
| | harmonic | 1.01 | 94.91 | 94.54 | 99.65 | 99.03 | 0.6 | 6.4 | 2.21 |
| | Geometric | 1.01 | 91.26 | 91.29 | 99.42 | 98.42 | 0.1 | 4.4 | 3.63 |
| | BW | 1.01 | **96.45** | **97.84** | **99.84** | 99.24 | **0.1** | **2.3** | **1.96** |
| Continuous | Linear | 2.15 | 25.45 | 10.23 | 76.85 | 28.82 | 0.7 | **5.6** | 14.47 |
| | harmonic | 1.01 | 11.38 | 11.64 | 73.48 | 99.65 | 1.2 | 17.2 | 15.04 |
| | Geometric | **1.00** | 42.07 | 46.08 | 91.13 | **99.87** | 1.0 | 12.7 | 8.03 |
| | BW | 1.02 | **62.02** | **61.76** | **95.99** | 97.72 | **0.6** | 8.7 | **7.80** |

\* Clearly, continuous flow matching models are not as comparative as discrete flow matching models.

**Geometric interpolation:**

$$W_t = W_0^{1/2} \left( W_0^{-1/2} W_1 W_0^{-1/2} \right)^t W_0^{1/2},$$

**Harmonic interpolation:**

$$W_t = \left( (1-t) W_0^{-1} + t W_1^{-1} \right)^{-1}.$$

Each interpolation methods actually handle each special manifold assumption, which should be designed under a comprehensive understanding of the task. In our experimental part, we conduct intensive analysis on the impact of interpolation methods to the graph generation quality.

### H.1 EMPIRICAL COMPARISON

In Table 7, we illustrate the numerical results for comparing the interpolation methods in plain graph generation without node features. It is clear that BWFlow outperforms other methods in planar and SBM graphs. But the performance was not good in tree graph generations. In Table 8 we compare the interpolation methods in molecule generation.

## I ADDITIONAL EXPERIMENT RESULTS

### I.1 EXPERIMENT SETUPS AND COMPUTATIONAL COST

The training and sampling computation time are provided in Table 9. The experiments were run on a single NVIDIA A100-SXM4-80GB GPU. The hyperparameter configuration in producing Tables 1, 2 and 14 is reported in Table 10.

Table 9: Training and sampling time on each dataset. TG means using target-guided velocity; BW means using BW velocity.

| Dataset | Min Nodes | Max Nodes | Training Time (h) | Graphs Sampled | Sampling Time (h) |
|---------|-----------|-----------|-------------------|----------------|-------------------|
| Planar | 64 | 64 | 45 (1.55x) | 40 | 0.07(TG); 0.13 (BW) |
| Tree | 64 | 64 | 10 (1.25x) | 40 | 0.07(TG);0.14(BW) |
| SBM | 44 | 187 | 74 (0.98x) | 40 | 0.07(TG) 0.14(BW) |
| Moses | 8 | 27 | 35 (0.76x) | 25000 | 5(TG); 6(BW) |
| Guacamol | 2 | 88 | 251 (1.8x ) | 10000 | 7(TG); 21(BW) |
| QM9 | 3 | 29 | 15 | 25000 | 5(TG) 6(BW) |
| GEOM | / | 181 | 141 | 10000 | 7(TG) 14(BW) |

Table 10: Best Configuration for Training and Sampling when producing Tables 1, 2 and 14.

| | Training | | Sampling | | |
|---------|----------------------|------------------|-------------------|----------------|---------------|
| Dataset | Initial Distribution | Train Distortion | Sample Distortion | Sampling steps | Stochasticity |
| Planar | Marginal | Identity | Identity | 1000 | 50 |
| Tree | Marginal | Polydec | Polydec | 1000 | 0 |
| SBM | Absorbing | Identity | Identity | 1000 | 0 |
| MOSES | Marginal | Identity | Identity | 500 | 200 |
| GUACAMOL | Marginal | Identity | Identity | 500 | 300 |
| QM9 | Marginal | Identity | Identity | 500 | 0 |
| GEOM | Marginal | Identity | Identity | 500 | 0 |

## I.2 BEST CHECKPOINT RESULTS IN PLAIN GRAPH GENERATION

We present the best checkpoint results in plain graph generation in Table 11.

## I.3 FULL EXPERIMENT RESULTS FOR PLAIN GRAPH GENERATION

We provide the full experiment results in Table 12, with exisitng graph generation models' performance reported. Most of the results are taken from Qin et al. (2024). We reported the detailed ratios for each model that we reproduced in Table 13.

## I.4 ADDITIONAL RESULTS FOR 2D MOLECULE GENERATION

We wish to note that the 2D molecule generation task is relatively simple and are near saturated with most state-of-the-art models.

**Setup.** In 2D molecular generation, two scenarios with and without bond types information are considered to better evaluate the ability of generating graph structures.

**Metrics:** In addition to the novelty and uniqueness of moecules, we also utilize the relaxed stability of atoms (*Atom.Stab.*), the relaxed molecule stability (*Mol.Stab.*) and the *validity* of a molecule, are used for 2D molecule generation. In addition to these metrics, distribution metrics are also used for 2D molecules, which includes 1) *Fréchet ChemNet Distance (FCD):* whichMeasures the distance between the statistical distributions of generated and real molecules, and *Similarity to Nearest Neighbor (SNN)*, which measures the average similarity (e.g., Tanimoto) of each generated molecule to its closest neighbor in a reference dataset. Also, practicality & diversity metrics, including *Filters* Percentage of molecules passing medicinal chemistry filters (e.g., drug-likeness, synthetic accessibility), and*Scaffold Diversity (Scaf)* that measures the diversity of the core molecular frameworks (scaffolds), indicating structural variety.

**Model peformance.** The model performance is illustrated in Table 14. In both datasets, BWFlow can achieve competitive results near the state-of-the-art (SOTA) flow matching models (Qin et al., 2024), and outperforms the diffusion models. Given that MOSES and GUACAMOL benchmarks are approaching saturation, the fact that BWFlow achieves results on par with the SOTA models serves as strong evidence of its effectiveness.

Table 11: The best-checkpoint plain graph generation Performance. Results are obtained through tuning the probability path manipulation techniques. The remaining values are obtained from Qin et al. (2024).

| Model | Planar Dataset | | | | |
| | Ratio ↓ | Valid ↑ | Unique ↑ | Novel ↑ | V.U.N. ↑ |
|---|---|---|---|---|---|
| Train set | 1.0 | 100 | 100 | 0.0 | 0.0 |
| GRAN (Liao et al., 2019) | 2.0 | 97.5 | 85.0 | 2.5 | 0.0 |
| SPECTRE (Martinkus et al., 2022) | 3.0 | 25.0 | 100 | 100 | 25.0 |
| DiGress (Vignac et al., 2023a) | 5.1 | 77.5 | 100 | 100 | 77.5 |
| EDGE (Chen et al., 2023) | 431.4 | 0.0 | 100 | 100 | 0.0 |
| BwR Diamant et al. (2023) | 251.9 | 0.0 | 100 | 100 | 0.0 |
| BiGG (Dai et al., 2020) | 16.0 | 62.5 | 85.0 | 42.5 | 5.0 |
| GraphGen Goyal et al. (2020) | 210.3 | 7.5 | 100 | 100 | 7.5 |
| HSpectre (one-shot) (Bergmeister et al., 2024) | 1.7 | 67.5 | 100 | 100 | 67.5 |
| HSpectre Bergmeister et al. (2024) | 2.1 | 95.0 | 100 | 100 | 95.0 |
| GruM (Jo et al., 2024) | 1.8 | — | — | — | 90.0 |
| CatFlow (Eijkelboom et al., 2024) | — | — | — | — | 80.0 |
| DisCo (Xu et al., 2024) | — | 83.6 ±2.1 | 100.0 ±0.0 | 100.0 ±0.0 | 83.6 ±2.1 |
| Cometh - PC (Siraudin et al., 2024) | — | 99.5 ±0.9 | 100.0 ±0.0 | 100.0 ±0.0 | **99.5** ±0.9 |
| DeFoG | 1.6 ±0.4 | 99.5 ±1.0 | 100.0 ±0.0 | 100.0 ±0.0 | **99.5** ±1.0 |
| BWFlow | **1.3** ±0.4 | 97.5 ±2.5 | 100.0 ±0.0 | 100.0 ±0.0 | 97.5±2.5 |

| | Tree Dataset | | | | |
|---|---|---|---|---|---|
| Train set | 1.0 | 100 | 100 | 0.0 | 0.0 |
| GRAN (Liao et al., 2019) | 607.0 | 0.0 | 100 | 100 | 0.0 |
| DiGress (Vignac et al., 2023a) | **1.6** | 90.0 | 100 | 100 | 90.0 |
| EDGE (Chen et al., 2023) | 850.7 | 0.0 | 7.5 | 100 | 0.0 |
| BwR (Diamant et al., 2023) | 11.4 | 0.0 | 100 | 100 | 0.0 |
| BiGG (Dai et al., 2020) | 5.2 | 100 | 87.5 | 50.0 | 75.0 |
| GraphGen (Goyal et al., 2020) | 33.2 | 95.0 | 100 | 100 | 95.0 |
| HSpectre (one-shot) (Bergmeister et al., 2024) | 2.1 | 82.5 | 100 | 100 | 82.5 |
| HSpectre (Bergmeister et al., 2024) | 4.0 | 100 | 100 | 100 | **100** |
| Cometh (Siraudin et al., 2024) | — | 75.0 ±3.7 | 100.0 ±0.0 | 100.0 ±0.0 | 75.0 ±3.7 |
| DeFoG | 1.6 ±0.4 | 96.5 ±2.6 | 100.0 ±0.0 | 100.0 ±0.0 | 96.5 ±2.6 |
| BWFlow | **1.4**±0.3 | 95.5 ±2.4 | 100.0±0 | 100.0±0 | 95.5±2.4 |

| Model | Stochastic Block Model ($n_{max} = 187$, $n_{avg} = 104$) | | | | |
| | Ratio ↓ | Valid ↑ | Unique ↑ | Novel ↑ | V.U.N. ↑ |
|---|---|---|---|---|---|
| Training set | 1.0 | 85.9 | 100 | 0.0 | 0.0 |
| GraphRNN (You et al., 2018) | 14.7 | 5.0 | 100 | 100 | 5.0 |
| GRAN (Liao et al., 2019) | 9.7 | 25.0 | 100 | 100 | 25.0 |
| SPECTRE (Martinkus et al., 2022) | 2.2 | 52.5 | 100 | 100 | 52.5 |
| DiGress (Vignac et al., 2023a) | 1.7 | 60.0 | 100 | 100 | 60.0 |
| EDGE (Chen et al., 2023) | 51.4 | 0.0 | 100 | 100 | 0.0 |
| BwR (Diamant et al., 2023) | 38.6 | 7.5 | 100 | 100 | 7.5 |
| BiGG (Dai et al., 2020) | 11.9 | 10.0 | 100 | 100 | 10.0 |
| GraphGen (Goyal et al., 2020) | 48.8 | 5.0 | 100 | 100 | 5.0 |
| HSpectre (one-shot) (Bergmeister et al., 2024) | 10.5 | 75.0 | 100 | 100 | 75.0 |
| HSpectre (Bergmeister et al., 2024) | 10.2 | 45.0 | 100 | 100 | 45.0 |
| GruM (Jo et al., 2024) | **1.1** | — | — | — | 85.0 |
| CatFlow (Eijkelboom et al., 2024) | — | — | — | — | 85.0 |
| DisCo (Xu et al., 2024) | — | 66.2 ±1.4 | 100.0 ±0.0 | 100.0 ±0.0 | 66.2 ±1.4 |
| Cometh (Siraudin et al., 2024) | — | 75.0 ±3.7 | 100.0 ±0.0 | 100.0 ±0.0 | 75.0 ±3.7 |
| DeFoG | 4.9 ±1.3 | 90.0 ±5.1 | 100.0 ±0.0 | 100.0 ±0.0 | 90.0 ±5.1 |
| BWFlow | 3.8±0.9 | **92.5** ±4.0 | 100.0 ±0.0 | 100.0 ±0.0 | **92.5** ±4.0 |

## I.5 ADDITIONAL RESULTS FOR THE TRAINING PATHS

We elaborate on the detailed experimental setting for training path comparison. Fig. 1a and 3a are generated using a representative plain graph dataset, SBM. At each time step $t$, we compute the average maximum mean discrepancy ratio (A.Ratio) between the interpolants and the real data graphs over multiple graph statistics, including orbit, clustering, spectral, wavelet, and degree ratios. The "ideal velocity" (green curve) in Fig. 1a is a synthetic reference path used purely for conceptual illustration.

Fig. 6 gives the training probability path construction for planar graphs and tree graphs. While planar graphs have a similar pattern as the SBM datasets as in Fig. 3a, the probability path constructed for

Table 12: Plain Graph generation performance. Given that the synthetic datasets are usually unstable in evaluation, we applied an exponential moving average to stabilize the results and sampled 5 times (each run generates 40 graphs) to calculate the mean and standard deviation. The experiment settings are in Table 10. The full version with explicit ratio numbers can be found in Table 11

| Model | Class | Planar | | Tree | | SBM | |
|---|---|---|---|---|---|---|---|
| | | V.U.N. ↑ | A.Ratio ↓ | V.U.N. ↑ | A.Ratio ↓ | V.U.N. ↑ | A.Ratio ↓ |
| Train set | — | 100 | 1.0 | 100 | 1.0 | 85.9 | 1.0 |
| DiGress (EMA) (Vignac et al., 2023a) | Diffusion | $61.5_{\pm10.1}$ | $9.9_{\pm3.3}$ | $56.0_{\pm11.0}$ | $8.9_{\pm3.2}$ | $56.0_{\pm8.5}$ | $3.5_{\pm0.5}$ |
| DisCo (CAVG) (Xu et al., 2024) | Diffusion | $57.5_{\pm2.5}$ | $9.0_{\pm1.41}$ | / | / | $55.0_{\pm5.9}$ | $11.6_{\pm2.9}$ |
| HSpectre (Bergmeister et al., 2024) | Diffusion | 67.5 | 3.0 | 82.5 | 2.1 | 75.0 | 10.5 |
| GruM (EMA) (Jo et al., 2024) | Diffusion | $74.4_{\pm5.15}$ | $3.2_{\pm0.4}$ | $52.5_{\pm3.2}$ | $2.4_{\pm0.7}$ | $73.5_{\pm6.7}$ | $2.6_{\pm0.6}$ |
| Cometh (EMA) (Siraudin et al., 2024) | Diffusion | $80.5_{\pm5.79}$ | $3.0_{\pm0.6}$ | $\mathbf{84.5}_{\pm7.8}$ | $2.0_{\pm0.4}$ | $77.5_{\pm5.7}$ | $4.7_{\pm0.6}$ |
| DeFoG (EMA) (Qin et al., 2024) | Flow | $77.5_{\pm8.37}$ | $3.5_{\pm1.7}$ | $83.5_{\pm10.8}$ | $1.9_{\pm0.4}$ | $\mathbf{85.0}_{\pm7.1}$ | $3.4_{\pm0.4}$ |
| BWFlow (EMA) | Flow | $\mathbf{84.8}_{\pm6.44}$ | $\mathbf{2.4}_{\pm0.9}$ | $81.5_{\pm4.9}$ | $\mathbf{1.3}_{\pm0.2}$ | $84.5_{\pm8.0}$ | $\mathbf{2.3}_{\pm0.5}$ |

| Dataset | Model | Validity Metrics | Spectral Metrics | | | | | Avg. Ratio |
|---|---|---|---|---|---|---|---|---|
| | | V.U.N | Orbit | Spec | Clustering | Degree | Wavelet | |
| Planar | Training set | 100 | 0.0005 | 0.0038 | 0.0310 | 0.0002 | 0.0012 | 1 |
| | Digress | 61.50 | 0.0126 (25.20) | 0.0100 (2.63) | 0.1204 (3.88) | 0.0031 (15.50) | 0.0031 (2.55) | 9.95 |
| | Hspectra (One-shot) | 67.5 | / | / | / | / | / | 3.0 |
| | GruM | 74.39 | 0.00445 (8.90) | 0.00755 (1.99) | 0.00428 (0.14) | 0.00075 (3.75) | 0.00258 (2.15) | 3.20 |
| | Cometh | 80.50 | 0.0038 (7.60) | 0.0080 (2.09) | 0.0334 (1.08) | 0.000224 (1.12) | 0.0036 (2.97) | 3.00 |
| | DeFog | 77.50 | 0.0045 (8.98) | 0.0055 (1.45) | 0.0648 (2.09) | 0.0005 (2.65) | 0.0029 (2.38) | 3.51 |
| | BWFlow | 84.75 | 0.0026 (5.14) | 0.0048 (1.27) | 0.0524 (1.69) | 0.0004 (1.78) | 0.0024 (2.02) | 2.38 |
| Tree | Training set | 100 | 0.0000 | 0.0075 | 0.00000 | 0.0001 | 0.0030 | 1 |
| | Digress | 56.00 | 0.0002 (/) | 0.0126 (1.68) | 0.0025 (/) | 0.0018 (17.80) | 0.0088 (7.30) | 8.90 |
| | Hspectra (One-shot) | 82.50 | / | / | / | / | / | 2.10 |
| | GruM | 52.50 | 0.0001(0.00) | 0.0045 (1.18) | 0.0000(/) | 0.0004 (2.15) | 0.0047(3.91) | 2.41 |
| | Cometh | 86.50 | 0.0000 (/) | 0.0102(1.36) | 0.0000 (/) | 0.0003 (3.20) | 0.0044(1.46) | 2.00 |
| | DeFog | 83.50 | 0.0001 (/) | 0.0126 (1.68) | 0.0000 (/) | 0.0002 (1.87) | 0.0066 (2.21) | 1.92 |
| | BWFlow | 81.50 | 0.0000 (0.00) | 0.0094 (1.17) | 0.0000 (0.00) | 0.0001 (1.27) | 0.0046(1.51) | 1.31 |
| SBM | Training set | 85.90 | 0.0255 | 0.0027 | 0.0332 | 0.0008 | 0.0007 | 1 |
| | Digress | 56.00 | 0.0748 (2.93) | 0.0061 (2.26) | 0.0584 (1.76) | 0.0018 (2.25) | 0.0048 (6.86) | 3.51 |
| | Hspectra (One-shot) | 75.00 | / | / | / | / | / | 10.50 |
| | GruM | 73.50 | 0.0412 (1.62) | 0.0068 (2.52) | 0.0495 (1.49) | 0.0028 (3.50) | 0.0017 (2.43) | 2.60 |
| | Cometh | 77.50 | 0.076 (2.98) | 0.0114 (4.22) | 0.052 (1.56) | 0.0063 (7.88) | 0.0048 (6.86) | 4.70 |
| | DeFog | 85.00 | 0.0426 (1.67) | 0.0045 (1.65) | 0.0501 (1.51) | 0.0062 (7.71) | 0.0030 (4.33) | 3.39 |
| | BWFlow | 84.50 | 0.0515 (2.02) | 0.0030 (1.10) | 0.0478 (1.44) | 0.0028 (3.50) | 0.0025 (3.52) | 2.32 |

Table 13: We reported the detailed ratios in three plain graph generation datasets. We omit the orbit and clustering ratio calculation in tree datasets as the training set values are close to 0 which makes the calculation unreliable.

tree graphs does not follow a similar pattern. We attribute this to the different geometry of tree graphs that reside in hyperbolic space (Yang et al., 2022), and different statistics evolution pattern as we discussed in Section A.3.

## I.6 MORE EXPERIMENTS ON PLAIN GRAPH GENERATIONS

**Additional results for sampling paths.** We then give the sampling path construction in Fig. 7. To better illustrate the advantage of BWFlow, we fix the sampling steps to be as small as 50. It is clear that in planar and SBM dataset, the BW velocity can still provide a smooth probability and stable convergence towards the data distribution. While the linear velocity does not give a good probability path and fails to converge to the optimal value, especially when the sampling size is small.

The maximum mean discrepancy (MMD) of four graph statistics between the set of generated graphs and the test set is measured, including degree (Deg.), clustering coefficient (Clus.), count of orbits with 4 nodes (Orbit), the eigenvalues of the graph Laplacian (Spec.), wavelet ratio (Wavelet.). To verify that the model learns to generate graphs with valid topology, we gives the percentage of valid, unique, and novel (V.U.N.) graphs for where a valid graph satisfies the corresponding property of each dataset (Planar, Tree, SBM, etc.).

**Full results for plain graph generation.** Table 11 gives the full results with other generative models aside from the diffusion and flow models. Table 16 gives the results on smaller datasets, i.e., comm20

Table 14: Large molecule generation results. Only diffusion and flow models are reported. Table 15 gives further experiments with binary edge types.

| Model | Guacamol | | | | MOSES | | | | | | |
| | Val. ↑ | V.U. ↑ | V.U.N.↑ | FCD ↑ | Val. ↑ | Unique. ↑ | Novelty ↑ | Filters ↑ | FCD ↓ | SNN ↑ | Scaf ↑ |
|---|---|---|---|---|---|---|---|---|---|---|---|
| Training set | 100.0 | 100.0 | 0.0 | 92.8 | 100.0 | 100.0 | 0.0 | 100.0 | 0.01 | 0.64 | 99.1 |
| DiGress (Vignac et al., 2023a) | 85.2 | 85.2 | 85.1 | 68.0 | 85.7 | **100.0** | 95.0 | 97.1 | **1.19** | 0.52 | 14.8 |
| DisCo (Xu et al., 2024) | 86.6 | 86.6 | 86.5 | 59.7 | 88.3 | **100.0** | 97.7 | 95.6 | 1.44 | 0.50 | 15.1 |
| Cometh (Siraudin et al., 2024) | 98.9 | 98.9 | 97.6 | 72.7 | 90.5 | 99.9 | 92.6 | **99.1** | 1.27 | 0.54 | **16.0** |
| DeFoG Qin et al. (2024) | **99.0** | **99.0** | **97.9** | **73.8** | 92.8 | 99.9 | 92.1 | 98.9 | 1.95 | 0.55 | 14.4 |
| BWFlow (Ours) | 98.8 | 98.9 | 97.4 | 69.2 | 92.0 | **100.0** | 94.5 | 98.4 | 1.32 | **0.56** | 15.3 |

Table 15: Large molecule generation results. Only comparing the representative diffusion and flow models. B.E. is the scenario that only considers binary edge types. The results are almost saturated, thus not very informative.

| Model | Guacamol | | | MOSES | | |
| | Val. ↑ | V.U. ↑ | V.U.N.↑ | Val. ↑ | Unique. ↑ | Novelty ↑ |
|---|---|---|---|---|---|---|
| Digress (B.E.) | 96.0 | 98.9 | 97.4 | 96.1 | 100 | 100 |
| Defog (B.E.) | 98.4 | 98.4 | 97.9 | 99.3 | 100 | 100 |
| BWFlow (B.E.) | 98.0 | 98.0 | 97.7 | 99.6 | 100 | 100 |

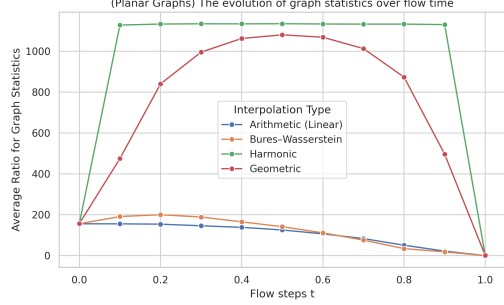

(a) Training-time Probability Path for Planar Graphs

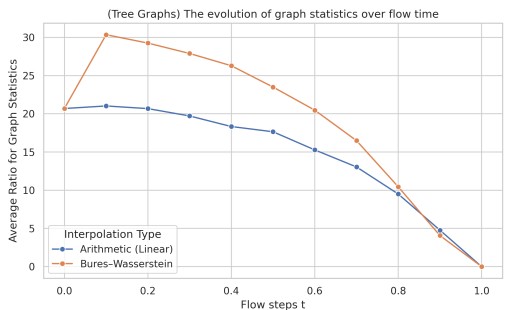

(b) Training-time Tree Graph Probability Path

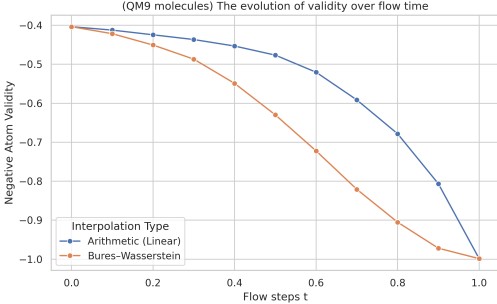

(c) Training-time QM9 Molecule Probability Path. We use negative validity for clear comparison.

Figure 6: Training-time probability path comparisons for planar, tree and QM9.

## I.7 3D MOLECULE GENERATION: QM9 WITHOUT EXPLICIT HYDROGEN

In Table 17 we report the results of QM9 without explicit hydrogen. This task is relatively easy compared to the generation task with explicit hydrogen, and both Midi and our BWFlow have achieved near-saturated performance with validity near to 100%.

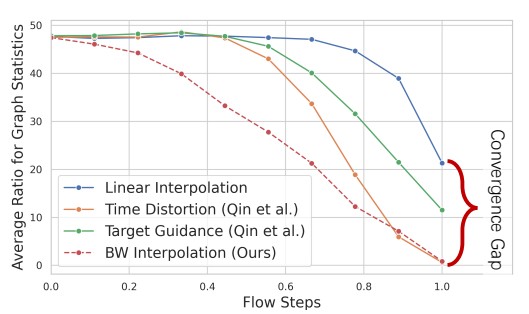 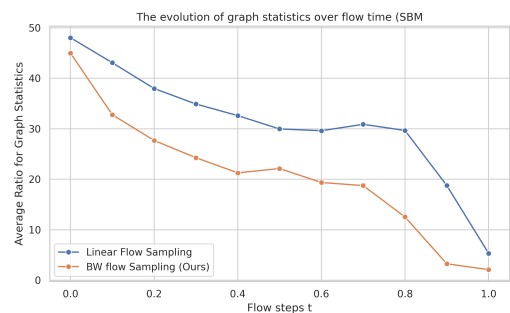

(a) Sampling path comparision on Planar dataset     (b) Sampling path comparision on SBM dataset

Figure 7: The probability path reconstruction in the sampling stage on a) Planar graphs and b) SBM graphs.

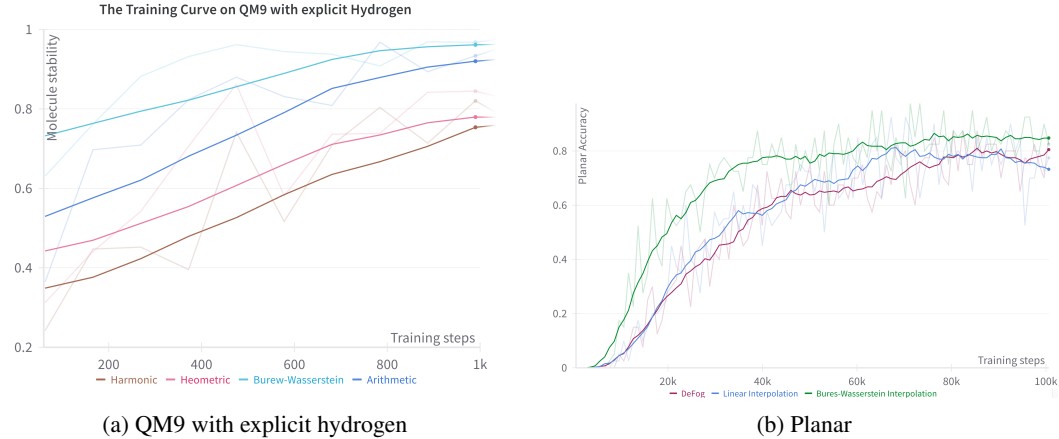

(a) QM9 with explicit hydrogen         (b) Planar

Figure 8: Training curves on QM9 and planar datasets with explicit hydrogen.

## I.8 CONVERGENCE ANALYSIS

Fig. 8 are the training convergence analysis on Planar and QM9 dataset, showing that BWFlow provides a fast convergence speed than others.

## J USAGE OF LARGE LANGUAGE MODELS (LLMS)

We used a large language model to assist with writing and editing this manuscript, primarily for grammar, style, and clarity. The authors are fully responsible for the content and scientific integrity of the work.

| FM type | Interpolation | comm-20 | | |
|---------|---------------|---------|-------|--------|
| | | Deg. | Clus. | Orbit. |
| Discrete | Linear | 0.071 | 0.115 | 0.037 |
| | Harmonic | 0.011 | 0.036 | 0.027 |
| | Geometric | 0.047 | 0.083 | 0.02 |
| | BW | 0.009 | 0.013 | 0.017 |

Table 16: Quantitative experimental results on COMM20 (smaller dataset).

| Dataset | Interpolation | Metrics | | | | | |
|---------|--------------|---------|--------------|--------------|--------------------|-----------------|-----------|
|         |              | $\mu$   | V.U.N(%)     | Connected(%) | Charges($10^{-2}$) | Atom($10^{-2}$) | Angles(°) |
| QM9 (w/o h) | MiDi     | 1.00    | 98.0         | 100.0        | 0.4                | 5.1             | 1.49      |
|         | Linear Flow  | 1.60    | 79.33        | 52.3         | 0.7                | 14.0            | 8.77      |
|         | BWFlow       | 1.02    | 99.8         | 100.0        | 0.4                | 4.8             | 1.53      |

Table 17: Quantitative experimental results on QM9 datasets without explicit hydrogen in 3D molecule generation.

| Symbol | Meaning |
|--------|---------|
| **General flow matching** | |
| $\mathcal{S}$ | State space of variables (e.g., $\mathcal{X} \in \mathcal{S}$) |
| $\mathcal{X}$ / $X$ / $\boldsymbol{X}$ (with $X \sim p(\mathcal{X}; \boldsymbol{X})$) | Random variable / realization / latent parameters |
| $p_0, p_1$ | Source and target distributions over $\mathcal{X}$ |
| $p_t(\mathcal{X}) = [\psi_t p_0](\mathcal{X})$ | Time–continuous probability path between $p_0$ and $p_1$ |
| $\psi_t$ | Time–dependent flow map (CNF flow) |
| $u_t$ | True velocity field generating $p_t$ |
| $v_t^\theta$ | Parameterized velocity field |
| $\eta_0, \eta_1$ | Probability measures on $\mathcal{S}$ in OT formulation |
| $\Pi(\eta_0, \eta_1)$ | Set of couplings between $\eta_0$ and $\eta_1$ |
| $\pi \in \Pi(\eta_0, \eta_1)$ | Transport plan (joint measure on $\mathcal{S} \times \mathcal{S}$) |
| $c(X, Y)$ | Transport cost between $X$ and $Y$ |
| $\mathcal{W}_c(\eta_0, \eta_1)$ | Wasserstein distance associated with cost $c$ |
| **Graphs and Graph Markov Random Fields** | |
| $\mathcal{G} = \{\mathcal{V}, \mathcal{E}, \mathcal{X}\}$ | Random graph (nodes, edges, node features) |
| $G = \{V, E, X\}$ | Realization of $\mathcal{G}$, with nodes, edges and node features |
| $\mathcal{V} = \{v\}, \mathcal{E} = \{e_{uv}\}, \mathcal{X} = \{x_v\}$ | Random node set, edge set, and feature set |
| $\boldsymbol{W} \in \mathbb{R}^{|\mathcal{V}| \times |\mathcal{V}|}$ | Weighted adjacency matrix of the graph distribution |
| $\boldsymbol{X} = [\boldsymbol{x}_1, \dots, \boldsymbol{x}_{|\mathcal{V}|}]^\top$ | Node feature matrix of the graph distribution |
| $\boldsymbol{D} = \mathrm{diag}(\boldsymbol{W}\boldsymbol{1})$ | Degree matrix ($\boldsymbol{1}$: all-one vector) of the graph distribution |
| $\boldsymbol{L} = \boldsymbol{D} - \boldsymbol{W}$ | Graph Laplacian matrix of the graph distribution |
| $\varphi_1(v), \varphi_2(u, v)$ | Node-wise and pair-wise MRF potentials |
| $\boldsymbol{\mu}_v$ | Node-specific latent mean for $\boldsymbol{V}x_v$ |
| $\boldsymbol{\Lambda}$ | Covariance Matrix |
| **Bures-Wasserstein Flow Matching** | |
| $\eta_{\mathcal{G}_j}, \eta_{\mathcal{X}_j}, \eta_{\mathcal{E}_j}$ | Measure over graphs (factorized as node and edge measure) |
| $d_{\mathrm{BW}}(\mathcal{G}_0, \mathcal{G}_1)$ | Bures–Wasserstein distance between two graph distributions |
| $\mathcal{G}_t = \{\mathcal{V}, \mathcal{E}_t, \mathcal{X}_t\}$ | Intermediate random graph along BW interpolation |
| $p(\mathcal{G}_t \mid G_0, G_1)$ | Conditional probability path induced by BW interpolation |
| $v_t(G_t \mid G_0, G_1)$ | Conditional velocity at graph state $G_t$ |
| $v_t(E_t \mid G_0, G_1)$ | Conditional edge velocity |
| $v_t(X_t \mid G_0, G_1)$ | Conditional node-feature velocity |
| $v_t^\theta(G_t)$ | Parameterized velocity used in training / sampling |
| $\mathrm{Categorical}([\boldsymbol{X}_t]_v)$ | Categorical distribution over node feature states at node $v$ |
| $\mathrm{Bernoulli}([\boldsymbol{W}_t]_{uv})$ | Bernoulli edge distribution between nodes $u$ and $v$ |
| $\delta(G_i, \cdot)$ | Dirac distribution concentrated at graph $G_i$ ($i = 0, 1$) |

Table 18: Notation Table.

