# OpenReview forum: "Bures-Wasserstein Flow Matching for Graph Generation"
_ICLR.cc/2026/Conference — ICLR 2026 Poster_

### Official Review · Reviewer_PTxT · 2025-10-31

**Soundness:** 3
**Presentation:** 3
**Contribution:** 3
**Rating:** 8
**Confidence:** 3

**Summary:**

This paper proposes a novel graph generation method using flow matching. First, it is pointed out that methods generating graphs via linear interpolation fail to smoothly change graph statistics, leading to poor flow velocity estimation accuracy and convergence gaps. Building upon this, the proposed method models a graph using Markov Random Field, estimates a vector field derived by the optimal transport on this model using the flow matching. Numerical experiments applied the proposed method to three graph generation tasks, 2D and 3D molecular graph generation tasks, verifying its generation performance.

**Strengths:**

1. (Originality) In graph generation tasks using flow matching, incorporating graph structure is non-trivial. This paper solves this issue by modeling a graph using the Markov Random Field. To my knowledge, this is the first paper to propose such a solution.
2. (Quality) There is design flexibility in how to set the velocity field in flow matching. This paper provides a theoretical background by employing the Optimal Transport. Furthermore, while optimal transport computations are typically high, the proposed method overcomes this problem using the Markov Random Field. This makes the approach practically implementable.
3. (Clarity) The writing is clear. It explains the necessary background knowledge (Markov Random Fields, Optimal Transport) to understand the proposed method. The mathematical descriptions are also appropriate. Therefore, I had no significant difficulties in understanding the paper's main points.
4. (Significance) Numerical experiments demonstrate generation performance of the proposed method surpasses existing methods, which strengthens the significance of the work in the area of graph generation research.

**Weaknesses:**

1. The paper points out that the problem with the existing methods is that they ignore graph structure and generate nodes and edges independently. However, the experiments verify the limitations of linear interpolations. Therefore, it seems unclear whether the problem lies in ignoring graph structure or in using linear interpolations.
2. I have a question about the interpretation of the results in Figure 3a. Looking at Figure 1a of the introduction, the authors assume that ideal interpolation changes graph statistics linearly. However, the discussion on Figure 3a suggests that it is desirable that statistics first increase and then decay. It seemingly contradicts the claim in Figure 1a and it would be more natural to claim that linear interpolation is preferable.

**Questions:**

1. I would like to clarify the experimental setting details for Figure 1a in the Introduction.
2. I would like the authors to address the question regarding the discussion of the results in Figure 3a, which is the second point under Weaknesses.

**Details Of Ethics Concerns:**

N.A.

---

> ### Author Response · Authors · 2025-11-20
> **Response to Reviewer PTxT**
>
> Dear reviewer PTxT,
>
> We sincerely appreciate your recognition of the quality of the paper. We address your remaining concerns below.
>
>
> > W1. … it seems unclear whether the problem lies in ignoring graph structure or in using linear interpolations.
>
> We wish to clarify that ignoring graph structure and using linear interpolations are not two issues disentangled. We apologize for the ambiguity, but by saying “linear interpolation”, we actually mean **using linear interpolation in the disjoint space of nodes and edges**. The action is problematic as it implicitly ignores the fact that graphs are connected systems in non-Euclidean geometry, where nodes and edges co-evolve rather than evolve independently. Therefore, the limitation is not merely the linearity of the interpolation, nor merely the neglect of graph structure, but the combination of both: linear interpolation carried out in a space that does not preserve structural dependencies. This joint effect makes the probability path suboptimal for graph generation and leads to degraded interpolation behavior.
>
> Thank you for pointing this out and we have edited the abstract to avoid this ambiguity.
>
>
> > W2&Q2. I have a question about the interpretation of the results in Figure 3a….
>
> Thank you for highlighting this ambiguity. We would like to clarify the interpretation of the paths shown in Fig. 3a from two perspectives:
>
> - **"Smooth" does not imply "linear"**: Our assumption is not that an ideal interpolation should make graph statistics evolve linearly. Rather, a good interpolation should 1) **Point the model correctly toward the data distribution** (i.e., avoid the flat region where the velocity provides little useful signal), and 2) **sufficiently explore the data space**, including both the transition regions and some out-of-distribution (OOD) regions. The evolution does not need to be linear, where Fig. 1a simply illustrates one of many valid paths.
>
> - **Straight line vs generalization**: The path visualized in Fig. 1a that linearly interpolates the graph statistics is the one ensuring the lowest cost. But in practice, this path may negatively impact the model generalization, as a straight line makes the algorithm vulnerable to OOD samples. In contrast, carefully permitting early-stage exploration of nearby OOD regions can improve robustness without violating the ideal interpolation assumptions. In Fig. 3a, we intentionally encourage BWFlow to maintain such early-stage exploration by adding a small diagonal identity term to the Laplacian (see eq. 8 where $\boldsymbol{\Lambda} = (\nu \boldsymbol{I}+\boldsymbol{L}) \dots$. This serves two purposes simultaneously: 1) it avoids numerical instability when inverting $\boldsymbol{L}$, and 2) it exposes the model to mild OOD perturbations without violating the ideal interpolation assumptions. We empirically find the design yielding a more robust path for training. Theoretically, if exploration of OOD regions were omitted, the interpolation would more closely resemble the green curve in Fig. 1a.
>
>
>
> > Q1. … experimental setting details for Figure 1a in the Introduction.
>
> We apologize that the detailed experimental setting for Fig. 1a was deferred to Section 4.1 and Appendix I.6 due to space limitations. Fig. 1a is generated using a representative plain graph dataset, SBM. At each time step $t$, we compute the average maximum mean discrepancy ratio (A.Ratio) between the interpolants and the real data graphs over multiple graph statistics, including orbit, clustering, spectral, wavelet, and degree ratios (definitions provided in [1–3]). The "ideal velocity" (green curve) in Fig. 1a is a synthetic reference path used purely for conceptual illustration. It does not correspond to a real model output, but is designed to help explain the intuition behind optimal interpolation.
>
>
> [1] K. Martinkus et al. “Spectre: Spectral conditioning helps to overcome the expressivity limits of one-shot graph generators.” In ICML 2022.
>
> [2] A. Bergmeister et al, “Efficient and scalable graph generation through iterative local expansion.” In ICLR, 2023.
>
> [3] Y. Qin et al, “Defog: Discrete flow matching for graph generation.” In ICML 2025.

---

> > ### Comment · Reviewer_PTxT · 2025-11-23
> >
> > Dear Authors
> >
> > Thank you for answering my questions. The responses mostly answered my questions, and I will keep my score.

---

> > > ### Author Response · Authors · 2025-11-24
> > > **Thank you!**
> > >
> > > Dear reviewer PTxT,
> > >
> > > Thank you again for your time and effort in reviewing our paper. We are always open to further discussions if any question arises. We sincerely appreciate your valuable insights, positive feedback, and insightful suggestions.
> > >
> > > Best regards,
> > >
> > > The authors

---

### Official Review · Reviewer_sj7x · 2025-10-31

**Soundness:** 3
**Presentation:** 2
**Contribution:** 3
**Rating:** 6
**Confidence:** 2

**Summary:**

This paper proposes Bures–Wasserstein Flow Matching (BWFlow), a generative modeling framework for graphs that integrates flow matching with the Bures–Wasserstein (BW) geometry on a Graph Markov Random Field (GraphMRF) representation. The authors first formalize graphs as random fields on nodes and edges (GraphMRF), establishing a Gaussian joint distribution whose covariance is tied to the graph Laplacian. They then derive a closed-form expression of the Bures–Wasserstein distance between two graphs, together with an analytical interpolation path and conditional velocity field, which enable flow matching without simulation. The method is further extended to a discrete version (with categorical node features and Bernoulli edges), and training/inference procedures are given. Comprehensive experiments on graph generation benchmarks are reported.

**Strengths:**

1. The paper presents a mathematically coherent framework connecting graph optimal transport and flow matching. The closed-form derivations of the BW interpolation and velocity fields are valuable theoretical contributions.

2. Using the BW metric on graph Laplacians provides a geometrically meaningful way to measure and interpolate between graph distributions, which addresses a long-standing issue of linear interpolation violating graph manifold constraints.

3. The experimental section is relatively comprehensive, covering various types of graph data, including plain graphs, molecular graphs, and 3D molecular structures.

**Weaknesses:**

1. The methodological novelty is mainly compositional: the work combines existing Bures–Wasserstein distance (already applied to graph covariance learning) with the established flow-matching framework under a Graph MRF formulation, resulting in limited originality.

2. The empirical evaluation lacks consistency with standard benchmarks (e.g., DiGress, GruM, DeFoG) and omits common molecular-graph metrics such as Validity, Uniqueness, and Novelty. The results would be more convincing if experiments used the same datasets and evaluation metrics as prior baselines.

3. The paper does not isolate the effect of replacing Euclidean interpolation with the BW distance, nor does it provide a systematic comparison between the continuous and discrete variants. Without such analysis, it is unclear whether BW geometry truly contributes to the observed performance.

4. The notation system is heavy, and some equations are symbolically dense without intuitive explanation or visualization.

5. The paper does not mention a code release to ensure reproducibility.

**Questions:**

1. In Fig. 1(c), when flow steps = 1.0, the Time Distortion method yields a slightly smaller value than the BW interpolation. Does this imply that Time Distortion is comparably efficient in this setting?

2. Could the authors elaborate on the modeling assumption $p(\mathcal{G};\mathbf{G})=p(\mathcal{X};\mathbf{X},\mathbf{W})\cdot p(\mathcal{E};\mathbf{W})$? Why are the node features and edge structures conditionally independent, except for the coupling through 𝑊? How critical is this assumption for the GraphMRF formulation?

3. Does the model scale to large graphs?

---

> ### Author Response · Authors · 2025-11-20
> **Response to Reviewer sj7x (Part 1)**
>
> Dear reviewer sj7x,
>
> Thank you for your comments. We aim to address the weaknesses and questions as follows.
>
> > W1. The methodological novelty is mainly compositional: …
>
>
> We respectfully disagree with the characterization that the methodology is mainly compositional. While BW distance and flow matching each have prior usage, our contributions lie not merely in combining known components, but in: 1) identifying probability path construction as a fundamental issue in graph generation; 2) establishing a theoretically grounded framework for path and velocity construction based on the inspection of graph characteristics, and 3) translating the theoretical results into practically implementable algorithms that can be used for a variety of graph generation tasks. We believe these go beyond compositionality and constitute a substantive methodological advance.
>
> **Identifying probability path design as a fundamental problem in graph generation.** To the best of our knowledge, our work is the first to systematically study how probability paths affect graph generative modeling. While prior works have noted the limited ability of linear interpolation in the disjoint space of nodes and edges, they only rely on heuristic design to mitigate this problem. None has analyzed why existing interpolations are suboptimal. We are the first to provide both empirical and theoretical evidence showing that linear paths violate the structural manifold assumptions underlying graph distributions, which prevents models from capturing the global co-evolution of the nodes and edges. This diagnosis itself is a conceptual contribution, as it reveals a core limitation of current graph generative models that has not been resolved.
>
>
>
> **A theoretically grounded framework for path and velocity construction based on inspecting graph characteristics.** The choice of BW interpolation, over alternatives like harmonic or geometric interpolation, was not made lightly by simply reusing a known distance, but **stems from our meticulous inspection of the graph properties** - we need an interpolation to capture global co-evolution of each component and smoothly connect two graph distributions. We also have to emphasize that **the integration of BW interpolation into a flow-matching (FM) framework for graphs is non-trivial**: one must derive both the OT path between graph distributions and the corresponding velocity field compatible with FM under the MRF formulation. These results do not exist in prior literature. Hence, the novelty is in developing a principled framework that replaces heuristic path design with a theoretically justified solution grounded in graphs and geometry.
>
>
> **A practical algorithm that can be used for graph generation.**
> As Reviewer PTxT correctly points out, **turning the theoretical framework into a practical and effective model requires substantial algorithmic development**. Although the main text focuses on the conceptual framework rather than implementation details, significant engineering and empirical efforts were taken to make the method stable, efficient, and interpretable. These include systematically exploring the design space of training and sampling strategies (Appendix E), developing algorithmic architecture that reduces computational complexity, and conducting detailed behavior analysis to understand and improve the dynamics of the learned flow matching model (Appendix F). Importantly, our contribution also lies in **generalizing BWFlow to seamlessly handle discrete, continuous, and mixed graph generation tasks**, each of which presents a distinct type of data structure. Ensuring compatibility across these settings is not a straightforward extension, but a necessary contribution that makes BWFlow applicable to a wide variety of real-world tasks.
>
>
> We hope this clarification helps highlight the originality of our work.

---

> ### Author Response · Authors · 2025-11-20
> **Response to Reviewer sj7x (Part 2)**
>
> > W2. …The results would be more convincing if experiments used the same datasets and evaluation metrics as prior baselines.
>
>
> We would like to clarify that our empirical evaluation **is consistent with standard benchmarks used in prior work**, such as DiGress, GruM, and DeFoG. Specifically, we conduct plain graph generation experiments on the Planar, SBM, and Tree datasets, and 2D molecular graph generation on MOSES and GUACAMOL (Appendix I.4), which are exactly the datasets used in the aforementioned baselines.
>
> We suspect that the confusion may arise because we additionally include 3D molecular graph generation, a task that DiGress and DeFoG cannot address. This setting requires models to generate not only atom and bond types, but also 3D coordinates for each atom. For this task, we compare against Midi and FlowMol, two state-of-the-art diffusion/flow-based models specifically developed for 3D molecule generation.
>
> Regarding evaluation metrics, we would also like to clarify that **Validity, Uniqueness, and Novelty (V.U.N.) are reported in both the graph generation (Table 1) and molecular generation benchmarks (Table 2). The V.U.N. score in the main tables is the standard unified evaluation metric widely adopted in graph generation**. For completeness, the three metrics, Validity, Uniqueness, and Novelty, are also reported individually in Appendix H.1 (Tables 7 and 11).
>
>
> > W3. … the effect of replacing Euclidean interpolation with the BW distance… comparison between the continuous and discrete variants
>
>
> We would like to clarify that **we do provide an ablation study isolating the effect of the interpolation choice to compare linear vs. BW interpolation, while keeping all other factors fixed**. The visual comparison is shown in Fig. 3b, and detailed numerical results are reported in Tables 7 and 8 (Appendix H.1), where Table 7 corresponds to plain graph generation and Table 8 to 3D molecule generation. In this study, we control for all other variables, including model architecture (same graph transformer), path manipulation techniques (no target guidance or time distortion), and all training and sampling configurations.
>
> We acknowledge that the linear interpolation was labeled as “Arithmetic interpolation” in the tables, which may have caused confusion. We revised the terminology in the updated manuscript. We believe the results clearly demonstrate that BW interpolation provides meaningful improvements over linear interpolation in graph generation.
>
>
> Regarding the **continuous vs. discrete variants**, we would like to clarify our motivation for deriving both is not to identify which formulation is superior, **but rather to ensure that BWFlow is compatible with discrete, continuous, and mixed graph generation tasks**. Prior work has shown that the choice between discrete and continuous diffusion is task-dependent: discrete diffusion is typically more effective for plain (2D) graph generation [1], while mixed continuous-discrete formulations achieve superior performance for 3D molecule generation [2]. Our design therefore aims to generalize BWFlow so that it can be applied to all of these scenarios.
>
> We also provide a small-scale ablation in Appendix H.1 (Table 8) comparing the two variants, and the results empirically align with [1] [2] (discrete methods excel in 2D generation and mixed methods win 3D generation). Since their advantages differ across tasks, we view the continuous and discrete variants as complementary rather than competing.
>
>
>
>
> > W4. The notation system is heavy, and some equations are symbolically dense without intuitive explanation or visualization.
>
> While we are devoted to providing intuition for key theoretical results via the gray remark boxes and to making visualizations for the overall pipeline and key results, we agree that the rather theoretical foundations require a relatively heavy notation system, which can make some equations appear symbolically dense. To improve clarity, we added a notation table (Table 18) in the appendix in the revised version. We hope this will help readers better navigate the notation and follow the theoretical developments.
>
> Please kindly let us know if you still feel some parts are unclear. We are committed to addressing the ambiguities/unclearness, and we value every opportunity to improve the manuscript!
>
> > W5. Code release
>
> We now provide the code in the supplementary material illustrating the main pipelines of our method and hopefully this will address the concern about reproducibility.

---

> ### Author Response · Authors · 2025-11-20
> **Response to Reviewer sj7x (Part 3)**
>
> > Q1. In Fig. 1(c), ... Does this imply that Time Distortion is comparably efficient in this setting?
>
> Thank you for the question. We address it as follows:
>
> - Though conceptually simple, **time distortion requires extensive hyperparameter tuning in practice, which is far from “efficient”**. As for DeFog, achieving the best performance required grid search across multiple distortion schedules (uniform, cosine, beta, polynomial increase/decrease, etc.). Even for the polynomial decrease schedule (the best-performing one reported) with a general form $t^\prime = 2t - t^\eta$, DeFog searches $\eta \in (1, 3)$ with an interval of 0.1. The optimal distortion also varies across datasets, so the heavy tuning process cannot be avoided when adapting to new settings. In contrast, BW interpolation requires no additional hyperparameters and applies uniformly across datasets, avoiding this heavy tuning overhead.
>
> - **BW interpolation and Time Distortion are not mutually exclusive**; they are two orthogonal techniques that can independently improve performance. BWFlow built on top of time distortion is particularly advantageous in training stability and sampling efficiency, especially when the number of sampling steps is small. To support this statement, beyond the main experiments, we further ran an ablation varying the sampling steps from 10 to 50 and consistently observed that adding BW interpolation improves performance. This suggests that even when time distortion is helpful, switching from linear to BW interpolation remains beneficial.
>
>
> ### Comparison of linear interpolation+time distortion (Linear+TD) vs BWFlow+TD across sampling steps
>
> | Sampling Steps | Method       | planar V.U.N (mean ± std) | average_ratio (mean ± std) |
> |--:|---|----|----|
> | 50 | Linear+TD  | 0.74 ± 0.07  | 5.40 ± 1.28 |
> |    | BWFlow+TD   | **0.825 ± 0.05**   | **3.19 ± 0.93**  |
> | 30 | Linear+TD  | 0.72 ± 0.074| 6.3 ± 1.90   |
> |  | BWFlow+TD   | **0.77 ± 0.04**         | **4.1 ± 1.00**  |
> | 20| Linear+TD | 0.09 ± 0.03| 7.39 ± 1.39  |
> |   | BWFlow+TD   | **0.22 ± 0.06**        | **6.34 ± 0.27**   |
> | 10 | Linear+TD  | 0.0 ± 0.0 | 84.74 ± 2.03   |
> |   | BWFlow+TD   | 0.0 ± 0.0| **28.97 ± 3.73**           |
>
>
>
> - **Interpretation of Fig. 1(c).** For visualization clarity, Fig. 1(c) plots BWFlow without time distortion. This design choice avoids confusing the effects of BW interpolation and time distortion on the path, which would be misleading. Both lines converge to a similar value because sufficient sampling steps are given. Given that time distortion and BW interpolation can in parallel boost graph generation, we believe that, for a fair performance comparison, it is more meaningful to fix the time distortion and solely compare the methods with and without BW interpolation (just as we did in the main experiment and the above example).  We sincerely apologize for the ambiguity and will address this in the paper.
>
>
>
>
>
> > Q2. Could the authors elaborate on the modeling assumption… ?
>
> We apologize for the ambiguity and would like to clarify both the notation and the modeling assumption.
>
> First, $G \sim p(\mathcal{G}\mid \boldsymbol{\mathcal{G}}) = p(\mathcal{X}, \mathcal{E}\mid \boldsymbol{X}, \boldsymbol{W})$ denotes a **hierarchical graphical model** parameterized by the latent variables $\boldsymbol{X}$ (latent node features) and $\boldsymbol{W}$ (latent graph structure). To make the dependencies explicit, we provide the corresponding graphical model visualization in Appendix A.2, Figure 4.
>
> Applying the chain rule gives:
> $$p(\mathcal{X}, \mathcal{E} \mid \boldsymbol{X}, \boldsymbol{W})
> = p(\mathcal{X} \mid \mathcal{E}, \boldsymbol{X}, \boldsymbol{W}) \, p(\mathcal{E} \mid \boldsymbol{X}, \boldsymbol{W}).$$
> From the factorization encoded by the graphical model, we have the conditional independence $\mathcal{E} \perp \mathcal{X} \mid \boldsymbol{W}, \boldsymbol{X}$. The only modeling assumption used to simplify the likelihood further is,
> $$p(\mathcal{E} \mid \boldsymbol{W}, \boldsymbol{X}) = p(\mathcal{E} \mid \boldsymbol{W}),$$
> i.e. $\boldsymbol{W}$ alone governs the structural prior. In other words, edges do not depend on the latent node features $\boldsymbol{X}$. This follows the classical formulation of Markov random fields [3], in which 1) the graph structure serves as a prior and is generated first, and 2) the node features are *emitted* based on that structure. This assumption is commonly used and primarily for simplicity and tractability. It avoids the redundancy to model the interaction between node features and graph structure twice, i.e. no need to model $p(\mathcal{E} \mid \boldsymbol{X}, \boldsymbol{W})$ that do not provide extra dependency information but only make the joint density complex.

---

> > ### Author Response · Authors · 2025-11-20
> > **Response to Reviewer sj7x (Part 4)**
> >
> > > Q3. Does the model scale to large graphs?
> >
> > Thank you for this question. We address scalability from two perspectives: *computational efficiency and performance when scaling to larger graphs*.
> >
> > - **Computational efficiency**. We show in App. F.4 that BWFlow can scale to large graphs with reasonable computational cost. As noted in the limitations, exact BW interpolation introduces an additional $O(N^3)$ linear algebra operations during path construction. For large but sparse graphs, we demonstrate in App. F.4 that the complexity can be reduced to $O(TN^2)$ using Newton–Schulz iteration or to $O(TNE)$ using least-squares QR factorization, where $T$ is the number of iterative steps and $E$ is the number of edges. These complexities are comparable to those of previous methods (typically $O(N^2)$). Our preliminary experiments in App. F.4 further show that iterative solvers with 3-10 iterations provide performance comparable to exact BW interpolation.  We believe this already highlights the potential of BWFlow to scale computationally to larger graphs.
> >
> >
> > - **Performance under graph size scaling**. One of the purposes for conducting 3D molecule generation experiments (Table 2) is to explicitly validate the performance scalability of BWFlow to large graphs. Compared with the commonly used 2D molecular benchmarks in prior work, 3D molecular graphs, especially the GEOM-DRUGS dataset, are significantly larger, with up to ~200 nodes, whereas GUACAMOL has up to 88 nodes (see Table 9 for a comparison). BWFlow performs successfully on the 3D molecule generation task, which we believe provides strong evidence of its capacity to scale to larger graphs.
> >
> >
> > [1] C. Vignac et al. “DiGress: Discrete Denoising diffusion for graph generation”. ICLR 2023
> >
> > [2] C. Vignac et al. “MiDi: Mixed Graph and 3D Denoising Diffusion for Molecule Generation” ECML 2023
> >
> > [3] X. Zhu et al. “Semi-supervised learning: From Gaussian fields to Gaussian processes. “ School of Computer Science, Carnegie Mellon University

---

> > > ### Author Response · Authors · 2025-11-27
> > >
> > > Dear reviewer sj7x,
> > >
> > > Once again, we sincerely thank you for your time and for the initial reviews that allowed us to strengthen the paper. We have revised the manuscript to address each weakness and question, and have also released the code.
> > >
> > > As the discussion period is concluding, we want to make sure that our responses have fully addressed your concerns. We are keen to take advantage of the period to clarify any remaining concerns!
> > >
> > > Best regards,
> > >
> > > The authors

---

### Official Review · Reviewer_W2yQ · 2025-10-31

**Soundness:** 3
**Presentation:** 3
**Contribution:** 3
**Rating:** 6
**Confidence:** 4

**Summary:**

The paper proposes BWFlow, a flow-matching framework for graph generation that constructs a smooth, joint probability path by modeling graphs as Markov random fields (MRFs) and leveraging optimal transport between MRFs. It argues that existing diffusion/flow methods use independent, linear node/edge interpolations that disrupt graph dependencies, leading to irregular paths and unstable training. By enforcing co-evolution of nodes and edges along the derived optimal path, BWFlow aims to improve training dynamics and sampling convergence. Experiments on plain graph and molecular generation report competitive or better performance, faster convergence, and more efficient sampling.

**Strengths:**

The paper is well-written, clear, and methodologically sound, presenting complex ideas with precision and rigor.

It establishes a theoretical framework for probability-path construction directly within Markov Random Field (MRF) space, enabling coherent joint node–edge evolution that respects graph-structured dependencies.

It introduces a novel integration of MRFs with optimal transport, using MRFs as the ambient representation space for probability paths in a way that naturally aligns with graph topology and conditional dependencies.

**Weaknesses:**

While acknowledged in the limitations, BWFlow incurs substantial computational overhead, and reducing this cost appears non-trivial.

The experimental evaluation omits several GNN-based baselines, including DisCo (Xu et al., 2024), limiting the completeness of the comparison.

**Questions:**

Given the computational overhead, would a hybrid schedule be feasible—one that leverages linear interpolation for early stages and switches to the proposed BWFlow when most beneficial?


In Section A.3 there are cases where GraphMRF may be suboptimal for certain graph structures. Can structural information be injected to improve performance—for example, via higher-order graph features, subgraph/ motif features, or structural role embeddings?

line 479 "When scaled up to large
but sparse graphs, the complexity can be reduced to O(TN2)", what is T

---

> ### Author Response · Authors · 2025-11-20
> **Response to Reviewer W2yQ (Part 1)**
>
> Dear reviewer W2yQ,
>
> Thank you for acknowledging the quality of our manuscript. Please find a detailed reply for each of the points below.
>
> > W1: … BWFlow incurs substantial computational overhead …;
>
>
> While we acknowledge that exact BW interpolation is indeed expensive, BWFlow can be made computationally efficient with certain designs, as shown in App. F.4. By replacing the exact solver of matrix inverse (for graph laplacian) with iterative approximations, the complexity reduces to $O(TN^2)$ with Newton-Schulz iteration or to $O(TNE)$ with least-squares QR factorization (LSQR), where $T$ is the number of iterative steps and $E$ is the number of edges. These complexities are comparable to linear interpolation (typically $O(N^2)$). Our preliminary experiments in App. F.4 shows that with 3-10 iterations, the iterative solvers match the performance of exact BW interpolation. Thus, although the exact formulation is costly, the approximate versions are scalable and already effective in practice.
>
> > Q1: … would a hybrid schedule be feasible?...
>
>
> A hybrid schedule could be an appealing direction. The main challenge, however, lies in determining when to transition between the two regimes without degrading generation quality. As discussed in our response to reviewer mpf9, the exact transition point from the sharply evolving and the flattened region varies across datasets: approximately 0.8 for SBM, 0.5 for Planar, and 0.6 for QM9. Without an adaptive scheduling mechanism, identifying this transition point would require dataset-specific hyperparameter search, which may limit practicality.
>
>
> We sincerely appreciate the reviewer for highlighting this promising research direction, and we will consider it for future work.
>
>
> > W2. Comparison with GNN-based baselines, such as Disco
>
> We wish to clarify that we do provide a comprehensive comparison with mainstream graph generation models in Appendix I.2, Table 11, including **Disco** [5] and others like CatFlow [6], EDGE [7], etc. However, as these models have been consistently reported to underperform more recent state-of-the-art approaches such as DeFog [3] and Cometh[8], we did not reproduce them in our setting at the time of submission.
>
> To further strengthen the performance comparison, we have now reproduced Disco under our experimental setting and report the results below:
>
> | Model | Class | V.U.N. ↑ (Planar) | A.Ratio ↓(Planar) | V.U.N. ↑ (SBM) |  A.Ratio ↓ (SBM) |
> |----|-----|-----|---|----------|-------------|
> | DisCo | Diffusion | 57.5 ± 2.5 | 9.0 ± 1.4 | 55.0 ± 5.9 | 11.6 ± 2.9 |
>
>
>
> It is clear that Disco still underperforms other models in Table 1. We have added this to the main comparison table 1.

---

> ### Author Response · Authors · 2025-11-20
> **Response to Reviewer W2yQ (Part 2)**
>
> > Q2. …Can structural information be injected…?
>
> We thank the reviewer for their suggestions. We wish to address the question from two complementary perspectives:
>
> - *Can structural information be injected into the model as information for prediction?* Yes and this is already supported by our framework. There has been extensive investigation into incorporating structural information such as Relative Random Walk Probabilities (RRWP) [1] and lower-frequency eigenvectors of the Laplacian matrix [2]. We refer to Defog [3], Appendix G.4, for a comprehensive discussion of structural feature choices. In our experiments, we follow Defog and incorporate RRWP as additional features. However, we note that such features generally encode low-frequency (global) and lower-order structural information, and do not fully capture higher-order structural patterns. Thus, injecting higher-order information into the model could be a promising future work.
>
>
> - *Can higher-order information and subgraph/ motif features, be used to inform the probability path construction i.e., the interpolation?* For BWFlow, this is theoretically possible but practically nontrivial. As stated in Eq. (7) and in the surrounding discussion, we follow the MRF assumption in [4] and decompose the density into node-wise and pair-wise potentials, which is actually a second-order approximation of the full potential. In general, the *full version* of the  density of node features is defined as:
> $$p(\mathcal{X}) = \frac{1}{Z} \prod_{c \in C(\mathcal{G})} \Phi_c\left(\mathcal{X}_c\right)$$
> where Z is the partition function, $C(\mathcal{G})$ is the clique set defined on the graph $\mathcal{G}$ and $\Phi_c\left(\mathcal{X}_c\right)$ is potential corresponding to clique $c$ with the node subset $\mathcal{X}_c$. When $C(\mathcal{G})$ only contains first-order (node-wise, $\Phi_1 = \prod \psi_1 (v)$) and second-order (edge-wise, $\Phi_2 = \prod \psi_2(u, v)$) potential, the above density reduces to Eq. 7. To incorporate higher-order information, one could extend second-order potentials with higher-order potentials, such as ones representing triangle, motif, or k-clique. However, doing so would lose the key properties that second-order MRFs can be represented by a colored Gaussian distribution, thus making the interpolation construction computationally intractable. As a result, incorporating higher-order potentials directly into the transport path would likely lead to prohibitive computational complexity and significantly complicate algorithmic design.
>
> > Q3. What is T?
>
> We apologize for the ambiguity. $T$ is the number of iterations steps in iterative solving (e.g. the steps of gradient descent). In our preliminary trials with LSQR, we observe that with 3-10 steps of iterative solving are sufficient to make the algorithm converge, which is discussed in App. F.4.
>
>
>
>
> [1] L. Ma et al. “Graph Inductive Biases in Transformers without Message Passing”, ICML 2023
>
> [2] K. Martinkus et al. “SPECTRE: Spectral Conditioning Helps to Overcome the Expressivity Limits of One-shot Graph Generators”, ICML 2022.
>
> [3] Y. Qin et al. “DeFoG: Discrete Flow Matching for Graph Generation”, ICML 2025.
>
> [4] X. Zhu et al. “Semi-supervised learning: From Gaussian fields to Gaussian processes. “ School of Computer Science, Carnegie Mellon University
>
> [5] Z. Xu et al. “Discrete-state Continuous-time Diffusion for Graph Generation.”, NeurIPS 2024
>
> [6] F. Eijkelboom et al. “Variational Flow Matching for Graph Generation”. NeurIPS 2024
>
> [7] X. Chen et al. “Efficient and Degree-Guided Graph Generation via Discrete Diffusion Modeling” ICML 2023.
>
> [8] A. Siraudin “Cometh: A continuous-time discrete-state graph diffusion model”. TMLR 2025

---

> > ### Author Response · Authors · 2025-11-27
> >
> > Dear reviewer W2yQ,
> >
> > Thank you again for your time in reviewing our paper and the valuable feedback. As we are approaching the end of the author-reviewer discussion period, we want to ensure our responses have sufficiently addressed your concerns. We would be grateful if you could let us know whether any further clarification or details are needed to finalize your evaluation. We remain fully open to further discussion!
> >
> >
> > Best regards,
> >
> > The authors

---

### Official Review · Reviewer_mpf9 · 2025-11-01

**Soundness:** 3
**Presentation:** 3
**Contribution:** 3
**Rating:** 8
**Confidence:** 2

**Summary:**

This paper studies the graph generation problem, via flow (matching) model. The notable point is that this paper formulates the graph with Markov Random Fields (MRFs) so that the interpolation on GraphMRFs can include the interaction between nodes and edges, greatly different from the previous work, whose node and edge interpolations are independent (Eq. 5). Some followup results, e.g., what is the closed-form interpolation, and the velocity at the interpolation point, are presented.

**Strengths:**

S1. Good motivation. "MRFs organize the nodes/edges as an interconnected system and interpolating between two MRFs captures the joint evolution of the graph system" sounds reasonable to me.

S2. This paper's presentation is nice. Even without checking the detailed mathematical derivations, I can still follow the story in the main content.

S3. The experimental results seem impressive.

**Weaknesses:**

Generally, i am satisfied about the quality of this paper. Here are some questions/suggestion which might be able to improve this paper.

Q1. From lines 66 to 82, it introduces the motivating example of this paper. It is a bit confusing why t=0.8 is such an important point. If this is purely empirical observation, or there are some mathematical reasons?

Q2. Can we understand the proposed method as a kind of latent diffusion/flow method, just like the stable diffusion did? I understand that the proposed method does not include an explicit encoder/decoder, and the I like the interpolations/velocity derived based on the colored
Gaussian distribution. Asking because the Figure 2 somewhat shows such a connection between flows in raw graph space and latent space.

Q3. About the OT paths. To be specific,

Q3.1. According to the Algorithm 1, the interpolation is on the OT path between a pair of G_0 and G_1, but not the in-batch OT coupling between a batch of {G_0} and {G_1}, is that right? But I think it should be easy to generalize to the mini-batch OT coupling setting as [1] did, is that right?

Q3.2 I can buy the story that "capture global co-evolution (Haasler & Frossard, 2024) of the graph components" for the interpolation is important, but whether the OT interpolation/path really matters? Asking because in many recent flow model studies, the straight line interpolation's best advantage is more on the efficiency (so fewer sampling steps), but not on the effectiveness.

[1] Improving and generalizing flow-based generative models with minibatch optimal transport

**Questions:**

Please check the above questions.

---

> ### Author Response · Authors · 2025-11-20
> **Response to Reviewer mpf9 (Part 1)**
>
> Dear reviewer mpf9,
>
> Thank you for your positive feedback on our manuscript. We appreciate the opportunity to discuss the potential points to improve the paper and please find our reply below:
>
>
>
> > Q1. … why t=0.8 is such an important point…?
>
>
> We apologize for the ambiguity. The specific value \(t = 0.8\) is an empirical observation from our experiment and does not have a theoretical significance.  The goal here is not to emphasize this particular value, but rather the presence of a transition point:  before this, the velocity does not correctly point to the target distribution, and whereas after the transition, convergence becomes noticeably sharper. The transition point is identified empirically and varies across datasets. To further support this interpretation, we add the visualization of the training path in Figure 6 (a), (b), and (c). We observe that the transition point consistently exists for Planar, SBM, tree, QM9 datasets, but its exact value differs: approximately 0.8 for SBM, 0.5 for Planar, and 0.6 for tree and QM9.
>
>
>
>
> > Q2. Can we understand the proposed method as a kind of latent diffusion/flow method …?
>
> Thank you for the comment. We agree that BWFlow can be interpreted as a special case of latent diffusion/flow, and we believe this perspective can inspire meaningful follow-up work. We add a short remark before Sec 3.2 to reveal the intrinsic connections between BWFlow and the latent generative model, as we found this viewpoint offers interesting insight into the behavior of BWFlow, which we describe as follows.
>
> To the best of our knowledge, latent diffusion serves two main purposes: (1) improving computational efficiency, and (2) mapping data from the original space to a smoother space that is easier to model. Although BWFlow does not perform dimensionality reduction, it is fundamentally motivated by the assumption that *transforming graphs into a smoother space can benefit training stability and sampling efficiency*. Though in BWFlow, this space is manually constructed rather than learned, in contrast to standard latent diffusion.
>
> **The MRF space we use exhibits properties that are desirable in latent diffusion**. As discussed in App. A.3, transforming a graph into the MRF domain enhances the modeling ability of low-frequency components. This parallels the behavior observed in diffusion models with latent space (e.g., [1]), where early-timestep latent representations retain a larger proportion of low-frequency information (see Fig. 6 in [1]), which is proven helpful in generative models. Thus, viewing BWFlow through the latent diffusion lens reinforces our original intuition, and we agree that this perspective could be valuable for guiding future algorithmic design.
>
>
> Moreover, your comment inspires a promising simplification of our current approach. As shown in Fig. 2, we currently convert the MRF representation back to the graph domain and use a graph transformer to approximate $v_\theta (G_t)$. From the standpoint of latent diffusion, an alternative is to directly parameterize the velocity in the MRF domain and train the model via the KL divergence between colored Gaussian distributions, which could potentially improve efficiency. We greatly appreciate your inspiration and view this as a promising future work.

---

> ### Author Response · Authors · 2025-11-20
> **Response to Reviewer mpf9 (Part 2)**
>
> > Q 3.1: … the interpolation is on the OT path, but not the in-batch OT coupling…? is it easy to generalize to the mini-batch OT coupling as in [2] …?
>
> Yes, you are correct that Algorithm 1 is based on the OT path between a pair of $G_0$​ and $G_1$, rather than in-batch OT coupling across {$G_0$} and {$\{G_1\}$}. However, we have to point out that **generalization to the mini-batch OT coupling is theoretically sound but practically non-trivial**.
>
>
> The practical difficulty lies in computational cost. As discussed in Sec. 3.2.3 of [2], in-batch OT coupling scales cubically in time and quadratically in memory w.r.t. the batch size $B$. The computation of the BW distance alone, without acceleration techniques described in App. F.4, requires $O(N^3)$ per pair (linear algebra operations rather than model complexity). Therefore, performing full in-batch OT coupling would result in an overall computational complexity of $O(B^3N^3)$, which is unfortunately prohibitive in practice. For comparison, even the $O(BN^3)$ cost incurred by our current design already leads to an average 0.26x increase in clock-time during training (as reported in Table 9), underscoring the challenge of scaling to $O(B^3N^3)$.
>
>
> Nonetheless, we fully agree that developing computationally feasible mini-batch OT CFM techniques would be a valuable research direction, and we sincerely appreciate your insightful comment.
>
>
> > Q3.2: … but whether the OT interpolation/path really matters? …
>
> We first give our answer: OT path matters for graph generation effectiveness as it ensures the interpolation lies in a valid data manifold representing graphs.
>
> To the best of our knowledge, in the idealized scenario where a model is perfectly trained (i.e., it provides the correct velocity pointing to the data distribution everywhere) and sampling is performed with infinitely many steps, the choice of interpolation path would have little impact on generation quality. In this limit, any valid transport from the reference distribution to the target distribution should theoretically converge [4].
>
>
> However, such ideal conditions never hold in practice. While a linear path is suboptimal, it typically remains functional. But as illustrated in Fig. 3a, certain interpolations, e.g., the harmonic path in the green line, may still connect the distributions but become highly challenging for the model to follow during finite-step sampling. This leads to clear degradation or even failure in terms of generation quality, as observed in Fig. 3b (blue bars).
>
>
> We believe this highlights a broader point in generative modeling on non-Euclidean geometries (graphs being one example): in these spaces, OT paths can benefit not only efficiency and stability, but also generation quality. Arbitrary interpolation may lead the trajectory outside the trustworthy region of the data manifold (see Fig. 1 in [3] for an intuition). In geometric generative models, the paths close to the OT one ensure interpolation stays in the valid domain [3]. Similarly, BWFlow exploits a path that naturally keeps the interpolants within the valid graph domain throughout the evolution, which we believe is one of the key reasons for its effectiveness.
>
>
> [1] Y Park et al. “Understanding the Latent Space of Diffusion Models through the Lens of Riemannian Geometry. “ NeurIPS 2023.
>
> [2] A. Tong et al. “Improving and generalizing flow-based generative models with minibatch optimal transport.” TMLR 2023
>
> [3] K. Kapusniak et al. "Metric flow matching for smooth interpolations on the data manifold." NeurIPS 2024
>
> [4] M. S. Albergo et al. “Stochastic Interpolants: A Unifying Framework for Flows and Diffusions”. JMLR 2025

---

> > ### Author Response · Authors · 2025-11-27
> >
> > Dear reviewer mpf9,
> >
> > Thank you for your time and the questions that inspire us to improve the quality of the paper. We humbly ask if our rebuttal has helped clarify these questions. If any points remain unclear or unaddressed, please don’t hesitate to let us know - we are open to clarifying them further before the discussion period ends.
> >
> >
> > Best regards,
> >
> > The authors

---

### Author Response · Authors · 2025-12-02
**Revised Manuscript**

We sincerely thank all the reviewers for their valuable feedback and the time they dedicated to reviewing our manuscript. Their constructive comments have significantly helped us improve the quality of the paper.

We have included the following improvements in the revised manuscript to ensure that all the reviewers’ concerns are properly addressed.


- Clarity:
    - We added a notation table in Table 18 to improve clarity (`sj7x`, W4).
    - We made several minor modifications to resolve the ambiguities raised by reviewers (`mpf9` Q1, `W2yQ` Q3, `sj7x` W3, `PTxT` W1, Q1).


- Methodology:
    - We added a remark before Section 3.2 and a detailed discussion in Appendix A.3 to explicitly show the connection between our methods and latent generative models. This viewpoint offers interesting insight into the behavior of BWFlow. (`mpf9` Q2)
    - We included a detailed discussion about the assumption for statistical modeling of GraphMRF and the corresponding graphical models for GraphMRF in Appendix A.2 (`sj7x` Q2).

- Experiments:
    - We integrated the experimental results for DisCo into the main table. (`W2yQ` W2)
    - We added the Training-time probability path comparisons for planar, tree, and QM9 in Figure 6 to present the consistent existence and varying value of transition points. (`mpf9` Q1)
- Codes:
    - We have uploaded the BWFlow's main pipeline codes in the supplementary materials (`sj7x` W5). We are actively preparing a clean and more consolidated version of the codebase, and commit to releasing it together with the final version of the paper.

For any remaining concerns, we believe they have been thoroughly addressed within the individual reviewer threads. We kindly refer to the corresponding sections for detailed explanations and responses.

We thank all the reviewers again for their encouragement and positive evaluation of our work.

---

### Meta-Review · Area_Chair_eGCV · 2026-01-03

**Summary:**

This paper proposes a new method, BWFlow, for graph generation. Its core idea is to construct a smooth probability path that jointly evolves nodes and edges by representing graphs as Markov random fields (GraphMRFs) and leveraging an optimal-transport–motivated displacement in that space, rather than using conventional independent + linear node/edge interpolations. The authors argue this produces a better-behaved path, leading to improved training dynamics and sampling convergence, and they evaluate the method on plain graph and molecular generation tasks.

Across reviews, there is fairly consistent agreement among the reviewers that the paper is technically well-motivated and grounded in a coherent theoretical framework. Reviewers highlight the clean connection between flow matching, optimal transport, and Bures–Wasserstein geometry including closed-form derivations of interpolation/velocity under the proposed modeling and generally find the paper understandable despite nontrivial mathematical results. Reviewers also view the approach as meaningful for graph generation specifically because it addresses the “path construction” issue in a structured way and is empirically competitive on multiple benchmarks. Overall, I also believe the core idea is timely and relevant to the community, as it tackles a fundamental and increasingly important issue in flow-based and diffusion-based graph generative modeling.

**Reviewer Concerns:**

The main technical concern is computational overhead: multiple reviewers point out that BWFlow can be  more expensive than simpler interpolations, and it is not obvious that the added cost can always be avoided.
Another concern is evaluation completeness and attribution of gains: one reviewer questioned whether the experimental suite aligns cleanly with common baselines/metrics and whether the paper isolates the specific benefit of BW interpolation vs. other factors (and also noted heavy notation and the lack of an explicit code-release statement in the initial version).
Another reviewer raised a related interpretability issue: it is not fully disentangled whether the observed limitations of prior methods come primarily from ignoring graph structure or from using linear interpolations (and asked for clarification around the “ideal” interpolation discussion/figures).

**Reviewer Scores:**

Overall, the reviews are all positive, and the reviews did not present any major unresolved correctness issue. The paper’s theoretical framing of probability-path construction in a graph-structured space, combined with competitive empirical results and a responsive revision, makes it a strong acceptance candidate.

---

### Decision · Program_Chairs · 2026-01-26

Accept (Poster)